# The Fabrication of Micro/Nano Structures by Laser Machining

**DOI:** 10.3390/nano9121789

**Published:** 2019-12-16

**Authors:** Liangliang Yang, Jiangtao Wei, Zhe Ma, Peishuai Song, Jing Ma, Yongqiang Zhao, Zhen Huang, Mingliang Zhang, Fuhua Yang, Xiaodong Wang

**Affiliations:** 1Engineering Research Center for Semiconductor Integrated Technology, Institute of Semiconductors, Chinese Academy of Sciences, Beijing 100083, China; yangliangliang@semi.ac.cn (L.Y.); weijt@semi.ac.cn (J.W.); mazhe@semi.ac.cn (Z.M.); pssong@semi.ac.cn (P.S.); majing@semi.ac.cn (J.M.); yqzhao@semi.ac.cn (Y.Z.); zhenhuang@semi.ac.cn (Z.H.); Zhangml@semi.ac.cn (M.Z.); fhyang@semi.ac.cn (F.Y.); 2College of Materials Science and Opto-Electronic Technology, University of Chinese Academy of Sciences, Beijing 100049, China; 3School of Electronic, Electrical and Communication Engineering, University of Chinese Academy of Sciences, Beijing 100190, China; 4Beijing Academy of Quantum Information Science, Beijing 100193, China; 5School of Microelectronics, University of Chinese Academy of Sciences, Beijing 100190, China; 6Beijing Engineering Research Center of Semiconductor Micro-Nano Integrated Technology, Beijing 100083, China

**Keywords:** micro/nano structures, micro/nano fabrication, femtosecond laser, laser machining, mental, semiconductor, material, application

## Abstract

Micro/nano structures have unique optical, electrical, magnetic, and thermal properties. Studies on the preparation of micro/nano structures are of considerable research value and broad development prospects. Several micro/nano structure preparation techniques have already been developed, such as photolithography, electron beam lithography, focused ion beam techniques, nanoimprint techniques. However, the available geometries directly implemented by those means are limited to the 2D mode. Laser machining, a new technology for micro/nano structural preparation, has received great attention in recent years for its wide application to almost all types of materials through a scalable, one-step method, and its unique 3D processing capabilities, high manufacturing resolution and high designability. In addition, micro/nano structures prepared by laser machining have a wide range of applications in photonics, Surface plasma resonance, optoelectronics, biochemical sensing, micro/nanofluidics, photofluidics, biomedical, and associated fields. In this paper, updated achievements of laser-assisted fabrication of micro/nano structures are reviewed and summarized. It focuses on the researchers’ findings, and analyzes materials, morphology, possible applications and laser machining of micro/nano structures in detail. Seven kinds of materials are generalized, including metal, organics or polymers, semiconductors, glass, oxides, carbon materials, and piezoelectric materials. In the end, further prospects to the future of laser machining are proposed.

## 1. Introduction

Richard Feynman (Nobel Laureate in Physics) gave a famous lecture called ‘‘There’s Plenty of Room at the Bottom’’ which stimulated numerous research projects about the fabrication of micro/nano structures in 1959 [1]. The micro/nano structures are an artificial structure with special features, dimensions, levels, and physical properties on the micro-nano scale [2]. Compared with bulk materials, micro/nano structured materials have different properties. For example, the nanocrystal hardness of copper is 5 times that of bulk materials, and ceramic materials become materials that are easily deformed after reaching nanometer size [3]. In the past 20 years, micro/nano structural materials have had several application fields. In bionics, it is possible to mimic nanomaterials, nanodevices that provide the desired properties in biology or nature. For example, according to the “Lotus Effect”, a natural biomimetic artificial surface having superhydrophobicity, self-cleaning property, low adhesion, and drag reducing property is prepared [4,5,6,7,8]. In the field of biomedicine, micro/nano functional scaffolds for in vitro cell culture are used to mimic a physiological/pathophysiological environment. The artificially controllable micro/nano 3D environment provides a platform for cell biology research and clinical transplantation research which has great research prospects [5,8,9,10]. In the field of materials science, 3D micro/nano structured metamaterials have been designed and manufactured with reasonable mechanical properties [11]. Manipulating fluids in tens of micron-sized channels (microfluidic) has become a unique new field. Microfluidic technology has the potential to influence disciplines ranging from chemical synthesis and biological analysis to optical and information technology [12,13,14,15,16,17,18]. Therefore, the development of microfluidic technology is the product of the development of micro/nano technology. In microelectronics, micro electro mechanical systems (MEMS) and nano electro mechanical systems (NEMS) implement many different microsensors, microactuators, and microsystems. Its mature commercial products include silicon pressure sensors, micro accelerometers, micro gyroscopes, etc., which are widely used in automobiles, mobile phones, and video games [1,19,20]. In addition, micro/nano structured materials are also widely used in photonics, plasma, optoelectronics, photofluidics, and other fields [21].

The preparation of micro/nano structural materials is inseparable from the advanced micro/nano processing technology. For example, photolithography, X-rays, electron beams, ion beams, particle beams, and mechanical methods. Lithography has achieved unique success in creating the ability to create 2D patterns in the range of 10 nanometers to a few microns [22,23,24,25,26,27,28,29,30,31], and is a key technology that has enabled Moore’s Law to expand and revolutionize computing power [32]. Nanoimprinting or injection printing produces a pattern by mechanical deformation of the imprinted resist or by pushing droplets of variable size liquid material (ink) or powder onto the substrate [33]. Despite having nanoscale resolution, the available geometries directly achieved by these means are typically limited to 2D mode. Manufacturing a three-dimensional structure on this length scale has a more challenging prospect [16,34,35,36,37].

In recent years, laser machining has received extensive attention as a new technology for the fabrication of micro/nano structures. The term laser was first proposed in 1957 and represents light amplification by stimulated radiation emission [38]. In 1960, Maiman built the first functional laser at the Hughes Research Laboratory [39]. Laser interaction with matter results in material removal and ultimately formation of micro/nano structures. Since then, lasers have been widely used in a range of scientific and technical fields, and laser-assisted preparation of micro/nano structured functional materials is one of the most important applications. In 1987, Srinivasan et al. first reported the use of ultrafast lasers (lights emitted with pulses shorter than tens of picoseconds) for material processing. Femtosecond (fs) laser micromachining was first demonstrated in 1994 when femtosecond lasers were used to ablate the micron-scale features of silica and silver surfaces. Since 1996, Davis et al. published, for the first time, reports of fs laser writing waveguides in glass. Femtosecond lasers have certain advantages over other longer pulsed lasers, such as nanosecond or picosecond lasers. A key feature of the fs laser is the ability to emit high-intensity pulses in a very short period of time, allowing precise ablation of the material and formation of small heat affected zones. Therefore, nano resolution and accuracy are possible. Compared to other micro/nano structured material preparation methods, laser micromachining has several advantages [16,38,40,41,42]: (1) simpler equipment; no vacuum or clean room facilities required; (2) capable of handling almost all types of materials, including metals, semiconductors, glass, and polymer; (3) Many parameters can be easily adjusted to produce a variety of possible structures. (4) This procedure applies to the surface of any 3D object. Understanding the physical mechanisms of laser machining provides guidance for the preparation of structures with controlled morphology and good quality. The first step in the interaction of the laser with matter is electron absorption of photons. Linear absorption is the primary absorption mechanism of metals, while for semiconductors, dielectrics, and insulators, nonlinear absorption is dominant. The corrugations formed by long laser pulses and continuous wave (CW) illumination are similar to Wood’s anomalies (1902): diffraction of metal gratings at oblique incidence, which can be interpreted according to the surface interpretation proposed by Lord Rayleigh [43,44]. For the ripple generated by short laser pulses, people are still arguing. The details of the process depend on the laser intensity, pulse duration, laser wavelength, and material properties. The physical mechanism is not fully understood. The main ablation mechanisms of fs laser machining are phase explosion, critical phase separation, fragmentation, spallation, melting, vaporization, and Coulomb explosion. The short pulse laser ablation mechanism needs further understanding [38,45]. A more detailed study of the physical mechanism of laser machining can be found in the literature [39,46,47,48,49,50,51,52,53,54,55].

As a new micro/nano structured material processing technology, a review of laser machining has been reported for many decades. In 2013, A. Y. Vorobyev et al. reviewed in detail the new field of direct fs laser surface nano/micro structures and their applications. Direct femtosecond laser surface treatment is one of the best ways to create surface structures in nanometers and micrometers on metals and semiconductors due to its flexibility, simplicity, and ease of use [21]. In 2014, K. M. Tanvir Ahmmed et al. reviewed the latest knowledge of direct femtosecond laser micromachining to fabricate these structures on metals. Discuss the effects of various parameters (such as flux, number of pulses, laser beam polarization, wavelength, angle of incidence, scan speed, number of scans, and environment) on the formation of different structures. A brief review of the possible applications of laser-machined surface structures in different fields is reviewed [38]. Also in 2014, K. Sugioka et al. described the concepts and principles of femtosecond laser 3D micro and nano fabrication, and gave a comprehensive review of the latest technology, applications and future prospects of this technology [56]. In 2015, X. Wei et al. summarizes recent advances in laser-based material processing methods for growing and fabricating one-dimensional (1D), two-dimensional (2D) and three-dimensional (3D) nanomaterials and micro/nano structures. According to the literature reading, most of the previous reviews are about the micro/nano structures prepared by laser machining in a class of materials, and mainly based on short pulse laser machining [57].

In this review, the latest advances in the fabrication of micro/nano structures in metal, organic and polymeric materials, semiconductors, oxides, glass, carbon materials, and piezoelectric materials by laser machining and possible applications of the fabricated structures are described in detail. According to the structural features, we focus on the experimental results obtained by the authors, and analyze the feature sizes of laser machining micro/nano structures, possible applications, laser equipment, and laser machining techniques. Finally, the future development of laser machining is summarized and forecasted.

## 2. Fabricating Micro/Nano Structures on Metal by Laser-Assisted Machining

### 2.1. Nanoparticles or Sphericities

One of the most challenging tasks for preparing metal nanoparticles is still rapid and inexpensive production over a large area. Although very precise methods such as electron beam lithography (EBL) are very versatile, they are costly and time consuming, and most laser interference based techniques are fast but not applicable to any substrate or metal and are not suitable for preparing non-periodic structures. In 2008, M. Mader et al. discussed the patterning of thin metal films by diffraction mask projection laser ablation (DiMPLA). Well-ordered nano-lattices were fast and easy to manufacture and exhibited high uniformity over a large area. Metal nanodots were bonded to the substrate such that they were suitable for subsequent 3D nanostructure synthesis, such as grazing angle deposition and nanowire growth mechanisms [58]. In 2010, M. Mader et al. introduced a gold nanodots matrix prepared by diffraction mask-projection laser ablation (DiMPLA). The rice dots were ordered and can be synthesized on substrates that have hitherto been inaccessible due to the intermediate thin AlO*_x_* layer deposited by pulsed laser deposition (PLD) [59]. In 2009, C. H. Lin et al. reported the efficient fabrication of nanostructures on silicon substrates for surface enhanced Raman scattering (SERS). The silicon wafer substrate in the silver nitrate aqueous solution was directly written by the femtosecond laser to complete the laser-induced photoreduction by generating the lattice-like nanostructures on the surface of the substrate and forming the silver nanoparticles on the surface of the nanostructure in one step [60].

Biosensor based on localized surface plasmon resonance (LSPR) of metal nanoparticles has attracted more and more attention in the past few years. In 2008, L. Vurth et al. reported a technique for two-photon-induced reduction leading to gold deposition. By using a thicker active layer and optimizing the electroless plating step, it can be expected to implement a separate 3D conductive metal structure [61]. In 2011, A. Kuznetsov et al. developed a new method for high-speed manufacturing of large-scale periodic nanoparticle arrays. The method was based on a combination of nanosphere lithography and laser induced transfer. This high inductive performance and fast and inexpensive manufacturing process made the nanoparticle array sensor promising for biomedical applications [62]. Figure 1a showed a new approach to high-speed fabrication of large-scale nanoparticle arrays based on a combination of nanosphere lithography and laser induced transfer. The flat top profile of the laser beam allowed the same illumination conditions and transfer of many nanoparticles for each illuminated particle to be achieved by a single laser pulse. An example of gold nanoparticle structures fabricated on a receiver substrate by a single femtosecond laser pulse was shown in Figure 1b,c. In this experiment, a triangular donor structure was prepared using evaporation of a 1 μm diameter polystyrene sphere and a 45 nm gold layer. A laser energy density of 0.06 J/cm^2^ was used for the transfer process. As can be seen from Figure 1c, the transferred particles had a spherical shape and they were arranged in a hexagonal array.

Due to surface plasmons, various Au nanostructures have been shown to have an enhanced local electric fields around them. In 2018, a novel method was presented by Wei Cao et al. for making Au nanoparticle-decorated nanorod (NPDN) arrays through femtosecond laser irradiation combined with Au coating and annealing. The surface-enhanced Raman scattering (SERS) was affected from the Nanogap and diameter of Au NPs which could be adjusted with thickness-controllable Au films and substrate morphologies [63]. The fabrication procedure for Au nanoparticle-decorated nanorod (NPDN) arrays was depicted schematically in Figure 1d. In short, a linearly polarized fs laser was applied to generate a uniform periodic surface structure on a silicon substrate in air. Then, nanorod arrays were fabricated in water solutions with a 90° rotated polarization fs laser. In the end, Au NPDN arrays were produced by Au coating and subsequent annealing. The prepared SERS substrates were nanorod arrays decorated with abundant Au NPs, which exhibited enhanced optical field due to particle−nanorod intercoupling of localized SPs. A scanning electron microscope (SEM) image of Au NP formed on three types of substrates after Au coating and annealing was showed in Figure 1e–g. The substrate after annealing was covered by Au NP because when heated to a high temperature, most of the film was unstable and tended to form Au NP.

In 2012, M. Tseng et al. have demonstrated the use of laser direct writing (LDW) technology and femtosecond laser to process AgO*_x_* films deposited on glass substrates into Ag nanostructures composed of Ag NP aggregates. This technology had the advantage of processing large area substrates at low cost compared to other reported processing methods. It has used LDW technology to create patterns on substrates with different Raman enhancement levels by varying the laser power. Femtosecond LDW technology has proven to be a flexible and versatile method for processing large area AgO*_x_* films into plasma active substrates. It was expected to be an efficient and economical method for solar energy collection, molecular sensing, plasma photocatalysts and plasma nanolaser applications [64]. In addition to the above preparation of nanoparticle structures, in 2008, C. H. Liu et al. reported fabrication of bimetallic nanostructures using laser interference lithography (LIL) and two-layer resist stripping. LIL was a maskless lithography technology that produced a uniform two-dimensional pattern over a large area. The two-layer resisted stripping process ensures good structural manufacturing quality. The combination of these two technologies showed the potential for low cost large area plasma nanostructure fabrication. The tuning of the surface plasmon resonance (SPR) peak can be applied to many potential areas, such as adjusting the solar cell absorption spectrum to enhance absorption performance [65]. Weina Han et al. used a dual-wavelength fs laser detwetting to perform bimodal (annular) energy deposition on the surface of the sample for direct Au nanostructure fabrication. The proposed method was a new stage in the field of rapid development of large-area nanophotonics. The method can be used for metal coloration as well as high performance and inexpensive printing of functionally ordered substrates for plasmons, field emission devices, solar cells, biosensing, and metamaterials [66].

Dewetting of thin metal films were one of the most common methods of fabricating functional plasmonic nanostructures. However, simple thermally induced dehumidification does not allow the degree of nanostructures to be controlled without additional lithographic process steps. A method of dehumidification induced by a controllable femtosecond laser, without the need for lithography and mass production of plasmonic nanostructures, has been proposed [67]. A single element can be cut to a patch with a dimension greater than the edge uncertainty of the groove (Figure 2a). The thermal conductivity of the silica substrate and air was two to four orders of magnitude smaller than gold, respectively, so the cutting patch was thermally isolated. The dewetting temperature of the film may be much lower than the melting point of lump gold 1337 K. For example, a 10 nm gold film on fused silica dewetted even at 430 K, while a 60 nm gold film dewetted at temperatures below 870 K. The results showed that during the dewetting process, the hot cut gold nuggets were transformed into the same volume of nanoparticles (Figure 2b). An SEM image of the manufactured 2D structure the 30 nm Au film was shown as a reduced periodic sequence between 90° cross-laser scans at a fluence of 90 mJ/cm^2^ in Figure 2c. The laser energy density was chosen to provide a groove that was as narrow and reproducible as possible. Figure 2d shows a spherical nanoparticle array formed after laser dewetting.

In summary, different laser machining techniques such as DiMPLA, LIL, etc. are used to prepare large area metal nanoparticles. These particle sizes are all on the order of micro-nano, and possible applications include 3D nanostructure synthesis, surface-enhanced Raman scattering, plasma nanolasers, etc. As showed in Table 1, nanoparticles prepared by laser machining on metals are summarized.

### 2.2. Nanowires

In recent years, metal nanowire patterns have attracted great attention due to their widespread use in electronics, optoelectronics, and plasma. The need for higher integration densities, better portability and higher performance of modern devices continues to grow, requiring further improvements in mode resolution and quality. In 2007, J. Xu et al. reported the selective metallization of a direct electroless copper plating on an insulator substrate using a femtosecond laser. The geometric parameters of the obtained copper microstructure can be controlled by adjusting the parameters of the laser direct writing and/or electroless plating process. These metal microstructures exhibited good electrical conductivity and strong adhesion [72]. A new method was introduced in 2011 by S. Shukla et al. that was for fabricating highly conductive gold nanostructures in a subwavelength resolution polymer matrix. Features with line widths as small as 150 nm have been produced through this method. It was expected that this new method of in-situ fabrication of plasma and conductive structures will enable efficient fabrication of novel three-dimensional optical functional composite media [73]. In 2015, A. Wang et al. reported a new method for high-resolution nanowire patterning using a spatially modulated femtosecond laser beam that broke the light diffraction limit. A minimum nanowire having a width of 56 nm (1/13 of the holmium laser wavelength) can be achieved. The schematic diagram of experimental setup is showed in Figure 3a. This method provided a low-cost, high-resolution method for the fabrication of next-generation metal thin film devices based on electron, plasma and optoelectronics [74]. It is showed a scanning electron microscope (SEM) image of an ablative film using a single shaped pulse with increased pulse energy (Figure 3b1–b5). Between the two craters, short micro/nano wires are formed in the center. By increasing the pulse energy, the width of the central portion between the ablated regions is reduced until the central portion is completely ablated. The cross section of the central portion measured by atomic force microscopy (AFM) is shown in Figure 3e. As can be seen from the cross-sectional geometry in Figure 3e, a further energy increase squeezes the wire into higher and narrower wires. Figure 3f shows the change in height and width depending on pulse energy. The error bars show the standard deviation of the height, which is the statistics of 20 points made under the same influence. The standard deviation values were all less than 15 nm, indicating good repeatability. The minimum line width available is 56 nm. Figure 3c,d show SEM images of the smallest nanowires with film thicknesses of 20 and 10 nm, respectively. The cross section of the centerline was measured with AFM (Figure 3g).

In addition, in order to explore new ways to prepare nanowires, Wei-Er Lu et al. reported a method for the fabrication of 2D gold nanoparticles structure with excellent electrical and optical properties in aqueous gold ions by multiphoton photoreduction (MPR) with the aid of ionic liquids (IL). A U-shaped terahertz planar metamaterial have been successfully fabricated, whose spectral response was consistent and plays a role in setting theoretical expectations. The IL-MPR nanofabrication protocol was expected to play an important role in fine metal micro/nano structures in MEMS, nanoelectronics and nanophotonics [75]. Silver nanostructures were induced by aqueous solution of silver ions induced by amino acid-assisted femtosecond laser. This technology was expected to play an important role in the fabrication of metal micro/nano structures for further applications in photonics and electronics [76]. In 2018, T. Zhang et al. described the fabrication of a large area of parallel and uniform nanoscale lines covering millimeter-scale deposition areas by laser focusing techniques [77].

In summary, by the use of spatially modulated femtosecond laser patterning has achieved a minimum linewidth of metal nanowires of 56 nm. The prepared nanowires have high electrical conductivity. In Table 2, some key parameters of nanowires prepared by laser machining on metals are showed.

### 2.3. 2D Micro/Nano Structure

#### 2.3.1. Laser-Induced Periodic Surface Structures (LIPSS)

The leaves of certain plants, such as lotus, and parts of certain organisms such as insect wings, exhibit a strong water resistance, known as the “Lotus Effect.” Studies on the morphology of the lotus leaf indicate that the poor wettability of the surface is due to the combined effects of surface micro/nano roughness and chemistry. The preparation of laser-induced superhydrophobic surfaces is a research hotspot in the research of superhydrophobic surfaces.

In 2010, B. K. Nayak and M. C. Gupta observed the interaction of ultrafast laser pulses with matter, resulting in the formation of self-organized tapered micro/nano structures in various metals such as Ti, Al, Cu, and stainless steel. The effects of laser parameters such as fluence, number of shots, and gas environment on micro/nano structure formation have been investigated. The critical flux required for well-developed structure formation depends on the optical and thermophysical properties of the material. This simple metal cone micro/nano structuring approach opened up new areas of research and development activities in biomedical, microelectronics, photovoltaic, and other industrial applications [78]. In 2011, W. Zhang et al. demonstrated that LIPSS on the surface of single crystal superalloy had different levels of laser energy density and nominal pulse number after femtosecond laser irradiation. Microstructural studies have shown that a sudden transition from a low spatial frequency laser induced periodic surface structure (LSFL) to a high-altitude space-frequency laser induced periodic surface structure (HSFL) was caused by a new channel created by the main body of the LSFL ripple [79]. The effect of laser-patterned nanoscale structures on the wetting behavior of silanized stainless steel (SS) and Ti-6Al-4V sheets was studied. The metal substrate was processed with picosecond laser pulses with different laser machining parameters [80]. In 2013, C. A. Zuhlke, T. P. Anderson and D. R. Alexander reported the structure and chemical composition of two unique microstructures with nanoscale features formed on nickel through femtosecond laser surface treatment (FLSP) technology. These two surface morphologies are called pyramids covered by mounds and nanoparticles [81].

In 2009, A. Kietzig, S. G. Hatzikiriakos, and P. Englezos reported that after the femtosecond laser irradiation some double-scale rough structures were produced. Different metal alloys initially showed super-hydrophilic behavior and the structured surface was completely wet. However, over time, these surfaces were almost super-hydrophobic [82,83]. In 2011, M. S Ahsan et al. reported the coloration of stainless steel surfaces by periodic micropores induced by femtosecond lasers and micro/nano gratings on the surface of the sample. Proper adjustment of the laser-induced features on the stainless steel surface provided a variety of colors, including multi-color, gold, and black. Femtosecond laser-induced micro/nanoscale features introduce different colors on stainless steel surfaces was briefly explained [84]. In 2013, S. Moradi et al. studied in detail the effects of femtosecond laser irradiation parameters (flux and sweep speed) on the hydrophobicity of the micro/nano pattern morphology on stainless steel. According to the laser parameters, four distinct nanopatterns were produced, namely nano-corrugations, parabolic columns, elongated sinusoidal columns, and triple roughness nanostructures [85].

Therefore, wettability is an important property of solid surfaces. LIPSS on metal can produce super-hydrophobic metal surfaces, which have a wide range of applications in practical metal waterproofing. This surface has a wide range of applications in national defense, industrial and agricultural production and people’s daily lives. For example, in terms of preventing glass contamination, maintaining the satellite antenna and the radar surface clean, forbidding microbial adhesion and decreasing resistance on the pipe wall.

#### 2.3.2. Pattern or Arrays

The manufacture of conductive micro/nano patterns has become important for various applications in the electronics industry. Conventional vacuum metal deposition and lithography processes are widely used for high resolution metal patterning of microelectronics. However, those traditional methods require expensive vacuum conditions, high processing temperatures, many steps, and toxic chemicals to create a metallic pattern.

In 2011, Y. Son et al. demonstrated that direct femtosecond laser sintering of solution-deposited metal NP enables simple, direct, high-resolution metal nanoscale digital patterning without the need for traditional vacuum deposition and masking processes. Micro/nano-scale metal patterns with high electrical conductivity were produced in an all-digital and mask-free manner at atmospheric pressure and room temperature by precisely controlling the laser conditions. Due to the combination of the short pulse characteristics of the femtosecond laser and the novel thermal characteristics of the metal NP, a significant increase in resolution can be achieved. This approach offered significant advantages for direct, high resolution processes that may be suitable for nanoscale electronic devices [86]. Figure 4a showed the linewidths for various laser scanning speeds (≈ 100–600 μm/s) and laser power (150–400 mW). Uniform metal lines with a minimum width of 380 nm can be fabricated from 780 nm wavelength femtosecond lasers at 400 μm/s and 150 mW. These results indicate that the femtosecond laser selective NP sintering (FLSNS) process can overcome the diffraction limit of light and can directly fabricate metal nanopatterns that were even smaller than the irradiation laser wavelength (780 nm) of metal nanoparticles. The all-digital nature of the FLSNS process enabled the acquisition of any 2D high resolution metal pattern from computer aided design (CAD) data. Figure 4b illustrated 2D nano and microscale metal patterns produced by the proposed method. An Ag micro/nano electrode of 50 nm–10 μm thick can be produced. In 2016, S. Tabrizi et al. fabricated a plasmonic silver nanostructure array with a smooth surface and high conductivity using DLW reduction techniques for the preparation of Functional Optical Plasmonic Resonators [87].

In 2011, On-chip fabrication of silver micro/nano structures was proposed, hereinafter referred to as silver microfluidic arrays (SMA). Photoreduction of femtosecond laser-induced silver precursors was utilized to prepare these SMAs, which consisted of upright nanoplates and attached nanoparticles. The on-chip catalytic reaction indicated that SMA had high catalytic activity. Importantly, the silver matrix also exhibited high SERS enhancement, thus allowing in situ monitoring of the catalytic reaction by SERS detection. The development of this unique SERS active microreactor has shown great potential for studying catalytic mechanisms in a wide range of reactions. The fabrication of silver micro/nano structures in microfluidic channels by femtosecond laser machining will find wide application in chip functionalization and integration [70]. In 2015, C. Constantinescu et al. studied the focusing effects of polystyrene spheres on light and used them as microlens arrays. The microsphere-assisted laser machining method allowed parallel ablation and printing of the metal, generating the formation of negative patterns (nanopores) and positive patterns (nanodrops). The high density filled single layer transparent microspheres on the quartz substrate were covered with a thermally evaporated silver film of controlled thickness [88].

In summary, plasmonic nanostructure arrays, microlens arrays, and microfluidic arrays can be prepared by laser ablation or laser reduction. It can be seen that metal micro/nano structure arrays have broad research prospects. At the same time, the key to achieving functional plasmon resonance that can manipulate light at optical frequencies relies on the production of conductive metal structures with low structural defects at the nanometer scale that are the basis for advanced photonic device fabrication.

### 2.4. 3D Micro/Nano Structure

In recent years, the manufacture of conductive 3D structures on micrometers has caused great interest. Due to their potential applications in microelectronics, or applications in emerging fields such as flexible electronics, nanophotonics, and plasmons. In 2006, A. Ishikawa et al. reported that excitation of dyes by two-photon absorption process allowed low laser power to trigger the reduction of silver ions which achieved sub-diffraction limit fabrication [89]. Conductive silver wires with a minimum width of 400 nm were generated. The authors also demonstrated the fabrication of self-supporting three-dimensional silver microstructures of any shape on a glass substrate. Pulsed laser ablation in liquids has been established as a method for preparing nanoparticles from bulk materials, but it was still insufficient to make anisotropic and complex nanostructures, especially without the use of surfactants. Here, it have presented that silver (Ag) nanosheets can be fabricated by laser regeneration of the primary clusters produced by lasers and by abrupt excimer laser ablation of bulk Ag in water [90]. In 2016, E. Blasco et al. developed a new photoresist consisting of a water soluble polymer and HAuCl_4_ as a gold precursor. The 3D conductive microstructure was fabricated by photopolymerization and photoreduction of gold by DLW [91].

In 2008, M. Rill et al. reported high quality near-infrared frequencies magnetic metamaterials were produced by the combination of direct laser writing with polymer template and silver chemical vapor deposition. Whether the achievable feature size will cause the metamaterial to work at optical frequencies. The answer to this report was positive. This result means that a major obstacle in the field of photonic metamaterials has begun to disappear [92]. In 2009, Z. Kuang et al. demonstrated the fast parallel femtosecond laser surface microstructure using a spatial light modulator (SLM). Create true 3D microstructures on Ti_6_Al_4_V by applying synchronization with the application of computer generated holograms and the scanning system [93]. In 2012, A. Kiani et al. reported the concept of generating 3D nanostructured metal alloys by irradiating two or more immiscible materials powder (such as a mixture of nickel oxide (NiO) and aluminum (Al)) with high repetition of ultrashort laser pulses [94]. In 2011 A. Radke et al. reported for the first time 3D metallic bichiral crystals were prepared through direct laser writing and electroless silver plating. The manufacturing method opened up a way to a complex three-dimensional plasma structure in the optical range, such as a ring structure, with a new type of optical resonance [95]. It can be seen from the preparation process of Figure 5a that a 3D bimetal crystal was fabricated by two-photon femtosecond DLW in a negative-tone photoresist. Next, the dielectric template was coated with a conformal metal film from electroless silver. Finally, the silver plated crystals were separated by a glass capillary and transferred to a clean substrate to facilitate the transmission spectrum because the electroless plating also applied silver to the substrate. Scanning electron microscopy (SEM) images of bi-pigment crystals taken after electroless silver plating were shown in Figure 5b. This oblique view showed the excellent structural fidelity of the two-color spiral crystal.

In 2017, P. Barton et al. employed a femtosecond laser (800 nm center wavelength with 23 fs pulse width) to induce silver nano-reduction in solution to produce silver nanostructures. This method can be applied for laser direct fabrication of metal nanostructures for plasma, which had applications in sonophores and micro-nanoelectronics [96]. This material combination and deposition method was capable of fabricating a 2.5 dimensional silver structure. Between subsequent layers of the scan, 5 μm cubes were prepared by increasing the focus by 100 nm. One-step techniques were presented for direct writing of structured nanocomposites with discontinuous 3D silver nanostructures in the polymer matrix. This technique utilized nonlinear optical photo-material interactions and concomitant photoreduction reactions to produce nanoscale 3D silver structures. Previous work on direct metal writing was applied to generate two-dimensional (2D) structures, low-resolution 3D structures, or stand-alone 3D structures [97]. In 2011, K. Masui et al. demonstrated a new method for fabricating the microstructure of Au nanorod assemblies. The method was based on the cooperation of two optical processes occurring simultaneously; Laser-induced accumulation of Au nanorods, and adhesion of concentrated photons by two-photon polymerization (TPP) of local surface plasmon resonances (LSPR) induction. The purpose of the study was not to create a polymer structure in which metal nanoparticles were dispersed, but to assemble the Au nanoparticles into the desired geometry. To this end, the goal was to apply photopolymerization as a binder to paste the separately dispersed Au nanorods into a micro/nano structure [98]. Futhermore, in 2014, B. Xu et al. reported the programmable assembly of polarized femtosecond laser illumination growth and photoreduction of silver nanoparticles into a three-layer layered micropattern. It was clear that surface plasmon (SP) assisted metal optical nanofabrication technology had true 3D and high precision structural fabrication on nanoelectronic devices and optical metamaterials [99].

In 2015, C. W. Visser et al. utilized laser-induced forward transfer (LIFT) to control the deposition of copper and gold droplets to prepare pillars, lines and fill through-silicon vias. Pillars less than 5 μm in diameter and up to 2 mm in length had low porosity, electrical conductivity, and mechanical uniformity. It was expected that microfabrication of 3D structures can be realized using LIFT [100]. In 2010, J. Wang et al. demonstrated a new non-contact 3D laser direct writing process for the fabrication of 3D self-supporting microstructures. This technology enabled the creation of ultra-thin (<10 μm) pitch bonds and interconnects for LEDs embedded in flexible polyimide substrates. This non-contact 3D laser direct writing technology can match or even surpass current 3D interconnect processes such as wire bonding technology. In fact, the precise printing and assembly of components of different shapes and sizes using this technique can lead to the implementation of electronic and MEMS devices for a wide range of applications while reducing cost and complexity [101]. The basic setup of the 3D laser direct writing process was shown in Figure 6a. The voxel of the same shape and size as the laser spot was placed at a precisely defined location on the substrate by a laser decal transfer process, followed by the ability to produce a self-supporting structure. Primitive colloidal silver nanopastes with high viscosity and shear thinning characteristics required a controlled aging process to achieve high quality laser decal transfer and create a self-supporting structure. Conceptual diagram showed non-contact 3D laser direct writing process. Conceptual diagrams explained the basic steps for the non-contact 3-dimensional laser direct-write process. (i) the laser pulse energy was absorbed at the nanopaste/substrate interface, partially vaporizing the organic solvent; (ii) the resulting vapor pressure caused interfacial debonding at the laser beam focused area and the down shear of the nanopaste film from the surrounding material; (iii) a voxel matching the shape and size of the laser pulse was released; and (iv) the released voxel reached the substrate, forming the desired 3D structure. To demonstrate the capabilities of this printing technique, it has shifted voxels with various geometric shapes, stacked patterns, and aspect ratios (Figure 6b). Figure 6c showed a high aspect ratio 3D micropyramid (100 μm × 100 μm × 60 μm). The high aspect ratio and complex micron-scale 3D structures demonstrated by this technique were important for fabricating spanning and vertical interconnects. Figure 6d was an SEM image of interconnected copper electrodes on silicon. A smaller interconnect assembly was from two circular pressures (radius: 4 μm, thickness: 1 μm) and a rectangular bridged voxel (15 μm × 6 μm × 1 μm).

The preparation of 3D conductive structures by laser machining has been described above, and the structure has potential application prospects in the fields of microelectronics and the like. At the same time, plasma structures and multilayer micropattern structures were generated by femtosecond laser induced reduction. As above, some characteristic information about micro/nano structures on metal is summarized in Table 3.

### 2.5. Other Nanostructures

The 0D, 1D, 2D, 3D metal micro/nano structures prepared by laser machining were described above. Here, hybrid nanostructures and structures that can be fabricated in multiple dimensions using laser process are introduced.

In 2002, F. Stellacci et al. showed that polymer nanocomposites containing metallic nanoparticles, silver salts and suitable photoreducing dyes were effective precursors for direct laser writing of continuous metal structures. These nanocomposites were active for both photon and electron beam stimulation and produced a continuous metal structure. Optical or electron beam induced nanoparticle growth offered a versatile new method to metal patterning from micron to nanometer scale in 2D and 3D. The method can be applied to the fabrication of new types of structures, interconnects and components in electronic, optical and electromechanical devices [102]. In 2008, S. Maruo et al. produced a continuous silver microstructure by three-dimensional (3D) direct laser writing of a femtosecond pulsed laser beam with a polyvinylpyrrolidone (PVP) film containing silver ions. It was demonstrated that both the two-dimensional (2D) continuous metal pattern and the 3-D metal microstructure can be made of a polymer film containing silver ions. These sub-micron line patterns can be used to fabricate optical components such as gratings and plasma devices [103]. S. J. Lee et al. reported that surface-immobilized, densely packed gold nanoparticles were in contact with an aqueous solution of silver ions and exposed to red light, rapidly photoreducing silver ions in solution, producing radial symmetric metal deposits, and diameter ratio. The diameter of the irradiated laser beam was many times larger. The average particle size in the sediment increased with radial distance from the center of the deposit [104]. Schematic diagram of the construction of the samples used in the reported experiments. The symmetry and mass distribution of silver growth was interesting (Figure 7a). At low light doses (4.25 mW, 10 s), the growth showed a symmetrical flower-like pattern of continuous particles that increased evenly as it proceeded from the center. In most cases, the deposit consists of a continuous or at least infiltrated metal (Figure 7b).

In 2016, D. A. Zuev et al. produced asymmetric metal-dielectric (Au/Si) nanoparticles by photolithographic methods by femtosecond laser melting (pulse duration 100 fs, repetition rate 80 MHz) light-responsive hybrid nanostructures. The experimental results established the basis for the application of fs lasers to large-scale fabrication of hybrid nanostructures, which can be applied to efficient light operations, as well as biomedical and energy applications [105]. The authors have demonstrated an approach for the fabrication of hybrid nanodimers via femtosecond laser melting of asymmetrical metal–dielectric (Au/Si). Except for femtosecond laser machining, the preparation process also included e-beam lithography, metal evaporation, lift-off procedure, gas-phase chemical etching as shown in Figure 7c. It was a schematic diagram of the processing process. Firstly, on a properly cleaned substrate of fused silica, plasma-enhanced chemical vapor deposition of SiH_3_ gas was applied to deposit silicon with a thickness of about 200 nm. Next, by e-beam lithography, metal deposition, followed by the lift-off procedures. Cr/Au metal layer with thicknesses of 1 nm/10 nm, 1 nm/20 nm, and 1 nm/30 nm were produced. Then, in the SF_6_ and O_2_ gases environment, through radio-frequency inductively coupled plasma technology, the silicon was etched with metal mask. Finally, femtosecond laser was used to improve optical properties of the Au nanodiscs on the top of the Si nanocones. The typical result of the hybrid nanodimers was showed, demonstrating high repeatability of the fs-laser reshaping. It presented the results of hybrid nanostructure modification with an Au nanodisk thickness of about 10 nm. When the laser fluence (F) achieved the value of 28 mJ cm^−2^, the Au nanodisc begins changed a little. Au nanosphere were created when the fluence was up to 60 mJ cm^−2^. A laser energy density, exceeding the Au nanosphere modification threshold of more than 10 mJ cm^−2^ resulted in melting of the Si nanocone on the boundary with the Au nanosphere and movement of the nanosphere from the center to the side of the Si nanocone (Figure 7d,e). In summary, laser machining is used to prepare metal-dielectric (Au/Si) hybrid nanostructures and to prepare metal deposits by laser reduction of ions in solution.

## 3. Fabricating Micro/Nano Structures on Organics and Polymers by Laser Assisted Processing

### 3.1. Large Area Micro/Nano Structure Arrays

Nanostructure manufacturing technology has presented broad prospect in industries such as integrated circuits, quantum devices, and high-density data storage. Nanoimprinting, electron beam lithography, and focused ion beam patterning techniques were applied. Even so, they had problems with low speed, small pattern areas or high cost equipment.

In 2008, Q. Xie et al. studied the fabrication of maskless nanostructures by laser interference lithography (LIL) using a Lloyd specular interferometer. A high-contrast i-line positive photoresist PFI-88 A6 with a thickness of 100 nm was applied to achieve a stable dose of discrete uniform dot patterns [106]. In 2012, B. Hao et al. revealed a cost-effective and flexible approach to various microlens arrays on polymers that were essential for micro-optical components. The 800 nm femtosecond laser was utilized to control the hydrofluoric acid (HF) acid etching process on the silica glass, and concave microstructures with a smooth curved surface was produced by this method. The microstructured glass template can then be applied as a mold to duplicate the microlenses on the polymer. High order microlens arrays with more than 16,000 hexagonal lenses were fabricated on poly (dimethylsiloxane) [PDMS] [107]. Microscopic hollow tubular structures have attracted significant interest from chemistry and materials scientists due to their new capabilities in the development of advanced equipment and systems. Microscopic hollow tubular structures had applications in special important areas such as catalysis, topical drug delivery, sensors and other microfluidic applications. In 2012, E. Stankevicius et al. demonstrated femtosecond laser fabrication of micro-tubes with a height of several tens of micrometers in the photopolymer SZ2080 by three different methods: direct laser writing, using the optical vortex beam and holographic lithography [108].

In 2013, H. Lin et al. reported a multifocal diffraction-limited non-Ailey mode array using a phase-modulated circularly polarized vortex beam. In particular, by spatially moving the position of the phase eddy currents, an array of splitring (SR) patterns has been created that was capable of fabricating a split ring microstructure array in a polymeric material by a single exposure of a femtosecond laser beam [109]. In 2014, C. L. Sones et al. reported a rapid laser-based method for the construction of polydimethylsiloxane (PDMS) on the micron scale. This maskless method utilized a digital multi-mirror device as a spatial light modulator to prepare a given spatial intensity pattern [110]. In 2014, B. P. Chan et al. reported femtosecond laser-based free-writing of complex protein microstructures and micropatterns, with sub-micrometer features and controllability over voxel dimension, morphology, and porosity [111]. In 2015, X. Liu et al. reported the fabrication of SERS nanopillar substrates by laser-assisted replication and subsequent gold film deposition. The substrate was composed of an array of gold coated cyclic olefin copolymer nanopillars having a variable diameter of 60 to 260 nm. Several different nanopillar arrays have been produced to study the effect of nanocolumn diameter and spacing on Raman enhancement factors [112].

In 2017, J. Ni et al. presented a method that three-dimensional controllable chiral microstructures were generated inside the isotropic polymer using the helical phase wavefront and the plane wave interference beam. In an isotropic polymer, 3D chiral microstructures were achieved under illumination with the use of a coaxial interference femtosecond laser beam [113]. The experimental setup for the fabrication of chiral microstructures was showed in Figure 8a. A femtosecond laser at a wavelength of 800 nm was used, which was adjusted to a linearly polarized Gaussian beam. The beam reflected from the spatial light modulator (SLM) showing interfering vortex holograms (IVH) had a spiral phase wavefront and a spiral lobe pattern. To remove the spiral pattern from other diffraction orders and control the ideal spatial shape of the reflected beam, a circular tilt shifted phase hologram was added to the hologram. The shape and size of the chiral microstructures needed to be controlled to improve manufacturing parameters. As we can see in Figure 8b,c, a series of chiral microstructures having three helical blades can be produced with a fixed laser intensity and varying exposure times from 1 to 5 s. Although the diameter of the chiral structure increased with laser power or exposure time, the structure kept chiral with one spiral protrusion. Their diameter and height maintained spiral blades and chirality. When the 532 nm excitation light illuminated the array, a uniform speckle pattern diffracted on the received screen, which indicated a significant uniformity of the large area chiral microstructure array.

In 2018, Y. Zhang et al. reported a microscale local morphology with a reversible structure was produced by using a uniform, low-cost shape memory polystyrene film, which assisted a sensible controlled femtosecond laser scanning strategy to achieve symmetric and asymmetric growth. The structure had potential applications in particle capture/release and information encryption [114]. Shrinkage properties of polystyrene film was showed in Figure 8d. The polystyrene film shrank when it was heated with a heat gun. Laser-induced polymer self-growing (LIPS) of polystyrene film and mechanism of microcolumn formation: femtosecond laser machining had a heat affected zone with a high temperature to facilitate local operation of the polystyrene film. Here, the polystyrene film was treated with a femtosecond laser writing arrangement as schematically illustrated in Figure 8d. The laser was focused on the surface of the sample and moved along a pre-programmed 2D path during the manufacturing process. When using a repeating circular scan path of 1–80 turns, the microcolumn grew up, corresponding to the basic growth of the sunflower, as shown in Figure 8d. Just like a sunflower can bend to the sun. Based on an understanding of the bending mechanism and experiment, the manufacturing parameters were optimized to achieve a curved microcolumn (Figure 8e). Various ordered patterns were fabricated by using a curved microcolumn as a basic unit (Figure 8f). According to the above description, laser interference technology, femtosecond laser ablation, and other processing techniques can prepare large-area organic array structures, such as uniform dot patterns, microlens arrays, and microtube arrays, etc.

### 3.2. 2D/3D Polymer Structures by Two-Photon Polymerization Lithography

In 2002, T. Tanaka et al. reported the sub-diffraction limit (SDL) micro/nano processing by two-photon absorption photopolymerization. It was possible to find SDL spatial resolution in the threshold system. Experimentally, lateral spatial resolution down to 120 nm was achieved by using high numerical aperture optics [115]. Two-photon photopolymerization provided a simple, fast and highly accurate method for customizing a variety of complex 3-D polymer micro/nano structures. Spatial resolution can be compared to UV lithography. In 2003, H. Sun et al. realized spatial resolution much smaller than the diffraction limit by using nonlinear laser-substance interactions [116]. This technology made it attractive for nanophotonic and optical microelectromechanical (MEMS) applications. In 2004, T. Baldacchini et al. described an acrylic based prepolymer resin that was suited for the fabrication of three-dimensional structures using two-photon polymerization and a tower with 1.5 mm tall and only 20 μm on each side was produced. The photochemical and photophysical properties of the photoinitiator were characterized and presented a representative structure which demonstrated the advantageous mechanical and optical properties of the polymer [117]. In 2008, A. Ovsianikov et al. demonstrated two-photon polymerization (TPP) photonic crystals on zirconium propanol sol-gel [118]. 3D photonic crystals were prepared by TPP. The manufactured three-dimensional photonic crystal structure exhibits high resolution and clear band resistance in the near-infrared region. As shown in Figure 9a,b, Three-dimensional photonic crystal structures fabricated by the TPP technique demonstrated the advantage of zirconium-containing materials over other TPP materials for their negligible shrinkage.

In 2008, Z. Sun et al. presented a simple in-situ synthesis strategy for semiconductor-polymer nanocomposites and a microbull with a feature size of 20 μm was produced. Multicolor 3D micromachining of nanocomposites was demonstrated, combining the in situ synthesis of multiphoton photopolymerization-microstereolithography (MPP-MSL) with semiconductor nanoparticles in a polymer matrix with fine size control [119]. In 2008, P. Tayalia et al. proposed standard protocols for micromachining and cell culture, three-dimensional imaging techniques and quantitative analysis methods for studying 3D cell migration and comparing 2D and 3D cell migration. The scaffolds used in this study were interconnected stake structures with square shapes and apertures was 12, 25, 52, and 110 μm. The three-dimensional micromachining of two-photon polymerization utilized the spatial selectivity of two-photon absorption to prepare sub-micron resolution 3D polymer structures of any shape and size [120].

In 2010, H. Xia et al. reported that photopolymerizable iron fluorides were prepared in order to produce smart micro-nano machines with microscopic operating characteristics. Micron-sized springs and turbines were created for magnetic remote control. The combination of photopolymerizable ferric fluorinated resins and laser machining technology will enable nanotechnology breakthroughs to easily manufacture and remotely control micro/nano machines in a wide range of applications [121]. The procedures for the manufacture and remote control of micromachines were illustrated in Figure 10a. First, according to the literature, Fe_3_O_4_ nanoparticles were compounded. The surface of the synthesized Fe_3_O_4_ nanoparticles was modified by 3-(trimethoxysilyl) propyl methacrylate (MPS) to uniformly embed these magnetic nanoparticles in the photoresist. The manufacture and remote control of micro/nano machines was not limited to microsprings, but microturbines for remote control have also been developed. As shown in Figure 10b1–b4, a collared microturbine was established based on a pre-designed model. The microturbine had a diameter of approximately 35 μm and had a central axis and three blades. A ferromagnetic body was placed on a vortex device around a microturbine for magnetic control. The development of microturbines were on behalf of another example of remote control of micro/nano machines, and it was believed that microturbines had considerable significance in the field of microfluidics.

In 2012, T. Baldacchini et al. described fabrication of microstructures by two-photon polymerization using femtosecond laser pulses. The mechanism of TPP in acrylic-based resins containing commercially available photoinitiators was elucidated [122]. The TPP excitation which was from the 80 MHz fs laser pulse sequence was transformed to a series of excitations that motivated the fs laser pulse through a unique device that cut off the laser beam. Femtosecond laser fabrication of micro/nano structures by two-photon polymerization (TPP) was an effective method for achieving high resolution 3D devices for photonics, micromechanical systems and LOC technology. Compared to existing microfluidic filter manufacturing techniques, 2PP provided precise control of direct writing of 3D porous structures with submicron resolution, simple processing and ease of integration. In 2010, A. Zukauskas et al. reported a 3D microstructure made of a mixed zirconium-silicon hybrid sol-gel photopolymer ORMOSIL (SZ2080) (provided by Foundation of Research and Technology Hellas, bmm@iesl.forth.gr). A doped artificial scaffold for cell growth was prepared to facilitate observation of cell proliferation. An example of distributed feedback dye laser (DFBL) resonator model fabricated out of ORMOSIL doped with rhodamine 6G (R6G) with linewidth of 280 nm and period of 850 nm was fabricated [123]. In 2010, Z. Liu et al. demonstrated microdisk cavities were fabricated by using the TPP technique. The surface roughness was less than 12 nm. This carved out the way for the fabrication of complex three-dimensional microcavities based on optical resonator-based fundamental and applied physics research [124]. In 2011, J. Ku et al. reported presented whispering-gallery-mode microdisk lasers were fabricated by femtosecond laser direct writing of dye-doped resins. The diameter of the disc, the upper and lower ends of the cone were 20, 14 and 8 μm, respectively. The stent was empty inside and had a 3 μm thick wall. This work showed that laser micro/nano processing technology was not only suitable for passive micro-optical components, but also played an important role in the manufacture of active optoelectronic devices and their integrated photonic circuits [125]. In 2012, L. Amato et al. reported the integration of size-based three-dimensional filters and micron-sized holes in commercial microfluidic chips. The aperture size was 1.3 μm × 1.3 μm in the *yz* plane and 1.5 μm × 1.5 μm in the *xy* and *xz* planes. The minimum width of the polyline in the *yz* plane was 500 nm. Photopolymerizable polylactic acid (PLA) was applied to produce scaffolds by TPP and soft lithography [126].

In 2012, A. Koroleva et al. explored the production of macroscopic scaffolds defined by two-photon polymerization (TPP) and their application as nerve tissue engineering scaffolds. Soft lithography replication increased the productivity of 3D stents. The 2PP-made bracket consisted of a hollow cylinder of hexagonal arrangement in each layer. Each individual cylinder of the stent had a wall thickness of 20 μm and a diameter of 100 μm. The total height of the stent was 300 μm [127]. Polymer materials have attracted a lot of interest due to their low cost and good biocompatibility, both of which are ideal for microcavity applications. In 2015, A. Zukauskas et al. reported a study of polymer cross-linking assays, namely the degree of conversion and refractive index of microstructures prepared by two-photon polymerization (TPP). The effects of TPP processing parameters such as laser intensity and scanning speed were investigated [128]. Accurate assembly of carbon nanotubes (CNTs) in any 3D space with proper alignment was critical and desirable for CNT applications, but remained a long-standing challenge. 3D micro/nano CNT/polymer structures can be prepared using two-photon polymerization (TPP) technology. In 2013, Z. Liu et al. reported that a high performance asymmetric polymer microcavity was designed and produced through TPP technology. These high performance polymer microcavities had potential applications in low threshold micro-cutting and high sensitivity optical biosensing [129]. The asymmetric microcavity was laser fabricated using TPP directly, as shown in Figure 11a. Figure 11b depicted an oblique view scanning electron microscope (SEM) image of an asymmetric disk shaped microcavity after the manufacturing process. In order to reduce the effect of maintaining the microcolumn of the polymer disk on the cavity mode, the diameter of the microcolumn was designed to be much smaller than the diameter of the microcavity.

In 2016, X. Wei et al. proposed a method to combine a large number of well-aligned multiwalled carbon nanotubes (MWNTs) into 2D/3D polymer structures by TPP lithography [130]. In Figure 11c,d, experimental procedure was illustrated for preparing an MWNT-thiol-acrylate (MTA) composite resin for TPP manufacture. By directly adding different amounts of acid purified MWNT powder to a self-made TPP compatible thioacrylate resin, the MTA resin was produced. Using TPP lithography, the fs laser beam (center wavelength 780 nm, pulse width 100 fs, repetition rate 80 MHz) was focused into the MTA resin, and 3D scanning was performed according to the geometric user design to generate micro/nano structures with MWNTs simultaneously were incorporated into the interior of the polymer. Before TPP lithography, Electrode patterns were fabricated on the substrate to provide electrical contact to the MWNT-based micro/nano structures for device fabrication (Figure 11e). Some examples of 3D microstructures made of MTA resin included microcapacitor arrays, microcoil inductors, helical photonic crystals, micro-cones and micro-gears in Figure 11f–j. Various functional micro/nano structures can be prepared by TPP lithography using MTA composite resin.

### 3.3. 2D/3D Polymer Structures by Direct Laser Writing (DLW)

In the past decade, direct laser writing has existed from laboratory curiosity to mature multi-functional reliable lithography tools, which can be regarded as three-dimensional counterparts of planar electron beam lithography. Horizontal feature sizes as low as about 100 nanometers have become commercially available standards. In addition, the use of stimulated emission depletion excitation opened up the idea of obtaining line widths as low as tens of nanometers by direct laser writing (DLW) [131].

In 2004, M. Deubel et al. reported theoretical and experimental results for slit beam shaping configurations for the fabrication of photonic waveguides using femtosecond laser pulses. Most importantly, this method supported focusing the objective with a longer depth of field and allowed direct writing of microstructures with a circular cross section using a vertical writing scheme. The in-plane rod distance of several samples produced ranged from 1.5 to 0.65 μm. There was a residual roughness of about 10 nm at the bottom [132]. In 2005, M. Hahn et al. proposed a versatile confocal-based laser scanning lithography method for controlled, high fidelity 2D and 3D surface patterning of polyethylene glycol diacrylate (PEGDA) hydrogels. In the current work, the surface properties of the deformable, biocompatible PEG-DA hydrogel have been modified in a controlled manner using laser scanning lithography (LSL) to achieve pattern feature sizes as low as at least 5 μm [133]. In 2005, K. Seet et al. reported an extended periodic microstructure of a two-photon DLW having a square and circular 3D helical structure in photoresist SU-8, which was not suitable for widespread use of layer-by-layer fabrication [134]. DLW recording of 3D patterns was exhibited in Figure 12a. At the focal region of a tightly focused laser beam, a permanent photomodification which was sown by multiphoton absorption, was induced in the optically transparent dielectric material. By the multiphoton absorption point-spread function, the photomodification was localized within the spatial domain defined. Arbitrarily shaped patterns can be drawn through translating the sample. The cross-sectional area of the spiral arms resulted from its shape from the focal region that was an ellipsoid. A detailed view of the individual spirals was presented in Figure 12b,c, on which smooth turning points and surfaces of low roughness were obvious. In 2006, B. Kaehr et al. report nonlinear, direct-write protein was prepared with the use of a 532 nm Nd:YAG laser. As with previous studies using femtosecond titanium: sapphire lasers costing more than $100,000, physical strength and chemically responsive microstructures can be fabricated with feature sizes less than 0.5 microns. Current experiments presented that nonlinear excitation was available in photocrosslinking proteins into well-defined three-dimensional matrices [135].

In 2009, A. Kuehne et al. studied the spectral properties of the new oligofluorene truxene photoresist composites and showed the ease of processing of the composite through direct laser writing (DLW) of the waveguide structure, thus making the structure tunable size down to 2.5 μm, smooth sidewalls and high aspect ratio. This approach created a method for the fabrication of photonic device structures, including single photon sources, waveguides, and lasers [136]. In 2010, M. Thiel et al. reported the use of continuous-wave lasers to study conventional DLW to prepare 3D photonic submicron structures, better than previous best results over femtosecond laser pulses centered at 780 nm. The body-centered cubic wood photonic crystal consisted of 24 layers with a rod spacing as small as 450 nm [131]. In 2011, F. Klein et al. reported that custom 3D cell culture scaffolds in the micron to nanometer range were fabricated using direct laser writing (DLW). Composite polymer scaffolds with different protein binding properties were fabricated by continuous DLW of two different photoresists [137]. Direct laser writing (DLW) has recently been shown to be a versatile technology for manufacturing custom 3D cell culture scaffolds in the micron to nanometer range. The polyethylene glycol diacrylate (PEG-DA) scaffold had 4.8% of pentaerythritol tetraacrylate (PETA) composing of columns (about 7 μm in diameter and about 23 μm in height) connected by two different heights (1 μm in diameter) at a different height, offset by 10 μm (Figure 13a) using DLW. After development in isopropanol, the stent was cast with a second photoresist Ormocomp. Then, DLW process needed to be aligned in three dimensions relative to the first lithography step. To realize this, an Ormocomp cube with an edge length of 2.5 μm is attached to the middle of the PEG-DA beam by polymerization (Figure 13b,c). In 2011, M. Malinauskas presented that 3D polymeric structures with resolutions as low as a few hundred nanometers can be produced with high repetition rate ps laser pulses. The combination of low cost ps laser setup and high repetition rate was promising for the practical implementation of 3D polymerization in photonics, micro-optics and biomedical [138].

In 2012, Y. Sun et al. reported the application of maskless femtosecond laser direct writing to build protein microlenses. Due to its dynamically adjustable properties and complete biocompatibility, microlenses showed great potential for optical, electronic, and biomedical applications. The characteristic size of the protein system was dependent on the laser power density. Under the optimized conditions, the minimum feature size of the line width was about 250 nm (exposure time is 1000 μm, laser power density was about 20 mW μm^−2^, and bovine serum albumin (BSA) concentration was 400 mg mL^−1^) [139]. In 2013, M. Steger et al. proposed a method for creating a layered structure by two subsequent steps of laser ablation. Structural sizes ranged from 100 nm to micrometers. As the periodicity of the structure decreased, the process window also decreased, and the resulting structure had a lower aspect ratio. In order to build some functionalization of the surface, a hierarchical structure was required. These structures can be as microstructures with superimposed nanostructures [140]. In 2014, D. Wu et al. proposed a new hybrid femtosecond laser micromachining method. This novel microchip was fabricated by integrating various 3D polymer micro/nano structures into a flexible 3D glass microfluidic channel. Glass/polymer composites 3D bottled microchips costed more than paper microchips, but they were critical for specific applications, such as mechanism studies of 3D cell/tissue growth and control, fluid physics/chemical authenticity 3D microenvironment medical practices in military or aerospace systems, challenging and resource-constrained environments, require strong medical technology to minimize the burden on personnel and transport machinery [141]. Figure 14a showed a schematic diagram of a manufacturing process for making a biochip in a bottle by mixing fs laser micromachining. It involved two main steps. The first was the fabrication of 3D hollow microchannels by femtosecond laser-assisted wet etching (FLAE) photosensitive Foturan glass (Figure 14b,c). Surface smoothness was improved by thermal annealing (Figure 14d). The 3D polymer microstructure was then prepared by TPP for chip functionalization (Figure 14e). Figure 14f presented a 45° (top) and 0° (bottom)-tilted SEM image of a cross section of a fabricated microchannel showing the shapes of rectangles, circles, ellipses, pentagons, triangles, and hexagons.

In 2014, A. Greiner et al. reported the use of DLW to fabricate 3D cell culture scaffolds with different pore sizes on glass substrates or microporous polymer membranes. The resulting cell culture scaffold provided a universal tool for studying cell migration and invasion [142]. In order to incorporate a chemotaxis assays into our matrix invasion model, our 3D scaffolds were united with a Boyden chamber system. In this approach, the pentaerythritol tetraacrylate (PETTA) scaffolds were prepared by DLW directly on commercially available microporous polymeric films using a newly-developed sandwich approach (Figure 15a). 3D structures with different mesh sizes (7 μm, 8 μm, and 10 μm) were written onto a film with a pore size of 3 μm, which produced microporous 2D/3D composite structures (SEM images in Figure 15b) suitable for chemotaxis and invasion study. In 2014, J. B. Mueller et al. observed a change in local refractive index over time, which was related to the formation of polymer volume elements (voxels). By comparing the effects of various parameters on the reaction (exposure time, photoinitiator, monomer, oxygen concentration, and write laser power), new insights were proposed about the polymerization kinetics in the case of 3D DLW [143]. Liquid crystal elastomers (LCEs) were one of the best candidates for smart man-made materials, because they were capable of reversible shape changes in response to external stimuli, such as temperature, electric fields, or light. In 2014, H. Zeng et al. reported that 3D liquid crystal elastomer structures—rings, stakes, etc. which were prepared by direct laser writing with two-photon absorption, with submicron resolution and maintaining the molecular orientation of the design. The first layer of the prepared wood pile structure was in contact with the glass and had an average width of 0.7 microns. The second and third layers had an average width of 1 micron [144].

In 2012, T. Buckmann et al. presented a rationally designed true three-dimensional microstructured mechanical metamaterial with a custom Poisson’s ratio that was large enough for direct mechanical characterization. This has been made possible by the use of new techniques for immersive 3D direct laser writing optical lithography [145]. In ordinary 3D-DLW, the achievable height of the structure is limited (Figure 16a). Generally, the total height of a DLW structure is only a few tens of microns. To properly characterize mechanical metamaterials, larger structures are often required. For applications, larger structures may be required. Here, a new method called “immersive” 3D-DLW was applied (Figure 16b). Hence, structures with an overall height of up to sub-millimeter characteristic dimensions became possible for the first time. This structure was meaningful for biological cell culture studies. The total volume of the metamaterial sample fabricated using immersion in 3D-DLW lithography was 100 μm × 100 μm × 200 μm, and for all designs, the height was 10 μm and the width was 10 μm (Figure 16c). In 2015, D. Wu et al. developed a technology: in-channel integration of flexible two-dimensional (2D) and three-dimensional polymer micro-optics in glass microfluidics: hybrid femtosecond laser micromachining supported by plate supports. A stent having an optimum thickness of 1–5 μm was produced on the lower inner surface of the microfluidic channel. This flat scaffold-supported hybrid femtosecond laser microfabrication (FSS-HFLM) method was promising for the integration of various 1D to 3D passive and active micro-optics, including polymer optical switches, wavelength division multiplexing and on-chip microlasers for lab-on-a-chip optical flow control applications, and optical amplifiers [146]. 

In recent years, nanowire or nanofiber based biosensors have attracted great attention and efforts in many fields. In 2015, Y. Sun et al. demonstrated a simple femtosecond laser direct writing (FsLDW) method for generating whole-protein single nanowire optical biosensors from aqueous inks of probe proteins. The minimum line width of the high quality protein nanowires achieved was about 150 nm [147]. Microlenses have attracted much attention due to their wide application in the field of micro-optics. For example, micro microlenses having feature sizes of hundreds of micrometers to tens of nanometers have been used in optical communication, miniaturized optical systems, laser beam forming and steering, and optical sensing, as well as various biomedical imaging. In 2015, D. Lu et al. reported the fabrication of femtosecond laser direct writing (FsLDW) technology and in-situ integrated PDMS microlenses such as spherical microlenses and aspheric hyperbolic microlenses. Optical focusing tests were performed using PDMS hyperboloid microlenses with a radius of 20 μm, a height of 3.5 μm, and a theoretical focal length of 117 μm [148]. Ultrafast pulsed lasers were proven a technology for building custom free-form 3D micro-frames from proteins to biocompatible glass and a variety of materials. In 2015, J. Maciulaitis et al. reported preclinical studies of 3D artificial microstructure scaffolds of mixed organic-inorganic (HOI) material SZ2080 made by DLW technology and the microstructure scaffolds were tested for biocompatibility in rabbits [149]. The employed ultrafast laser offered 300 fs, 200 kHz, and 515 nm pulsed light radiation. Sample positioning synchronization with beam deflection was achieved by a custom assembled femtosecond laser system for the laboratory. The material that was chosen for the scaffolds was a HOI sol-gel photopolymer SZ2080, consisting of 20% inorganic and 80% organic parts.

True three-dimensional (3D) integrated biochips were critical to achieving high performance biochemical analysis and cell engineering, which remained the ultimate challenge. Direct laser writing (DLW) based on superlocal polymerization was a way to generate three-dimensional (3D) micro/nano structures for various applications in science and industry. In 2014, S. Rekstyte et al. described the DLW results of the hybrid organic-inorganic material SZ2080 on optically opaque and reflective surfaces [150]. Studies have shown that even on glossy or rough surfaces, 2D and 3D structures with spatial resolutions below 1 μm can be produced using sample translation speeds of up to 1 mm s^−1^. In 2014, M. Malinauskas et al. proposed a combination of fused filament fabrication 3D printing and DLW ablation for efficient 3D micromachining of polylactic acid (PLA) biocompatible and biodegradable polymer scaffolds. The microporous structure made of polylactic acid (PLA) had a controllable porosity (20–60%) and consisted of the desired volume pores (0.056 μm^3^). The primary goal of this study was to prepare artificial scaffolds with microfeatures of PLA characteristics that were suitable for stem cell growth studies and regenerative medicine applications [151]. In 2017, N. Mnson et al. demonstrated the fabrication of polymer nanopillars (NP) using 3D direct laser writing (3D DLW). It have been shown that polymer NPs generally provided a biocompatible environment for cell adhesion and sustained cell growth, therefore, synthetic nano- and micro-sized structures were of great importance in the field of tissue engineering [152]. Scanning electron microscopy (SEM) images of an array of 3 μm long NPs of all pitches tested in the cell studies described were shown in Figure 17b–g. It showed an overview of a 250 × 250 μm^2^ NP array implemented by TPS (Figure 17a). The illustrations corresponded to enlarged and oblique views of the same array. Figure 17b–g presented enlarged views of several arrays with center-to-center spacing of 1.5, 2, 3, 4, 6, and 12 μm, respectively. In the end, as showed in Table 4, 3D micro/nano structures prepared by laser writing directly on organics and polymers are summarized.

### 3.4. LIPSS

Polytetrafluorethylene (PTFE) was a bioscience and medical application material with biocompatibility, non-flammability, anti-sticking and heat resistance. In 2003, Z. B. Wang et al. demonstrated fast structuring of PTFE in ambient air with ultrashort pulses at a high repetition rate. The laser energy density was controlled below the critical energy density of 1.2 J cm^−2^, which can generated micron-quality microstructures [153]. In 2003, D. B. Wolfe et al. demonstrated a pulsed, femtosecond Ti:sapphire laser to write patterns in PDMS stamps for employ in soft lithography and to customize a microfluidic channel defined in PDMS. The process was a non-lithographic technique which was applied to fabricate PDMS stamps with widths as small as 1 µm and in a large area (>20 mm^2^) was a pitch of 2 µm [154]. In 2005, T. N. Kim et al. reported the use of focused femtosecond laser pulses to drill small diameter, high aspect ratio microcapillaries in a molded PDMS microfluidic device. This technology allowed for the fabrication of wider channel structures than molding alone, and will enable new microfluidic applications. Laser ablation was employed to drill micropores in the walls of molded PDMS channels with diameters as small as 0.5 μm and aspect ratios of up to 800:1 [155]. In 2011, S Turunen et al. reported photocrosslinking of protein microstructures induced by pico and femtosecond lasers. Submicron and micron structures can be made by multiphoton photocatalytic photocatalytic cross-linking of avidin, biotinylated BSA, and BSA using a non-toxic biomolecular flavin mononucleotide (FMN) as a photosensitizer. Avidin can be crosslinked into a surface pattern when the protein concentration was 400 mg mL^−1^ and the photosensitizer content was 1–4 Mm [156].

Fluoropolymer films were researched for developing new technologies and their numerous applications such as superhydrophobicity, self-cleaning, anti-fog, and anti-sticking, due to their extremely low surface energy. In 2011, C. Becker et al. achieved superhydrophobic surface with contact angle (θ = 160°) on a PET substrate using laser assisted magnetron sputtering [157]. In 2011, P. H. Wu et al. presented fast replication of large-area femtosecond laser-induced surface micro/nano structures on plastic parts by injection molding having a period of 600–700 nm, which were smaller than the laser wavelength at 800 nm in free space. The injection molded polypropylene part had a remarkable hydrophobicity comparable to that of a femtosecond processing mold [158]. In recent years, periodic nanostructures played an important role in key components of optical applications. In 2011, T. Yao et al. proposed an anti-reflective nanostructure fabrication method using a stainless steel template Polycarbonate (PC) film hot stamping. The anti-reflective nanostructures on the stainless steel surface whose period was 600–700 nm were produced by a femtosecond laser (wavelength 800 nm). Fabrication of the nanostructures from the template to the polycarbonate (PC) film was then generated [159]. In 2012, D. S. Correa et al. described how micromachining of polymer materials on a surface or volume can be applied to modify its optical and chemical properties, and was expected to be used in the fabrication of waveguides, resonators, and self-cleaning surfaces. Then, how to dope a basic resin to a molecule having biological and optical properties of interest to produce an active microstructure using two-photon absorption polymerization was discussed. Such microstructures can be used to fabricate devices with optical applications, such as microLEDs, waveguides, and medicine, such as stents for tissue growth. Manufacturing microstructures ranged from 1 to 100 μm with energy from 0.05 to 0.1 nJ [160].

In 2013, B. K. Nayak et al. reported a method that superhydrophobic surfaces were fabricated by replicating micro/nano structures on to poly(dimethylsiloxane) (PDMS) from a replication master made by ultrafast-laser machining [161]. No additional coating was required on the PDMS to realise a contact angle greater than 154°. The contact angle can be controlled by varying the height of the microtexture in the PDMS, and the replication surface entering the Cassie regime required a minimum texture height of about 4.2 μm. In 2014, J. Shao et al. presented a method for fabricating covered micro/nano-scale combined layered structures over large areas by nanoimprinting and improved laser swelling technology. The micro/nano structure can make the surface superhydrophobic [162]. As shown in Figure 18, the manufacturing process began with the preparation of two polymer films on a glass substrate for nanoimprinting and laser swelling, respectively (Figure 18a). After the preparation of the polymer film, a nano-imprint method was applied to fabricate a secondary structure of the graded surface (Figure 18b). During pattern transferring process, the PDMS soft mold replicated from the structured silicon wafer was employed to ensure uniform contact between the substrate and the mold (Figure 18c). Finally, the main structure was generated using laser expansion (Figure 18d). Due to the careful selection of materials and laser sources, changes in secondary structure were negligible and form a covered layered structure. An array of larger layered dome structures covered with micropillars were showed in Figure 18e,f. The distance between two adjacent regions of the layered structure was 180 μm as determined by the laser scanning process. For each layered structure, the main dome structure had a height of 26 μm and a diameter of 141 μm, while the auxiliary microcolumns had a height of 4 μm, a diameter of 3 μm, and a period of 7 μm. It can form a smaller dome structure covered with nanopillars by imprinting with a nanopillar mold and focusing the laser beam to a smaller point.

For microfluidic applications, the ability to fabricate tapered microchannels with customizable cross-sections in a variety of materials is highly desirable. In 2012, S. Darvishi et al. explored ultrafast laser machining of tapered microchannel trenches in hard (sodium-calcium and borosilicate glass) and soft (PDMS elastomer) transparent solids. The tapered channel was prepared using an ultrafast laser with a wavelength of 800 nm and a Gaussian beam profile focused on the surface of the sample. The geometry and dimensions of the channel cross section were then evaluated using an optical microscope [163]. PMMA mimics soft materials and was biocompatible and stable in physiological environments. It has been applied in a variety of applications, such as cosmetic surgery, dentistry, and ophthalmology. Surface structures on the micro and nano scales will improve cell adhesion and proliferation at the polymer-tissue interface, resulting in enhanced tissue integration leading to better implant binding. In 2013, F. Baset et al. studied parameter on the porosity of the PMMA surface by laser ablation and indicated that it depended on the pulse energy, the number of laser shots, and the scan speed. The porous area fraction decreased as the pulse energy increased, and the pore size distribution peaked at a higher energy at a pore area of 0.037 μm^2^. For deeper craters, the porosity evolved into a 3D honeycomb structure on the sidewalls [164].

## 4. Large Area Micro/Nano Structures on Semiconductors by Laser Assisted Processing

### 4.1. Laser-Induced Periodic Surface Structure

In 2007, M. Couillard et al. analyzed in detail the microstructure under the InP surface after laser irradiation of single femtosecond and multi-femtosecond laser pulses. High spatial frequency LIPSS (HSFL) was observed at 460–480 nm in the annular region of the ablation pit. Low spatial frequency LIPSS (LSFL) with 1670–1790 nm close to but lower than the laser wavelength, were visible near the center of the crater. Cross-sectional transmission electron microscopy TEM studies of laser-induced periodic surface structures were reported for the first time. The extent and nature of subsurface modifications caused by single femtosecond laser pulses and multi-femtosecond laser pulses in the transparent region were directly evaluated by cross-sectional TEM analysis [165]. In order to improve the actual efficiency of solar cells, many researchers proposed different methods. In 2008, M. Halbwax et al. prepared a laser-refined Si structure which reduced the reflection of the silicon surface. The 1 mm^2^ region was constructed on n-type silicon with a femtosecond laser under vacuum. The laser induced sample surface structure was generated by scanning a simple straight line (30 μm width) at a rate of 150 μm s^−1^. The results exhibited that the photocurrent in the laser textured area increases by about 30% [166]. In 2017, A. Wang et al. studied the material redistribution controlled by spatially modulated femtosecond laser pulses on the silicon surface. Through employed a spatial light modulator the intensity distribution was shaped. Firstly, the material was selectively melted and then redistributed by the laser-induced plasma [167]. The flux used was much higher than the ablation threshold. After a spatially modulated femtosecond laser pulse, a series of phenomena were triggered, including electron ionization, plasma formation, shock wave propagation, and electron-phonon coupling. Although the thermal effect of ultrafast lasers was minimal, material heating effects were observed in the high throughput range Plasma and shock waves can be simultaneously formed in different regions to redistribute the material into different patterns by using spatially modulated laser pulses. Adjusting the intensity distribution can fabricate complex structures. 

In 2008, S. Liu et al. reported recent research on the plasma induced by femtosecond laser pulses in the formation of surface microstructured silicon. The plasma generated in sulfur hexafluoride (SF_6_) presented the strongest signal among the four ambient gases (SF_6_, N_2_, air, and vacuum). The results showed the emission intensity initially increased with gas pressure and reached a maximum at a pressure of about 70 kPa and then decreased as the pressure further increased [168]. The physical and chemical properties of certain semiconductor and biological materials can be improved by adjusting surface nanopatterns. In 2009, X. Jia et al. reported a three-beam interference experiment with 800 nm femtosecond laser pulses and complex micro/nano structures on 6H-SiC crystals. 2D long-period microdots and short-period nanograins and nanoparticles were micro/nano structures and the period of the nano-ripple and the diameter of the nanoparticles were about 150–200 nm [169]. The direct-fs laser irradiation was used to prepare large-area uniform micro/nano structures. The morphological characteristics of semiconductor (GaAs and Si) and metal (brass) induced structures under different laser energy densities were investigated. In general, as the laser energy density increased, its characteristic scale also increased. The micro/nano structure enlarged, the groove deepened, and the roughness increased [170]. In 2013, S. Kaneko et al. reported that periodic nanostructures self-organized on the target surface after the CW laser scanned the target at a high speed of 300 m/min. The nanostructures exhibited a well-regulated structure with a period equal to the wavelength used for annealing. Depending on the laser power, the period varied from the main laser (532 nm) used in the annealing system to the auxiliary laser (797 nm). The nanostructures exhibited a well-regulated grating with precise periodicity of structural color and modified water repellency on the target surface [171]. As seen in Figure 19a, two continuous wave laser systems irradiated on a silicon substrate and a long-period nanoribbon grating line was formed. The SEM image describes laser scanning trajectories at a distance of 40 mm, and the inset depicts detailed images of well-organized nanostructures on the laser scanning trajectories. The period of the nanostructures varies from 12 to 18 W with a single laser power (Figure 19b).

In 2010, W. C Shen et al. applied radial and azimuthal polarization femtosecond lasers to induce new micro/nano structures on silicon surfaces. The results exhibited that the nanostructures were generated in the low laser energy density region with a period of about 600–700 nm. The micro-periodic structures were produced in the high laser energy density region with a period of about 2~3 μm [172]. In 2011, S. Kaneko et al. observed nano-bar grating lines formed by high-speed scanning CW laser method, and Raman spectroscopy showed that the laser irradiation region recrystallized due to exceeding the melting temperature. Periodic nano-striped grating lines (nano-SGL) were self-organized along a single scan trajectory of a continuous wave (CW) laser. The nano-grooves were produced with a period of 500 to 800 nm which was determined by the laser power. This simple method of combining a scanning laser with a high scanning speed of 300 m min^−1^ was expected to generate large-area nanostructure and high output [173,174]. In 2013, S. Kaneko et al. reported that periodic nanostructures self-organized on the target surface after the CW laser scanned the target at a high speed of 300 m/min. The nanostructures exhibited a well-regulated structure with a period equal to the wavelength used for annealing. Depending on the laser power, the period varied from the main laser (532 nm) used in the annealing system to the auxiliary laser (797 nm). The nanostructures exhibited a well-regulated grating with precise periodicity of structural color and modified water repellency on the target surface [171].

Sb_2_Te_3_ was one of the most widely used materials for thermoelectric applications. At present, it was significant to investigate the thermoelectric properties of low-dimensional structures such as thin films, wires, and quantum dots. In 2011, Y. Li et al. reported femtosecond laser-induced nanotracks in highly absorbing Sb_2_Te_3_. The nano-orbital group was observed to have a width of 50 nm and a period of 130 nm, and its coverage area increased as the laser energy density increased [175]. In 2011, M. Tang et al. reported a maskless multibeam laser lithography for large area nanostructure/microstructure fabrication. The pattern produced was a highly ordered periodic structure. AFM cross-sectional analysis of the etched profile exhibited a line width of 1 μm and a microstructure height of 1.7 μm. This lithography technology can produce arbitrary patterns at high speed and flexibility. The distribution of nanostructures/microstructures was determined by the distribution of the laser microlens array (MLA) geometry [176]. Over the past few decades, surface microstructures on silicon have attracted widespread attention because of the potential applications of well-defined micropatterned silicon in optoelectronic devices, sensor technology, solar cells, etc. In 2011, X. Wang et al. reported the formation of large-sized microstructures on a silicon surface having a size of 1 × 1 mm^2^ by femtosecond laser progressive scanning. Scanning electron microscopy studies have shown that uniform surface microstructures have been prepared, such as aligned rod-like mesoporous structures [177]. In 2011, D. A. Zuev et al. conducted experiments to produce uniformly textured mc-Si samples suitable for solar cell fabrication. The morphology and reflectivity of the polysilicon surface under laser action were studied. The change in the reflectance spectrum of the laser-deformed mc-Si sample after acid and alkali chemical etching was proved [178]. In 2012, K. Lou et al. produced complex two-dimensional subwavelength microstructures generated by a femtosecond vector light field on silicon. The fabricated microstructure had an interval between the two corrugations in the microstructure of about 670–690 nm and when the pulse fluence was 0.26 J cm^−2^ above the ablation threshold of 0.2 J cm^−2^. The depth of the trench was about 300 nm [179].

Intense laser femtosecond (fs) laser pulses have been shown to prepare periodic nanostructures on solid surfaces, where the observed structure size was 1/10–1/5 of the laser wavelength. In 2012, G. Miyaji et al. reported the mechanism of periodic nanostructure formation on crystalline silicon (Si) surfaces irradiated in water under conditions of interest was studied. 800 nm, 100 fs, 10 Hz laser pulses generated two periodic nanostructures with periods of 150 and 400 nm. Excitation of surface plasmon polariton (SPP) in the surface layer can be the basic mechanism for forming periodic nanostructures on the Si surface illuminated in water [180]. Ablation of solid surfaces illuminated with superimposed multiple low energy femtosecond (fs) laser pulses typically led to the formation of periodic nanostructures on the target surface [181]. In 2013, K. Miyazaki et al. demonstrated that the self-organization process of nanostructures can be adjusted to produce uniform nanoparticles on target surfaces in air. Self-organized periodic surface nanostructures had a non-uniform structural period in the range of 150–280 nm [182]. In 2014, X. Dong et al. reported the use of laser doping techniques to dope nitrogen in silicon. As the flux increased, the cone peak height ranged from 2 to 20 μm. By laser-assisted chemical etching and laser ablation processes, micro-structured forests can be created on the silicon surface, which facilitated light absorption [183].

In 2014, J. Yang et al. studied direct laser writing for creating anti-reflective microstructures on Si surfaces. Si nanowire arrays were fabricated using laser interference lithography and metal assisted chemical etching. The high-transverse wavelength structure improved the anti-reflection performance. Moreover, surface plasmon resonance can be used to further control broadband reflection by decorating Si nanowires with metal nanoparticles [184]. Black Si was prepared using a fiber laser ablation system (Figure 20a). The laser wavelength was 1064 nm, the pulse duration (full width at half maximum) was 1 ns, and the laser spot size was 20 mm. This method was capable of producing a microstructure of a large area of 10 × 10 cm^2^ on a Si substrate in only a few minutes, as shown in Figure 20b,c. Laser ablation produced a surface structure with a higher aspect ratio than KOH wet etching, which was a key factor for better light capture. Figure 20d showed the surface profile of the laser textured Si surface. The surface structure had a height of more than 15 microns, a width of 10 microns, and an aspect ratio of 3:2.

Recently, to alter the liquid wettability of solid surfaces, femtosecond laser micromachining has been used in interface science. In 2014, J. Yong et al. reported the fabrication of silicon surfaces with layered micro/nano structures by femtosecond lasers. Similar to fish scales, the laser-induced surface exhibited super-hydrophilicity in the air and super-oleophobicity under water. The oil contact angles can reach up to 159.4 ± 1°and 150.3 ± 2°, respectively, for 1,2 dichloroethane and chloroform droplets in water [185]. Although the laser-induced periodic surface structure was observed early, its formation mechanism remained the focus of intense debate. In 2015, M. Fule et al. studied the surface structure of femtosecond laser pulses on Si, GaAs, and V surfaces under the same conditions. The surface of all materials exhibited a trace of melting at low pulse counts and hundreds of nanometers of ripple were observed in all cases [186]. Periodic nanostructures within the semiconductor can be achieved by an infrared ultrashort pulse laser with a dual pulse configuration. In 2015, M. Mori et al. reported that the periodic nanostructures inside the semiconductor caused by infrared ultrashort pulse lasers were generated by a double pulse configuration without surface damage [187]. In 2015, M. Sobhani et al. studied irradiated sub-micro/nano structure of polished silicon using double nanosecond laser pulses (λ = 532 nm) in distilled water. After irradiation with single nanosecond and double nanosecond laser pulses, Submicron/nano clusters were created on the silicon surface. The results exhibited that the surface cluster density and optical reflectivity were affected by the number of laser pulses and the time interval between the double pulses [188].

### 4.2. 2D and 3D Micro/Nano Structures by Laser-Assisted Processing

In 1997, M. Heschel et al. put forward microfabrication and characterization of true three-dimensional (3-D) diffuser/nozzle structures in silicon. Chemical vapor deposition (CVD), reactive ion etching (RIE), and laser-assisted etching were applied to etch the flow cell and diffuser/nozzle elements. These rooms had an area of 2 × 2 mm^2^ and a depth of 200 μm. The side of the through hole was 400 μm [189]. In recent years, gallium nitride has been synthesized for potential applications in blue and ultraviolet light emission and high temperature/high power electronic devices. In 2001, W. S. Shi et al. reported an oxide assisted method for synthesizing GaN nanowires by laser ablation of GaN targets mixed with Ga_2_O_3_. Hexagonal and cubic GaN nanowires were observed by transmission electron microscopy. It also showed that the GaN nanowires were smooth and straight, and the core-sheath structure had an average diameter of 80 nm and a length of several tens of micrometers [190]. Laser beams, electron beams, and focused ion beams have attracted much research interest in the fabrication of functional nanostructures. Near-field laser illumination is one of the effective methods to break the diffraction limit of light and promote the feature size of the pattern down to several Tens of nanometers [190]. In 2006, M. H. Hong et al. described nanopore array patterning by parallel particle masks. The transparent nanoparticles self-assembled on the surface of the phase change Ge_1_Sb_2_Te_4_ (GST) film. After the pulsed laser irradiation, nano-hole arrays were uniformly prepared on the surface. The bowl-shaped nanopore prepared by the enhancement of the light intensity in the vicinity of the contact area between the transparent particles and the substrate had a uniform diameter of about 120 nm [191]. In 2006, Y. Lin et al. developed a phase-change nanolithography technology that combined femtosecond lasers, microlens arrays and wet etching processes to produce millions of 2D and 3D nanostructures (50 nm) in large areas at high speed. Near-field scanning optical microscopy, power microscopy, and atomic force microscopy were utilized to represent the optical and electrical properties of crystalline and amorphous states, respectively [192]. In 2006, L. Pramatarova reported that microscopic structures with controlled sizes and shapes were produced by laser illumination of various surfaces, followed by growth of nanostructured hydroxyapatite. The latter was considered to be the main inorganic component of bones and teeth. Therefore, the formation of this material can be applied to medicine, dentistry, and tissue engineering, especially as the population ages [193].

In 2007, N. N. Nedyalkov et al. described the study of the formation of nanopores on silicon surfaces mediated by near-electromagnetic fields in the vicinity of gold particles. Variations in laser energy density and particle size provided the possibility of fabricating structures having a lateral dimension in the range of 200 nm to less than 40 nm. Gold nanospheres with diameters of 40, 80, and 200 nm were utilized. The sample was irradiated with a laser pulse at a flow rate below the ablation threshold of the native Si surface which led to nano-sized surface modification [194]. Some possible application arrays for silicon tips covered laser marking of silicon wafers, emitters for field emission devices, scanning probe microscopes, and solar cell devices. In 2008, J. Eizenkop et al. put forward experimental data for computer modeling and the formation of a sharp tapered tip on a silicon-based three-layer structure consisting of a single-crystal Si layer on a 1 μm silicon dioxide layer on a bulk silicon substrate. In the case of a single pulse irradiation of a silicon film, different thicknesses of 0.8 to 4.1 μm were generated on the insulating substrate [195]. In 2008, Y. Jun et al. reported that a silicon inverted wood stack structure can be created by combining methylsilsesquioxane (MSQ) photoresist and direct laser writing (DLW). In a general way, MSQ and DLW can be applied to create templates that can withstand high temperatures without cracks or significant distortion. This should allow for a wider range of material penetration to generate photonic crystals and other useful microstructures [196]. In fact, various 3D photonic crystals have been obtained through DLW. In particular, the woodpile structure has been demonstrated (Figure 21a), whose theory predicts large photon band gaps. DLW was also applied to combine intentional structural defects, which were necessary for several applications of photonic crystals. As can be seen from Figure 21b, MSQ photoresist was used to generate the wooden pile structure. The transverse size of the photonic crystal was about 116 × 116 μm^2^. 17.3 mW. To obtain an inverted Si stack structure, low-pressure chemical vapor deposition (LPCVD) was employed to permeate preheated samples with Si. Reactive ion etching (RIE) was then performed to remove excess Si from the top of the structure. Figure 21c showed a cross-section image of the Si inverted stack structure obtained from samples.

In 2008, S. Lee et al. reported micro-grinding of silicon surfaces with femtosecond lasers was studied. The results showed that the optimum laser energy density for Si micronization was found to be in the range of 2–8 J cm^−2^, and the grinding efficiency reached a maximum between 10 and 20 J cm^−2^. The laser induced plasma formed during the microstructured process can offer information about the reaction process during laser irradiation [197]. In 2008, M. L. Taheri et al. reported the demonstration of dynamic transmission electron microscope (DTEM) as a tool that can image Nanowire (NW) during pulsed-laser ablation (PLA) synthesis. Therefore, in the future it was possible to determine the local microstructure and morphology before, during and after laser-assisted NW formation, allowing to gain insight into the processing parameters that affected growth [198].

In 2009, B. W. Liu et al. developed a high power wavelength tunable femtosecond fiber laser source based on photonic crystal fiber technology. Developed light sources have exhibited many promises for rapid silicon micromachining and chrome nanofilm patterning at high pulse repetition rates. A laser pulse with an average power of 10.4 W and a pulse width of 52 fs was provided [199]. Microspheres/nanostructured hollow spheres had broad applications for example in drug delivery, biotechnology, photocatalysts, photonic devices, and high energy electrode materials due to their special structure. The Si hollow sphere monolayer will be an excellent anode material in rechargeable lithium ion batteries. In 2009, S. Yang et al. proposed a simple and flexible approach to fabricate hollow sphere arrays (HSAs) with graded micro/nano structures based on polystyrene colloidal monolayer templates. The thickness of the shell appeared relatively uniform around the entire sphere. The shell was porous and was composed of Si-nanoparticles, which may have an average size of about 30 nm [200]. Cd*_x_*Zn*_x_*Te semiconductor solid solution was a promising material for the production of optoelectronic devices and ionizing radiation registration detectors. In 2010, A. Medvid et al. reported under Nd:YAG laser irradiation at an intensity range of 4–12 MW cm^−2^, self-organized size structure was observed on the surface of Cd_0.9_Zn_0.1_Te crystals [201]. Wide gap III–V nitride semiconductors such as GaN were expected to be applied in blue or ultraviolet (UV) emitting devices. In 2011, A. Ramizy reported porous silicon (PS) was fabricated by a laser induced etching (LIE) process [202]. When the laser power density was increased to 25 W cm^−2^, micropiles were formed on the surface. As the laser power density increased to 51 W cm^−2^, pores were created in structures of different sizes, and structural pores were limited to smaller sizes [203,204].

In 2017, X. Li et al. reported the fabrication of silicon micro/nano structures of controlled size and shape by chemical translation assisted femtosecond laser single pulse irradiation. Varying the pulse energy density to fabricate various laser printed patterns for etching masks resulted in sequential evolution of three different surface micro/nano structures, namely, ring microstructures, flat top pillar microstructures, and spike nanostructures. This method had broad application prospects in the field of photons, solar cells, and sensors [205]. Figure 22a–c illustrated the processing steps of a controllable micro/nano structure. Firstly, a focused single-pulse fs laser was applied to illuminate the sample Si wafer, which had an energy density close to the damage threshold. Samples were pretreated with HF solution, acetone, ethanol and deionized water to eliminate surface oxides and organic contamination. The treated wafer was then etched in an etchant (KOH solution) to form a surface micro/nano structure. The laser source was the Ti:sapphire Regenerative Oscillator-Spectra-Physics, which provided a basic Gaussian mode with a pulse width of 35 fs (FWHM), a center wavelength of 800 nm, and a repetition rate of 100 or 200 Hz. The number of pulses delivered to the sample was selected by a fast mechanical shutter synchronized with the laser repetition rate. Figure 22d–f showed the morphology after irradiation with a single pulse fs laser at different flow rates. As shown in Figure 22d, at a high pulse energy density, a modified region of a hollow shape is formed to obtain the highest energy intensity in the Gaussian laser beam. As the laser pulse energy density decreased, the modified region decreased accordingly, and the central hollow shape region gradually become smaller until it disappeared (Figure 22e,f). Figure 22g–i exhibited three typical Si structures, namely ring-shaped microstructures, flat-top column microstructures, and spiked nanostructures.

In 2014, M. Schule et al. studied incubation and nanostructure formation on various dopant types and concentrations of Si (100) and Si (111) surfaces. The sample was exposed to a closely focused, high repetition rate Ti:sapphire laser in water with a center wavelength of 800 nm and a pulse length of 12 fs to 1.6 ps. 130 nm periodic corrugations and sponge-like random nanoporous surface structures were formed by water immersion. Three types of nanostructures appeared on the surface of all silicon crystals: nano-lift, periodic nano-patterns perpendicular to laser polarization, and spongy random nano-porous surface patterns [206]. Laser interference lithography was the prior technique for achieving periodic and periodic structures like nanogrooves, nanopore arrays, and nanodots arrays in a large area at a fast and low cost, and has broad application in biosensors, solar cells, and photons. In 2014, H. F. Yang et al. introduced the simulation of single-photon and double-exposure dual-beam interference lithography. Conventional nano-trench and grid were prepared on the photoresist and then transferred onto a silicon substrate. The experimental results were in good agreement with the simulation results and the spacing width of the nano-grooves was 1.1, 1.7, and 2.4 μm, respectively [207].

Potential applications for porous silicon (ps) have expanded into advanced areas such as biological and chemical sensors, photocatalysis, energy, supercapacitors, and bioimaging. In 2016, X. Lv et al. demonstrated the preparation of porous silicon with dual-band photoluminescence (PL) by chemically assisting 1064 nm picosecond (ps) laser irradiation of polycrystalline silicon. The results showed that the maximum surface porosity of the sample was 90.48%, and that brought good hydrophobicity to the prepared sample [208]. Excitation of (surface plasmon polaritons) SPPs have been shown to be the primary process for forming nanoparticles using fs laser pulses. In 2016, G. Miyaji et al. reported the use of ultraviolet (UV) fs laser pulses of 266 nm in air to ablate GaN to form uniform nanoparticles of 50 nm [209]. Biocompatible sensing materials played an important role in biomedical applications. As a processing techniques, increasing the biocompatibility of these sensing devices typically leads to a decrease in total conductivity. Due to its semiconductor properties and availability, silicon is becoming a more viable and available option for these applications. In 2017, S. Hamza et al. presented a method for generating nanofibers using a nanosecond pulsed laser and customizing the electrical properties of laser-processed silicon to improve its sensing applications requiring a biocompatible environment using gold sputtering techniques [210]. In 2017, X. Liu et al. proposed a dry etching assisted femtosecond laser machining (DE-FsLM) method. Since the power required for the laser-modified material was lower than the power required for laser ablation, the manufacturing efficiency can be improved to form a complicated 2.5D structure [211]. Figure 23 illustrated the use of a femtosecond laser to modify and then perform a dry etch process to fabricate the microstructure. It was well known that Silicon was one of the most important materials in the IC industry. Therefore, it was a vital factor in evaluating the performance of new manufacturing technologies for its ability to fabricate silicon functional architectures. Silicon was chosen to evaluate the capabilities of DE-FsLM. For example, a silicon concave microlens was prepared by this method as shown in Figure 23a. Figure 23b showed the trend of the diameter and depth of the concave structure as a function of etching time.

In 2017, M. Aouassa et al. proposed a direct laser nanostructure remodeling and simultaneous temperature control method. It was showed the application of Raman thermometry to in-situ precise on-line control of Si column shape evolution during direct laser nanostructure or reformation [212]. MoS_2_ nanostructures like nanobelts, nanowebs, etc. can open up different applications in nanoelectronics and optoelectronic devices and sensors. In 2017, R. Rani et al. demonstrated unique nanostructures, such as nanoribbons, nanowebs, on MoS_2_ flakes using a direct one-step laser irradiation process. The minimum feature size was 300 nm, which was close to the diffraction limit of the laser used [213]. Due to the rapid development of laser forming technology, the intensity distribution of femtosecond laser pulses can be flexibly modulated, providing a new opportunity to control laser-material interactions. In 2018, W. Han et al. reported polarization-dependent concentric circular periodic surface structures were fabricated on Si by using a single shot femtosecond (fs) laser pulse based on pre-processed quasi-plasmonic annular-shaped nanostructure [214]. Unlike graphene nanostructures, the various physical properties of nanostructured MoS_2_ have not been developed due to the lack of established manufacturing routes. In 2018, R. Rani et al. reported the use of focused laser etching to create a MoS_2_ nanostructure like diamond and star-shaped of the desired shape and size [215]. In 2018, A. Verma et al. reported by using direct ns-laser writing, two dimensional periodic arrays of microsquares of different dimensions were prepared on silicon substrate. The method was used to fabricate low optical reflective silicon substrates which is based on laser patterns for SERS applications and optical reflection from the patterned surface was reduced to less than 11% over a wide wavelength range of 300 nm to 1200 nm [216].

In 2016, B. Quan et al. exhibited the fabrication of a three-dimensional (3D) container consisting of inverted pyramidal pits (IPP) and nano-sized openings fabricated by a combination of laser interference lithography and anisotropic wet etching processes. IPP structures are expected to have many interesting potential applications, such as surface enhanced Raman scattering substrates and micro/nano containers for biomedical research [217]. In order to prepare an IPP array with a nano opening capwas, LIL and anisotropic wet etching were combined. The manufacturing process was schematically illustrated in Figure 24. For anisotropic wet etching of 3D nanostructures (Figure 24a), single-sided polished n-type (100)-oriented silicon substrates having a resistivity of 10–20 Ωcm were used. To make large-area samples using IPP arrays, LIL was chosen to be a pattern definition technique. The silicon wafer having a 60 nm SiN*_x_* film was cleaned and coated with a 120 nm AR-P 5350 photoresist. The photoresist was re-exposed using the same sample rotated 90° to produce an array of photoresist dots, as shown in Figure 24b. The sample was then coated with a 20 nm Ni film by electron beam evaporation (Figure 24c). Ni was chosen as the dry etch mask. Following the lifting process, a Ni mesh was generated with a nanopore array (Figure 24d). The unprotected silicon nitride region was etched using a RIE reactor. The silicon nitride pattern having the nano-sized hole array formed in this step was used as a mask for anisotropic etching of the underlying silicon in KOH solution (Figure 24e). Anisotropic wet etching was carried out in 25 wt% KOH solution (Figure 24f). It was advantageous to use 25 wt% of tetramethyl ammonium hydroxide (TMAH) as an alternative etchant for silicon anisotropic etching. Finally, a 3D submicron container array consisting of IPPs with nano-openings was obtained by controlling the wet etching time, as shown in Figure 24g. Figure 24h showed a top view SEM image of an IPP array with nano-openings. Furthermore, the flat surface of the cap with nano-opening was characterized by AFM (Figure 24i). From the cross-sectional view of the inverted pyramid pit (Figure 24j,k), the period of the pit array was about 400 nm, the diameter of the nano opening was about 160 nm, the height of the cavity was about 190 nm, and the tilt angle was about 54°. As above, materials, feature size and the laser parameters of laser-assisted fabrication of 2D / 3D micro-nano structures are summarized in Table 5.

## 5. Fabricating Micro/Nano Structures on Other Materials by Laser Assisted Processing

### 5.1. Glass

In 2003, Y. Cheng et al. proposed the fabrication and integration of 3D hollow microstructures into glass for use in lab-on-a-chip applications. The cross-sectional profile of the fabricated microchannel was elliptical with a width of 17 microns and a depth of 71 microns (aspect ratio, 4.2). The use of high order multiphoton processes caused by fs lasers enabled the fabrication of complex 3D microstructures that were buried in the glass [218]. In 2005, M. Ams et al. reported theoretical and experimental results for slit beam shaping configurations for the fabrication of photonic waveguides using femtosecond laser pulses. Optical waveguide was fabricated by inserting a slit having a width of 500 μm, and the core diameter of the waveguide was less than 15 μm. It was showed that this method supported focusing the objective with a longer depth of field and allowed direct writing of microstructures with a circular cross section while using a vertical writing scheme [219]. In 2005, Y. Cheng et al. reported the theoretical and experimental study on the fabrication of three-dimensional microchannels of photostructuring glass using femtosecond (fs) lasers. A new method of placing slits prior to control the aspect ratio in 3D microchannel fabrication before focusing the system has been demonstrated. Thanks to the multiphoton process and focus forming technology, both horizontal and vertical resolutions were up to 10 microns and microreactors comprising microchannels, microcells and microvalves were made by this technique [220].

In 2005, K. Sugioka et al. reported the realization of three-dimensional (3D) micromachining of photosensitive laboratories by photochemical reactions using femtosecond (fs) lasers. The height and width of the channel were 17 and 4.2 μm, respectively. The modification mechanism of fs laser to photosensitive glass and the advantages of the process were discussed. By using this technology, various micro-components for chip lab devices such as microfluidics, microvalves, micro-optics, microlasers, etc. were manufactured [221]. In 2006, P. Gupta et al. reported a simple writing method and the mechanism of laser induced formation of polycrystalline lines, dots, and micron size features on Nd_0.2_La_0.8_BGeO_5_ (NLBGO) glass samples was discussed. The surface of the polycrystalline wire produced by the continuous wave of titanium sapphire crystallized, and the grain size gradually decreased from the center to the edge [222]. Chalcogenide glasses were known for their photo-stability, which made them widely used in optical disc storage media such as compact disc ROM (CDR) and digital versatile disc (DVD).

In 2006, S. Wong et al. presented for the first time direct laser written 3D photonic crystals (PCs) with a photonic bandgap (PBG) made from As_2_S_3_ chalcogenide glasses [223]. Figure 25a showed a schematic diagram of the direct laser writing of a three-dimensional photonic crystal with a complete photonic band gap in a chalcogenide glass. A four-layer stake with a rod spacing of 2 μm was illustrated in Figure 25b. The illustration showed four pairs of single pass bars, resulting in an overall almost rectangular cross section of the rod. Figure 25d showed a top view of the structure. The focused ion beam cross section exhibited the size of the rod (Figure 25c). The close-up on the surface exhibited the overall smoothness of the final material (Figure 25e).

Femtosecond laser nanofabrication is an attractive method in glass engineering, with the ability to eliminate lateral damage in the dielectric, reduce the heat affected zone, and create sub-diffracting to limit the target area. In 2006, Y. Zhou et al. reported nanoscale pores can be formed on the surface of the glass by depositing a single layer of silica microspheres as a particle mask on the surface of the substrate by using femtosecond laser irradiation. At low flux, feature sizes of 200 to 300 nm were found with an average depth of 150 nm [224]. At high throughput, a three-hole structure was formed when the laser flux was greater than 43.8 J cm^−2^. In 2007, Z. Wang et al. demonstrated the fabrication of a low-gloss cylindrical lens and a flat lens embedded in a bulk glass by using fs laser direct writing followed by heat treatment, wet chemical etching and additional annealing to form 3D hollow microstructures. One type of lens was cylindrical microlens having a radius of curvature *R* of 1.0 mm, and the other was plano-convex microlens having a radius *R* of 0.75 mm. In addition, 3D integration of microlenses with other micro-optical elements, such as micromirrors and optical waveguides, in a single glass chip was achieved [225]. The nano-aquarium was a microchip for dynamically observing living cells and microorganisms, directly written into a photostructural glass by an fs laser, and then fabricated by annealing and continuous wet etching. In 2008, Y. Hanada et al. demonstrated the technology of 3-D micromachining for the manufacture of nanoaquaria for the dynamic observation of living cells and microorganisms in fresh water. The microstructure had a 1 mm long channel with a cross-sectional dimension of 150 μm × 150 μm × 150 μm below the glass surface [226]. In 2008, Y. Zhou et al. demonstrated a direct femtosecond laser nanopatterned glass substrate enhanced by particle-assisted near-field enhancement. Nanoscale pores can be created by depositing a single layer of microspheres on the surface of the substrate as a particle mask and using a femtosecond laser as the light source and the aperture was measured from 200 to 300 nm with an average depth of 150 nm and increased with increasing laser flux [227].

Microlenses and microlens arrays had broad applications such as laser alignment, optical computing and photon imaging. In 2009, H. Liu et al. reported the fabrication of spherical microlenses on optical glass by femtosecond laser direct writing (FLDW) in ambient air. Positive spherical microlens having a diameter of 48 μm and a height of 13.2 μm was fabricated on the surface of the glass substrate. Compared with traditional laser direct writing (LDW) technology, this work can provide an effective method for precise shape control to manufacture three-dimensional (3D) microstructures with curved surfaces for difficult-to-cut materials for Practical application [228]. In 2010, M. K. Bhuyan et al. introduced a systematic study of femtosecond laser microchannel processing in glass using non-diffractive Bessel beam. With proper light source parameters, channels with a diameter of 2 μm and an aspect ratio of up to 40 can be machined. An important goal here was to demonstrate the basic novelty and advantages of Bessel beam processing with the aim of stimulating further research in this area [229]. In 2010, Y. Liao et al. demonstrated that three-dimensional microfluidic channels of any length and configuration inside the glass was fabricated by direct writing laser. A rectangular wavy passage of about 250 μm was prepared under the glass surface, with a total length of about 1.4 cm and a diameter of about 64 μm [230]. In 2012, J. Choi et al. proposed a new method based on femtosecond laser induced depletion to produce three-dimensional (3D) optical second-order nonlinear microstructures in silver-containing photosensitive glass with a χ^2^ (second-order susceptibility) value of about 2.5 times that of quartz. This laser writing approach enabled non-linear micron-scale structuring in a precision-designed 3D configuration, including versatile lab-on-a-chip equipment, active 3D optical storage devices, and high-damage threshold optical converters [231]. In 2015, J. Rueda et al. studied the use of tightly focused, time-shaped femtosecond (fs) laser pulses to create nanostructures (100–200 nm) in two dielectric materials (sapphire and phosphate glass) whose response to pulsed laser radiation has different characteristics [232]. Understanding the phase transitions in glass and the morphology of the relevant nanostructures after femtosecond laser irradiation is vital for the fabrication of functional optics referring to glass crystallization to achieve nonlinear optical properties. In 2017, J. Cao et al. reported the crystallization of lithium silicate silicate glass induced by fs laser and the nanostructures had applications in the fabrication of second order nonlinear optics [233].

### 5.2. Oxide

#### 5.2.1. SiO_2_

In 2001, A. Marcinkevicius et al. demonstrated direct three-dimensional (3-D) micromachining in a volume of quartz glass. This technology permitted the fabrication of three-dimensional channels as small as 10 μm in diameter, with any interconnect angle and high aspect ratio [234]. In 2004, X. Ding et al. reported the use of the laser-induced backside wet etching (LIBWE) method and a mask projection system to fabricate various micropatterns on quartz glass. For fine tuning of the projection system, a grating as narrow as 0.75 μm was etched on the glass surface [235]. In 2006, B. Le Drogoff et al. exploited a laser-based nanofeatured rapid printing process called laser-assisted nano transfer printing (LA-nTP). By using this technique, chromium nanodots array (80 × 80 nm^2^, 200 nm period, and 35 nm thick) were transferred onto a prefabricated hot SiO_2_ surface and applied it as an etch mask to complete the fabrication of 3D nanostructures [236]. In 2007, M. Dubov et al. reported the use of focused laser pulses to record the periodic structure of 150 nm spacing for the first time in a permanently moving pure fused silica sample [237].

In 2009, M. Kim et al. reported the fabrication of microchannels by scanning femtosecond laser pulses and subsequent selective wet etching processes. The integrated three-dimensional microchannel with 5 μm neck diameter and optical waveguide structure were fabricated inside fused silica. Single red blood cells in human blood diluted inside the fabricated microchannels were detected by two optical protocols [238]. Femtosecond laser micromachining was a promising technology for developing multi-functional integrated biophotonic devices that can be coupled to a microscope platform for complete characterization of the cells being tested. In 2010, F. Bragheri et al. demonstrated the design and manufacture of a single chip for optical capture and stretching of individual cells. A chip produced in a fused silica glass substrate by femtosecond laser micromachining covered a microchannel in which cell suspension flowed, and a waveguide that transmits a laser beam forming a double beam optical trap [239]. Femtosecond laser micromachining was an attractive solution for three-dimensional (3D) micromachining in transparent materials. In 2010, F. He et al. reported the preparation of a hollow microfluidic channel which had a circular cross-sectional shape embedded in fused silica by a spatially focused femtosecond laser beam [240].

In 2012, J. Lin et al. reported the preparation of a three-dimensional (3D) high-Q vocal wall micro-cavity on a fused silica chip by femtosecond laser microfabrication, which was realized by the three-dimensional characteristics of femtosecond laser direct writing [241]. The process flow for fabricating a high Q micro-cavity consisted of two main steps: (1) femtosecond laser exposure followed by selective wet etching of the irradiated region to produce a micro-disk structure; (2) selective reflow by CO_2_ laser annealing Silica chamber to improve quality factors.

In 2012, P. S. Salter et al. presented an improved method of fabricating optical waveguides in bulk materials by femtosecond laser writing. The waveguide was demonstrated in fused silica with a coupling loss of 0.2 to 0.5 dB and a propagation loss was smaller than 0.4 dB/cm. An LC spatial light modulator (SLM) was applied to shape the beam focus by creating adaptive slit illumination in the pupil of the objective [242]. Materials with curved surface microstructures are suited for micro-optical and biomedical devices. Nevertheless, the implementation of such a device is still technically challenging. In 2013, H. Bian et al. presented an easy and flexible way to prepare curved microstructures with controllable shapes and sizes. The method consisted of a femtosecond laser exposure and a chemical etching process of a hydrofluoric acid solution. Schematic diagrams of the fabrication process and the laser exposure methods are shown in Figure 26a. Their PDMS reverse replicas were showed in Figure 26b,c. A 4 mW laser pulse was applied with a drilling depth of approximately 150 μm. After 90 min of chemical treatment, a micro-cone was produced [243]. Driveless multi-layer reading systems have proven to be suitable for permanent digital memories using fused silica. In 2013, T. Watanabe et al. reported recording data from a 2 mm thick fused silica plate by a femtosecond laser to form four layers with a point spacing of 2.8 μm and a layer-to-layer distance of 60 μm. The total recording density was 40 megabytes per square inch that was as high as a conventional optical disc [244].

The adhesion of glass with extraordinary mechanical, chemical, and optical properties played a key role. Even with many techniques known, the combination of two glass samples was still a daunting task. In 2013, F. Zimmermann et al. reported that fused silica welding and ultrashort laser pulse bursts which can achieve fracture resistance of up to 96% of bulk materials [245]. Nanofoams were nanostructured porous materials that had many applications in capacitors, batteries, and biological sciences. In 2014, J. Gacob-Jacob et al. proposed the results of a study of femtosecond laser radiation parameters used to make glass nanofoams composed of three-dimensional meshes of interconnected nanowires. Nanofoams with a total volume more than 107 μm^3^ were fabricated, which were composed of glass filaments with a diameter of approximately 70 nm and with an average spacing of approximately 1 μm [246]. In 2014, J. Song et al. reported the preparation of integrated miniature pulse resonators with high-quality factor in microfluidic channels by using femtosecond laser three-dimensional (3D) micromachining and the fabricated miniature ring resonator was only 80 μm in size [247]. Under critical coupling, the Q factor of the microring resonator in air was 3.24 × 10^6^. In 2014, J. Zhang et al. reported the recording of multiplexed digital data which was realized by femtosecond laser nanostructures of fused silica. This storage allowed for unprecedented parameters, including hundreds of terabytes of data per disk, thermal stability of up to 1000 °C, and virtually unlimited lifetime at room temperature [248]. Super-hydrophobic self-cleaning glass surfaces with good transparency and robustness were used in a variety of applications, such as in automotive windshields, windows, anti-fouling mirrors, safety glass, and solar panels. In 2018, Y. Lin B et al. presented a new ultrafast laser machining method for the preparation of sturdy superhydrophobic and transparent silica glass surfaces with excellent properties and chemical surface modification and the diameter of the micro-pits was 20 μm and the interval of the periodic array was 30 μm [249]. Table 6 shows some experimental papers for the preparation of 3D micro/nano structures on silica.

#### 5.2.2. Other Oxides

Three-dimensional periodic microstructures of alumina are vital for producing photonic band gap structures (PBG), which are fabricated by laser rapid prototyping by laser induced gas phase direct write deposition. In 1997, M. C. Wanke et al. showed the construction of 3D PBG materials by laser chemical vapor deposition (LCVD) and the structure consisted of parallel rod layers which formed a square lattice with a face center and a lattice constant of 66 and 133 microns [250]. In 2001, M. Khakani et al. reported that nanostructured SnO_2_ films have been deposited on alumina substrates by PLD. TEM observations showed that the SnO_2_ column (about 10 nm in diameter) constituting the film was composed of almost spherical nanoparticles in turn. A CO gas sensor can be prepared by in-situ doping SnO_2_ with a Pt metal catalyst [251]. Ceramic alumina was an important substrate for hybrid circuits due to its high dielectric strength, excellent thermal stability and high thermal conductivity of 25 W m^−1^ K^−1^. In 2005, W. Perrie et al. studied the micromachining of alumina ceramics in air with a repetition rate of 1 kHz, 180 fs light pulses to optimize processing conditions. By optimizing the process parameters, the residual surface roughness can be reduced to the original surface 0.8 μm. The generated debris consisted of alumina single crystal nanoparticles with a diameter of 20 nm to 1 μm and an average diameter of 300 nm [252]. The manufacturing of the nanopatterns relied on the ambient gas pressure during the PLD, the well-separated nano-islands at low pressure, and the interconnected nanodisks formed under high pressure. In 2006, W. Ma et al. reported that various well-ordered perovskite-type composite oxide nanostructures were produced by pulsed laser deposition (PLD) on conductive single crystal substrates by a latex ball single layer and a double layer mask and the BaTiO_3_ nanocrown with a height of about 9 nm was a ferroelectric, and the SrBi_2_Ta_2_O_9_ nanoring with a thickness of about 5 nm had at least piezoelectric activity [253]. In 2009, X. Gao et al. reported the fabrication of well-ordered CoFe_2_O_4_ (CFO) nanodot arrays by a combination of PLD and ultra-thin self-organizing anodic aluminum oxide (AAO) masks. The dot size ranged from 35 to 300 nm, and the distance between dots was from 60 to 500 nm [254]. The nanodot array was fabricated based on the PLD by an ultra-thin AAO stencil mask, as schematically illustrated in the flow chart of Figure 27a. A representative atomic force microscope (AFM) image of the resulting sample nanodots array was also showed in Figure 27b, which showed a uniform hexagonal arrangement of nanodots with an average diameter of 50 nm and a pitch of 104 nm. Figure 27c showed an array of nanodots after partial removal of the AAO mask to visualize CFO points and masks.

In 2009, J. Gottmann et al. reported the modification of a femtosecond laser followed by chemical etching with an aqueous HF acid solution to produce a sub-micron-wide deep hollow nanoplane in the sapphire volume. Hollow nanoplanes (width 200 nm, length 1 mm) were etched using volume selective laser and nanochannels (100 nm diameter) were obtained in sapphire [255]. In 2009, C. H. Lin et al. reported the preparation of Cr_2_O_3_ single crystals nano condensates in air by pulsed laser ablation, and characterized the shape-dependent local internal stress of anisotropic crystals by analytical electron microscopy. On average, a compressive stress of up to 5 GPa was formed [256]. TiO_2_ was a broadband semiconductor with numerous applications including the use of a photocatalytic air purifier to manufacture electrodes for solar energy conversion. In these applications, performance can be optimized in the presence of nanostructured surfaces. In 2009, M. Sanz et al. proposed nanostructured TiO_2_ deposits were fabricated by vacuuming Si and above at 266, 355, and 532 nm with nanometer PLD and oxygen below 0.05 Pa. The narrowest size distribution and the smallest nanoparticle diameter (about 25 nm) were generated by deposition at 266 nm under 0.05 Pa of oxygen [257]. Dense SnO_2_ nanocondensates having specific shapes, sizes, and phase transitions to relatively stable phases had possible applications for optoelectronic/catalytic applications. In 2009, W. Tseng, et al. introduced the synthesis of dense SnO_2_ nanocondensates with fluorite-related structures using PLA technology [258]. In 2009, Y. Wang et al. reported the fabrication of plasma sprayed nanostructured coatings on titanium alloy (Ti-6Al-4V) substrates using the prepared nanostructured Al_2_O_3_-13 wt% TiO_2_ feedstock. After laser remelting, the microhardness of the coating was much higher than the microhardness of the plasma sprayed coating, which was 2–3 times that of the substrate and the surface roughness was also reduced [259].

In 2010, Y. Zhou et al. reported the use of focused laser beam technology to create an extended region of micropatterned GO and reduce GO multilayers on quartz substrates in a fast and controlled manner and the height profile showed that the thickness of the GO nanosheet was 1.5 nm [260]. Nanostructured networks were of considerable importance due to the high surface to volume ratio. Their unique properties made them suitable for nanodevices and nanomembranes. In 2011, P. Waraich et al. demonstrated a new approach to fabricate 3D nanoparticle networks. Under ambient conditions two samples containing lead oxide (Pb_3_O_4_) and nickel oxide (NiO) particles were ablated and a nanoparticle network having a sub-100 nm nanoparticle size range was generated [261]. In 2012, H. Teoh et al. proposed a versatile technique to create a three-dimensional graphene oxide-reduced GO (rGO) stack layer structure with additional properties that defined micropatterns in each layer. These 3D micropatterning and multilayer structures can provide more potential applications for graphene-based devices [262]. In 2012, D. Kim et al. proved that polarization-dependent nanomaps with a period of about 250 nm were produced on the sapphire surface by scanning the femtosecond laser beam under appropriate illumination conditions [263].

In 2012, C. Samarasekera et al. proposed a new method for synthesizing flower-like nanotips on soda-lime glass by femtosecond laser ablation and the bottom and head of the nanotip have widths of 100 and 20 nm and a length of 10 μm, respectively. Synthetic nanotips can potentially be applied in antimicrobial and hydrogen storage applications [264]. In 2012, C. Wang et al. reported the use of femtosecond laser pulse irradiation to fabricate two types of periodic nanostructures, self-organized nanodots and nanowires on the surface of an indium tin oxide (ITO) film. A plurality of cycles (about 800 and 400 nm) were observed on the ITO film having nano-dots and nanowire structures [265]. In 2013, C. G. Li et al. proposed a detailed microstructure study of laser remelted Al_2_O_3_-TiO_2_ coating. The Al_2_O_3_-TiO_2_ coating on the surface of titanium alloy was prepared by a combination of plasma spraying and laser remelting. The microstructure evolution of laser remelted Al_2_O_3_-TiO_2_ coatings was investigated by X-ray diffraction (XRD), scanning electron microscopy (SEM), energy dispersive spectroscopy (EDS), and transmission electron microscopy (TEM) [266]. Pulsed laser deposition (PLD) was a universal technique for fabricating nanostructures because it controls the size and shape of nanostructure deposits by changing laser parameters. In 2013, M. Sanz et al. reported the results generated at 213, 532 and 1064 nm wavelengths on Si (100) substrates of homemade sintered hematite targets. When the substrate was heated to 750 K, deposition at 1064 nm will result in a thin film of stoichiometric magnetite nanoparticles with sharp edges and sizes ranging from 80 to 150 nm [267]. In 2015, J. Huotari et al. reported vanadium oxide thin films were produced by pulsed laser deposition. The microstructure and crystal symmetry of the deposited films were investigated by X-ray diffraction, scanning electron microscopy (SEM), and Raman spectroscopy. Pulsed laser deposition of vanadium oxide film was investigated as a resistive gas sensor for NH_3_. The film composition was determined to be a pure V_2_O_5_ phase structure or a mixed phase structure of a V_7_O_16_ phase having a V_2_O_5_ phase and a triclinic crystal [268].

In 2015, Z. Zhang et al. reported a high quality TiO_2_ epitaxial layer grown on a LaAlO_3_ substrate by pulsed laser deposition. A prototype of a TiO_2_-based metal-semiconductor-metal ultraviolet (UV) photodetector was prepared using Au as the Schottky contact metal [269]. Compared with bulk materials, CeO_2_ films with nanostructures have potential applications in nanoscale devices. In 2016, B. Wang et al. reported the preparation of a cerium oxide (CeO_2_) film with a series of nano-array superstructures by pulsed laser deposition (PLD). These CeO_2_ nano-array films have high levels of oxygen defects and absorbance by rotating the nanostructures, which may be helpful for their application [270]. In 2017, A. Talbi et al. introduced the surface structure of a titanium oxide film by an ultraviolet femtosecond laser beam. The results exhibited the formation of regular points and laser induced periodic surface structures (LIPSS) with periods close to the period of the beam wavelength. After 13,000 shots, at a very low flux of 15 mJ cm^−2^, a two-dimensional dot with a diameter of 100 nm can be obtained with two different periods (260 and 130 nm) [271]. TiO_2_ has been one of the most studied semiconductors due to its high requirements for various functional photocatalysts in environmental applications, and highlights the application of multiphase photocatalysts. In 2017, R. Zhou et al. reported pulsed laser ablation for the continuous fabrication of Ag/TiO_2_ composite nanoparticles for photocatalytic degradation [272]. Among various metal nanoparticles, copper and copper oxide nanoparticles have become the focus of attention due to their catalytic, optical, and electrical properties. In 2018, B. Eneaze B et al. reported ablation of a copper target immersed in distilled water by a pulsed Nd:YAG laser. The effects of ablation time on the structure and optical properties of the grown copper oxide nanoparticles were investigated. The results of transmission electron microscopy images showed that the synthesized nanoparticles were spherical with an average size of 24–37 nm [273]. In 2018, S. Yu et al. presented a new strategy for producing TiO_2_ or TiO_2_/carbon microstructures using the DLW process. Due to the laser writing method, complex microstructures can be designed as needed. Femtosecond laser irradiation combined with the two-photon absorption path can achieve micron resolution. The minimum line width was 650 nm. Under the experimental conditions required to obtain crystalline TiO_2_ and TiO_2_/C, a typical line width was 3 μm. TiO_2_/C nanocomposites exhibited piezoresistive behavior used in pressure sensor devices. Using this method, DLW can be applied to fabricate a miniature pressure sensor [274].

### 5.3. Carbon Materials

In 1993, F. T. Wallenberger et al. proposed the use of laser-assisted chemical vapor deposition to strong and flexible carbon fibers growing than 0.3 mm per second under high reaction chamber pressure (>1 bar) [275]. In 2004, M. Khakani et al. proposed an “all laser” synthesis method for controlling carbon (single wall carbon nanotubes) SWNTs grown on SiO_2_/Si substrates. Atomic force microscopy and micro-Raman spectroscopy indicated that the “all laser” process resulted in horizontal random networks of SWNT beams that bridged two adjacent nanoparticle strips together. Single-walled carbon nanotubes were found to have a diameter of about 1.1 nm, while bundles had a diameter in the range of 10–15 nm [276]. In 2006, T. Csako et al. proposed the performance improvement of boron carbide film deposited by femtosecond laser pulse. Ultrashort pulse processing was effective and energy densities between 0.25 and 2 J cm^−2^ resulted in an apparent growth rate ranging from 0.017 to 0.085 nm per pulse [276]. In 2006, S. W. Youn et al. reported that focused ion beams, femtosecond lasers, excimer lasers, and cutting techniques were used to prepare glassy carbon (GC) micro-molds. The femtosecond laser was combined with focused ion beam (FIB) milling to reduce the roughness of the machined surface to 45 nm. The microstructure was microstructured at a wavelength of 248 nm using a KrF excimer laser with surface roughness of about 45 and 70 nm, respectively [277]. In 2008, K. Zimmer et al. reported that stimulation of local growth of carbon nanotubes (CNTs) have been realized through excimer laser irradiation of silicon substrates coated with iron (III) nitride and TEM measurements confirmed different sizes of CNTs with diameters in the range of 10–40 nm [278].

In 2012, M. C. D. Bock et al. reported the growth of multi-walled CNT forests on transparent substrates by laser-induced chemical vapor deposition using iron nanoparticle catalysts and the TEM of the cross-sectional samples of these structures confirmed that the CNTs in the entire structure were multi-layered with an average diameter of 5 nm [279]. Although electrochemical capacitors (EC) charged and discharged faster than batteries, they were still limited by low energy density and slow rate. In 2012, M. F. El-Kady et al. reported direct laser reduction of graphite oxide films to graphene using a standard LightScribe DVD drive. The film produced was mechanically strong, with high electrical conductivity (1738 Siemens per meter) and specific surface area (1520 square meters per gram) [280]. In 2012, V. Strong et al. presented a simple, inexpensive solid state method for the formation, patterning and electronic adjustment of graphene-based materials. Laser scribing graphene (LSG) was exhibited to be produced and selectively patterned by direct laser irradiation of the graphite oxide film under ambient conditions. This inexpensive method of generating LSG on a thin flexible substrate provided a means of fabricating a low cost graphene-based NO_2_ gas sensor. LSG also exhibited excellent electrochemical activity and exceeded other carbon-based electrodes in terms of electron charge transfer rate [281].

In 2013, M. F. El-Kady et al. reported the scalable manufacturing of large-area graphene micro-supercapacitors by direct laser writing on a graphite oxide film using a standard LightScribe DVD burner. More than 100 micro-supercapacitors can be prepared on a single disc in 30 min or less [282]. The interdigitated graphene electrode can be photolithographically processed on an optical disc. This form of graphene is called laser scribing graphene (LSG). Due to its nearly insulating properties, GO has a good barrier between positive and negative graphene interdigitated electrodes. Therefore, these graphene circuits can be used as planar micro-supercapacitors after receiving the electrolyte overcoat. This direct “writing” technique did not require masking, expensive materials, post-processing or clean room operations. For example, 112 micro-supercapacitors can be produced on a single piece of GO deposited on a DVD disc. Applications of carbon nanoparticles (CNP) included polymer nanocomposites, functional fillers, supercapacitors, and water purification. In 2014, A. Al-Hamaoy et al. prepared carbon nanostructures of various forms and sizes with different morphological properties by liquid phase pulsed laser ablation (LP-PLA) using high frequency Nd: YAG lasers. According to the TEM results, the sample fraction having nanoparticles with a particle size below 200 nm was composed of nanocrystalline carbon nanoparticles (CNP), while the sample fraction having a particle size greater than 200 nm contained multi-walled carbon nanotubes (MWCNTs) [283]. In 2014, H. Tian et al. demonstrated low cost, no transmission, and flexible laser scribing graphene-resistive random access memory (LSG-ReRAM). The rGO pattern can be grown directly on the PET substrate using laser scribing [284]. (resistive random access memory: ReRAM) The main flow of LSG-ReRAM was shown in Figure 28a. A maskless and programmable laser-scribing technology was utilized to achieve inexpensive laser scribed rGO (LSG) production on flexible substrates. The SEM morphology and structure were shown in false color (Figure 28b). The Ag top electrode was blue. The bottom GO was identified in red according to its different texture. The GO was reduced to rGO by laser pulses after laser scribing.

In 2014, J. Lin et al. exploited a one-step and scalable method for preparing porous graphene from commercial polymer sheets using CO_2_ laser irradiation under ambient conditions and it can be called laser induced graphene (LIG). LIG can be written to a variety of geometric shapes and the physical and chemical properties of the LIG structure made it uniquely suitable for energy storage devices that offered promising electrochemical performance [285]. In 2014, H. Tian et al. reported that wafer-level flexible graphene headphones which can play sounds from 100 Hz to 50 kHz can be generated in 25 min by applying a one-step laser scribing technique [286]. In 2014, H. Tian et al. reported the wafer-level direct preparation of graphene-based transistors, photodetectors and speakers by using a common LightScribe DVD recorder. Laser scribing in-plane graphene transistors have been shown to have large on/off ratios of up to 5.34. Light responsiveness and specific detection ratios of up to 0.32 A/W and 4.996 × 10^10^ cmHz^1/2^W^−1^ were observed in the laser-scored graphene photodetectors. The sound generation of laser scribing graphene was also well proven, with sound frequencies ranging from 1 to 50 kHz. This work showed that laser scribing can be a way to integrate high performance graphene-based devices [287]. In 2014, S. Ushiba et al. reported the fabrication of 3D nanostructures comprising aligned single-walled CNTs (SWCNTs) by two-photon polymerization (TPP) lithography using direct laser writing [288]. To prepare SWCNT/polymer composites, TPP lithography was performed on a SWCNT-dispersed photo-resin while the SWCNTs were immobilized in a tiny polymer structure (Figure 29a). The 3D structure and the characteristic resolution of suspended nanowires exceeded the diffraction limit of light, and SWCNTs are intercalated (Figure 29b–d). Polarization Raman spectroscopy was applied to study the orientation direction and arrangement degree of SWCNTs in the obtained structure.

Due to excellent properties, DLC membranes have been employed in a variety of critical applications such as magnetic recording discs, cutting tools, optical components, and protective coatings for microelectromechanical devices (MEM). In 2015, A. Modabberasl et al. reported the deposition of Diamond-like carbon (DLC) films (the thickness of flim: 165–335 nm) on Si (100) substrates using magnetic-assisted PLD techniques and investigated their microstructure, morphology, mechanical properties and deposition rates [289]. In 2015, Z. Peng et al. reported a method for laser-induced process to fabricate boron-doped graphene structures from polyimide films, being applied as active materials for flexible planar microcapacitors. As a result of boron doping, the highest surface capacitance of the device was 16.5 mF cm^−2^, which was three times that of undoped devices, and the accompanying energy density increased by 5–10 times at various power densities [290]. In 2015, R. Rahimi et al. proposed a technique for manufacturing stretchable (up to 100% strain) and sensitive (up to 20,000 strain factor) strain sensors. This technology was based on the transfer and embedding of carbonized patterns formed by selective laser pyrolysis of thermosetting polymers such as polyimide. At the optimum laser setting, highly porous carbon nanoparticles with sheet resistances can be generated [291]. In 2015, X. Ren et al. reported the growth of graphene shells sealed on the surface of Ga NP using pulsed laser deposition (PLD) techniques. It is applied as a high performance substrate for surface enhanced Raman scattering (SERS). The SEM images showed that all Ga NPs were spherical and smooth with an average diameter of about 265 nm. TEM images indicated that Ga NP was completely encapsulated by multilayer graphene having a thickness of about 5 nm [292]. Graphene oxide (GO) was a compound of carbon, oxygen, and hydrogen and was a material similar to graphene. Most importantly, it can be reduced to an intermediate that can be used to expand graphene. This synthetic graphene was considered to be reduced graphene oxide (rGO) due to the presence of oxygen after reduction. In 2016, N. Deng et al. reported the rapid synthesis and patterning of large scale multilayer graphene films from graphene oxide films utilizing a 650 nm commercial laser on any substrate by one-step laser scribing technique [293]. In 2016, D. Wang et al. reported the direct reduction of GO to rGO by laser. The rGO and electrodes are then encapsulated by PDMS. It is manufactured as a SOGS strain sensor that can be used in wearable devices. SOGS strain sensors had a good balance between high sensitivity and large strain. Therefore, they had great potential in flexible electronics and wearable devices [294]. In 2017, S. Lin et al. reported the preparation of graphene heaters with excellent electrothermal performance and high sensitivity through drop casting and laser reduction. The laser source had a wavelength of 650 nm, a power of 143 mW, and a scan speed of 8.49 mm s^−1^. Reducing GO can improve the electrothermal performance of laser-reduced graphene oxide (LRGO) [295].

Graphene patterned on a wooden surface can be fabricated into various high performance devices such as hydrogen evolution and oxygen evolution electrodes. In 2017, R. Ye et al. demonstrated the conversion of wood to layered graphene by using CO_2_ laser scribing and when a laser was emitted in an inert atmosphere, the absence of an oxidant resulted in a more stable graphene structure. The higher resolution TEM image showed clear graphene stripes with a characteristic d-spacing of 0.34 nm and few layered structures [296]. The simplicity of obtaining a porous graphene surface on biodegradable substrates such as wood, paper, coconut, potato, cardboard, and cloth. In 2018, Y. Chyan et al. reported a method for converting a variety of substrates into laser induced graphene (LIG) using multi-pulse laser scribing [297]. A well-defined and ordered phenyl-C_61_-butyric acid methyl ester (PCBM) nanostrip pattern can be achieved with a linewidth of approximately 200 nm and a repetition rate of approximately 2.900 mm^−1^ over a large area of 1 cm^2^. In 2018, J. Enevold et al. proposed a laser interference patterning method for fabricating large-area semiconductor fullerene nanostructure arrays [298]. In 2018, L. Guo et al. presented a report on the production of NO_2_ gas sensors by reducing the graphene oxide (GO) in room temperature by two-beam laser interference (TBLI). The SEM image of the GO film showed that the GO film was patterned into layered nanostructures. The layered nanostructures can be produced by periodic reduction and ablation of the layered graphene oxide stack [299]. In 2018, M. Martinez-Calderon et al. presented a fast, accurate and convenient method for customizing the optical properties of diamond surfaces by using laser induced periodic surface structures (LIPSS) [300]. Diamond is considered a promising electrode material in the fields of cell stimulation, energy storage, (bio) sensing, and catalysis. In 2018, A. F. Sartori et al. reported an integrated characterization of the electrochemical and morphological/structural properties of laser-induced periodic surface structures (LIPSS) formed on heavily boron-doped CVD diamond (BDD) [301]. The vertical step between each horizontal grid line was set to 100 μm to align the ablated regions. The laser pulse energy was kept constant at 0.65 mJ, thus corresponding to a fluence of 3.25 J cm^−2^.

### 5.4. Piezoelectric Material

Zinc oxide was a promising functional material with potential applications from catalytic, sensing, optoelectronic to photonic and piezoelectric applications. Its wide direct bandgap (3.37 eV) and large exciton binding energy (60 meV at room temperature) made it an interesting candidate for solid-state lasers. In 2007, F. Claeyssens et al. reported that ZnO NR arrays have been generated on Si substrate patterning with Zn metal by laser induced forward transfer (LIFT) technology in air and vacuum and the Zn transfer pattern in vacuum resulted in a micron resolution densely aligned array of ZnO nanorods (NRs) [302]. In 2007, S. Z. Li et al. reported the fabrication of self-assembled ZnO nanorod arrays on silicon substrates by PLD in an argon atmosphere. Then the ZnO nanorods were coated with a layer of Er_2_O_3_ on the surface to form a core-shell structure. TEM showed an approximately 10 nm thick Er_2_O_3_ shell around a 50 nm thick ZnO core [303]. In 2007, Z. W. Liu et al. reported the growth of C-axis oriented ZnO nanorods on sapphire and silicon substrates by conventional pulsed laser deposition without catalyst. Studies on the effects of PLD parameters have shown that nanorods can be synthesized over a relatively wide substrate temperature range (550–700 °C) at high oxygen pressures (5–20 Torr) [304]. In 2007, L. Yang et al. exhibited that liquid phase pulsed laser ablation (LP-PLA) was a method for synthesizing a series of nanomaterials with controlled size and morphology like synthesizing zinc oxide ‘nano-leaf’ structure and XRD studies showed that after only 30 min of ablation in 0.01 M SDS, one-dimensional spherical particles with a size of about 5–25 nm appeared [305]. In 2008, Y. Guan et al. described the synthesis and combination of Zn and ZnO nanoparticles into linear arrays by laser-assisted chemical vapor deposition (LCVD). After annealing at 900 °C for 3 h in air, the crystal morphology changed. Nanocrystals with a diameter of 50–100 nm were grouped in groups of 2 or 3 with a linewidth of approximately 180 nm [306]. In 2008, R. Guo et al. studied the effects of different substrate conditions on the morphology, crystal structure and photoluminescence of nanoparticle-assisted pulsed laser ablation deposition of ZnO nanostructures [307].

In 2008, M. Huang et al. reported the production of uniform, planar nano-grating, micro/nano-square structures of any size on wide-bandgap materials by direct femtosecond (fs) laser ablation. The feature size of the formed nanostructures can be adjusted over a wide range by varying the illumination wavelength, which was about 200 nm under 800 nm fs laser irradiation [308]. In 2008, T. Jia et al. reported the preparation of two-dimensional periodic nanostructures on ZnO crystal planes using double-beam interference of 790 nm femtosecond laser. The long period was determined by the interference pattern of the two laser beams. Another short-period nanostructure was a period of 220–270 nm embedded in a long-period structure [309]. In 2008, L. C. Tien et al. reported the microstructure and growth behavior of vertically aligned zinc oxide (ZnO) nanowires generated by pulsed laser deposition (PLD) on ZnO thin film templates. At a background pressure of 500-mTorr argon or oxygen, the growth of the nanowires leaded to nanowire diameters as small as 50–90 nm, while the diameter was dependent on growth pressure and temperature [310]. In 2009, C. Fauteux et al. reported the synthesis of high quality ZnO nanostructures with various morphologies in a very short time scale (several seconds) by laser-induced chemical liquid deposition (LCLD). In fact, nanowires having an aspect ratio of up to 80 (average length of 2.3 µm and average diameter of 29 nm) have been obtained [311]. In 2009, B. Yang et al. reported the fabrication of different 1D self-assembled ZnO nanostructures on different substrates by pulsed laser deposition, such as nanoneedle, nanowire, flower, and nanocrystal particle [312]. In 2010, C. Cibert et al. reported the deposition of aluminum nitride films at room temperature using pulsed laser deposition and studied their nanostructure and piezoelectric properties as a function of fluence. For all fluences, the films were composed of an amorphous AlN matrix containing crystalline AlN nanoparticles ranging in size from 6–7 nm. These nanoparticles exhibited a good piezoelectric response [313].

In 2010, C. Lo et al. demonstrated the use of polymeric CO_2_ lasers to heat bulk ZnO rods to generate various forms of ZnO nanostructures, including comb, nanotubular, fence-like, pencil-like and flower-like nanostructures, and nanowires [314]. As shown in the SEM image of Figure 30, the ZnO nanostructure morphology fabricated by conventional furnace heating techniques can be prepared in simple laser heating technique. These different nanostructures were grown on the edges of the laser burned holes. A typical burned hole having an irregular shape was showed in the inset of Figure 30a which was about growth momentum and time and a region close to the burned hole (indicated by a dotted square) was enlarged and shown in Figure 30a. It can be seen from Figure 30a that most of the nanowires grew there except in the region very close to the hole. According to the study, the length distribution of nanowires can range from less than 5 µm to more than 100 µm, and diameters ranged from less than 100 nm to greater than 1000 nm. In addition to nanowires and nanoparticles, various shapes of ZnO nanostructures, as seen in the literature, including nanowalls, nanotubes, nanotubes, nanorods, and as shown in Figure 30b–g, pencil-like and flower-like nanostructures were observed, respectively. In 2010, W. Liang et al. introduced the periodic surface nanostructure induced by femtosecond laser pulses on polycrystalline ZnO. A grating-like nanostructure having an average period of 160 nm can be fabricated by shifting the sample line by line under appropriate irradiation conditions [315]. In 2010, R. Vinodkumar et al. reported the preparation of Al-doped ZnO thin films by pulsed laser deposition on a quartz substrate at a substrate temperature of 873 K and a background oxygen pressure of 0.02 mbar. Particle size calculations based on XRD analysis indicated that all films were nanocrystals with quantum dots between 8 and 17 nm in size [316]. In 2011, S. Garry et al. reported the application of conventional Nanosphere lithography (NSL) technology with gold as a catalyst to demonstrate an ordered array of ZnO nanowires on a ZnO-coated substrate by PLD. Nanowires were fabricated by gas phase transport (VPT) growth in a tube furnace system and grew only in areas where Au was pre-patterned [317].

Recently, two-dimensional (2D) nanostructured ZnO ordered porous arrays have received extensive attention due to their unique properties and potential applications in catalysis, ultrafiltration and biomineralization of microelectronics, optoelectronics and biomedical devices. In 2011, H. Niu et al. demonstrated a novel method for the selective growth of ZnO nanoparticles and nanoporous membranes through vertical pulsed-laser catalyst-free ablation (VPLD) [318]. In 2014, B. Zein et al. reported the synthesis of a catalyst-free zinc oxide (ZnO) nanowire network with a honeycomb structure by pulsed laser deposition (PLD). At 7 min of deposition, coalescence occurred between the ZnO islands. Starting from 10 min, the two-dimensional ZnO Nanowalls Network (NWaNs) grew vertically on the substrate. The pores were between 50 and 140 nm in size and the wall thickness between the cells was about 50 nm [319]. In 2014, R. Fonseca et al. proposed a two-photon-based direct laser writing method to fabricate ZnO nanowire/polymer composites into three-dimensional microstructures that achieved sub-micron spatial resolution [320]. In 2014, A. Marcu et al. reported the growth of ZnO nanowires on gold patterned and unpatterned alumina surfaces with high repetition rate laser UV ablation [321]. In 2014, M.A. Susner et al. reported the growth of ZnO nanostructures on Si (111) by pulsed laser deposition. The morphology of ZnO was adjustable based on the atmospheric pressure during deposition. For all laser powers (0.6, 0.8, 1.0 and 1.2 W), SEM studies have shown that as the laser power increased, the nanowire length increases and the morphology improved [322]. In 2016, N. Nedyalkov et al. proposed a laser machining of a thin zinc oxide film deposited on a metal substrate. The ZnO thin film was generated on a ruthenium substrate under oxygen atmosphere by classical nanosecond pulsed laser deposition method. The resulting film was then processed through a nanosecond laser pulse at a wavelength of 355 nm [323]. In 2017, M. P. Navas et al. reported that ZnO nanoparticles were grew in different shapes using laser ablation, like spheres, nanorods, branched nanorods, nanoflowers, nanoflakes, and nanosquares [324]. In 2018, P. Ghosh et al. reported the deposition and characterization of ZnO nanostructures using pulsed laser deposition at various ambient pressures of oxygen on a silicon substrate at 600 °C [325].

## 6. Conclusions and Outlook

Laser machining has become a three-dimensional, maskless, high-throughput, versatile technology for the production of a wide range of nanostructured materials. In this review, we show the laser fabrication of micro/nano structures of various materials and give the feature sizes and possible applications for each structure. The first part introduces the performance of micro/nano structural materials, the application field and introduces the characteristics of laser machining as a new processing technology and simply gives the physical mechanism of long pulse laser and short pulse laser machining. The second part introduces the micro/nano structures prepared by laser machining on metal materials. According to the topographical features of the structure, the basic information of the prepared structure such as laser source, feature size, possible applications, and the like are analyzed. The third part introduces various micro/nano structures prepared by laser machining in organic matter and high molecular polymers. Various 3D micro/nano structures prepared by TPP and DLW techniques are highlighted. The fourth part is divided into four small parts according to the four materials of salt glass, oxide, carbon material, and piezoelectric material. The characteristic dimensions of micro/nano structures prepared by laser machining of four materials, laser source equipment and possible applications of micro/nano structures are introduced. This article is a summary of the preparation of micro/nano structures on various materials by lasers in the past two decades. It provides a guide of laser machining for researchers involved in the preparation of micro/nano structures.

Although remarkable scientific achievements have been made so far, laser machining is still an emerging research field. From the initial laser surface texturing to the current laser machining to prepare a variety of design micro/nano structures. Laser machining has made great progress. Short pulse lasers, especially femtosecond laser machining, have proven to be an impressive technique for producing a variety of surface micro/nano structures that are otherwise difficult to obtain. In order to improve laser machining technology, efforts should be focused on potential mechanisms to better control the structural changes and/or morphology of the laser-affected zone, and to perfect beam shaping techniques to precisely adjust the geometry of the laser writing structure and thereby improve efficiency and resolution of manufacturing. Recently, N. Kurra et al. [326] demonstrated that laser-derived graphene (LDG) electrodes are becoming a promising three-dimensional graphene electrode. The LDG can be obtained by irradiation with various laser sources including a CO_2_ infrared laser and a femtosecond laser pulse. Controlling the doped microstructure, number and type, and post-deposition methods enable a variety of applications, including energy storage, catalysis, sensing, and biomedical. In summary, laser machining represents an emerging micro/nano structure processing technology that has been shown to be useful in a variety of applications. However, the technology is still in the early stages of development and it is expected that fine control of material properties and discovery of new applications will continue in the future.

## Figures and Tables

**Figure 1 nanomaterials-09-01789-f001:**
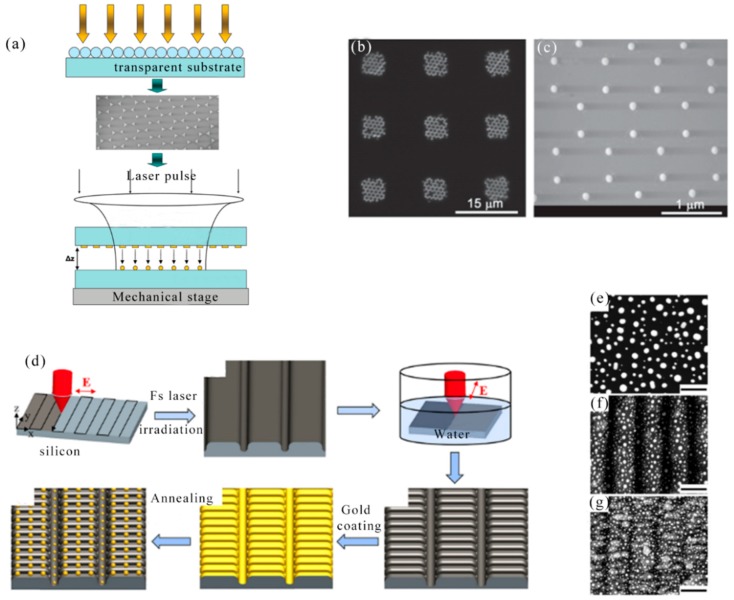
Laser-assisted preparation of metal nanoparticle arrays (**a**) Scheme of nanoparticle structure fabrication by a combination of the nanosphere lithography and laser-induced transfer. (**b**) Dark-field microscope image of arrays of gold nanoparticles fabricated by single laser pulses on a receiver substrate. Laser fluence is of 0.06 J/cm^2^. (**c**) Top-view scanning electron microscope (SEM) image of nanoparticle arrays prepared by a single laser pulse. (**d**) Schematic process of the Au nanoparticle-decorated nanorod (NPDN) substrate fabrication. SEM images of Au nanoparticles (NPs) formed on three categories of substrates coated with 10 nm Au films and annealing: (**e**) flat Si, (**f**) ripple, and (**g**) nanorod substrates. Scale bars indicate 500 nm. (**a**–**c**) reproduced with permission from [62], ACS, 2011; (**d**–**g**) reproduced with permission from [63], ACS, 2018.

**Figure 2 nanomaterials-09-01789-f002:**
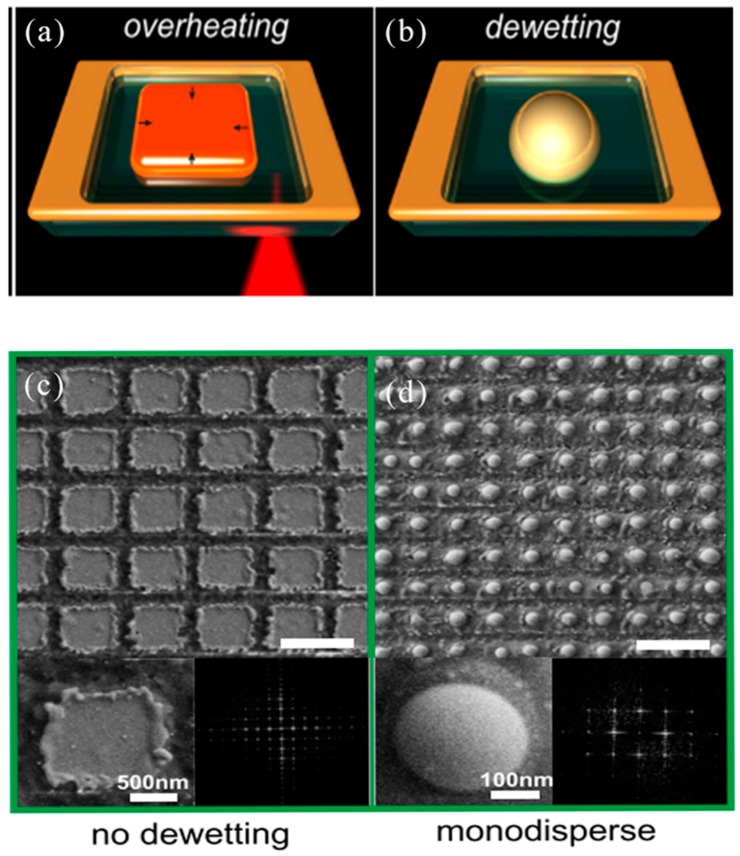
Metal nanoparticles fabricated by laser dewetting (**a**,**b**) Schematic diagram of a femtosecond laser cutting a patch on an Au film on a dielectric substrate to form a single nanoparticle; (**c**) SEM image before laser dewetting; (**d**) SEM image of an Au nanoparticle array made of a 30 nm film on a SiO_2_ substrate at a fluence of 40 mJ/cm^2^. The lower right insets: Fourier spectra of the SEM images. The lower left insets: enlarged SEM images of typical nanoparticles from the arrays. (**a**–**d**) reproduced with permission from [67], Wiley, 2016.

**Figure 3 nanomaterials-09-01789-f003:**
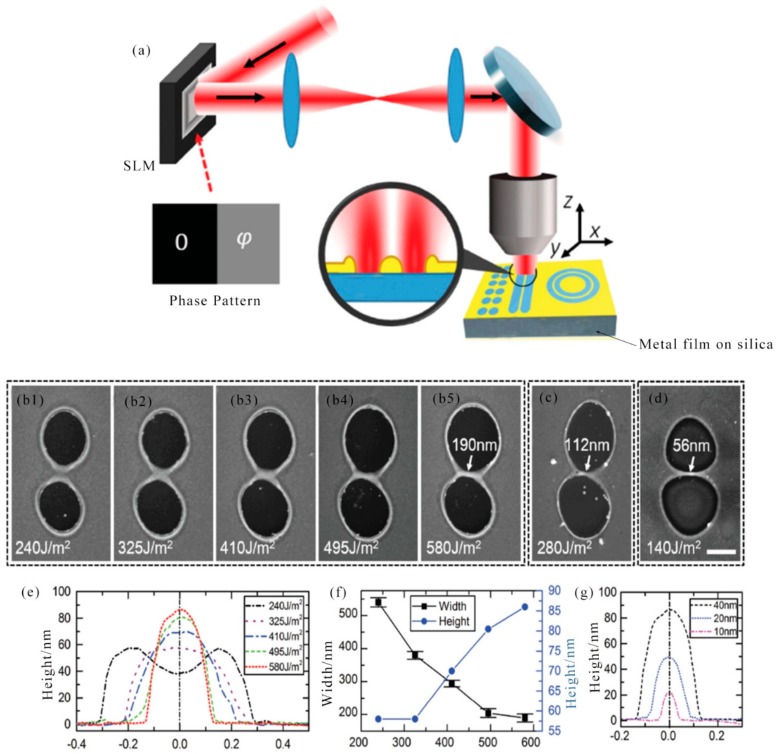
The preparation of high-resolution nanowires by Space-modulated femtosecond laser beam. (**a**) Schematic diagram of the experimental setup. (**b1**–**b5**) SEM images of the short nanowires formed between the two spots with increasing pulse energy. (**c**,**d**) Short nanowires formed using thin gold film with (**c**) 20 nm and (**d**) 10 nm thickness (both consisting of 3 nm Cr as the adhesion layer). The scale bar is 1 µm. (**e**) The cross-section of the nanowires shown in (**b**) measured by atomic force microscopy (AFM). (**f**) Section area decreased and height increased with the pulse energy. (**g**) The cross-section of the minimum nanowires using different film thicknesses was measured by AFM. (**a**–**g**) reproduced with permission from [74], Wiley, 2015.

**Figure 4 nanomaterials-09-01789-f004:**
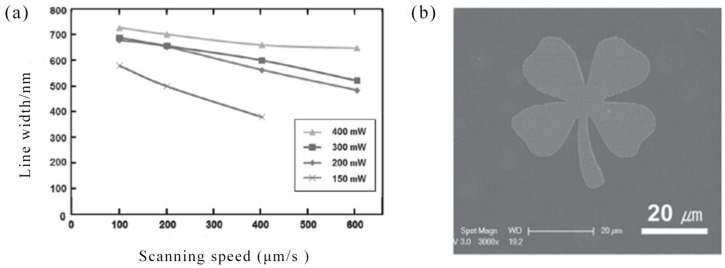
The preparation of high-resolution patterns by femtosecond laser selective nanoparticles (NPs) sintering (FLSNS). (**a**) Variation of the line width depending on the scanning speed at 150, 200, 300, and 400 mW laser powers. The inset shows a scanning electron microscopy (SEM) image of the fabricated metal line at 150 mW and 400 μm s^−1^. (**b**) SEM images of the fabricated 2D metal clover patterns. (**a**,**b**) reproduced with permission from [86], Wiley, 2011.

**Figure 5 nanomaterials-09-01789-f005:**
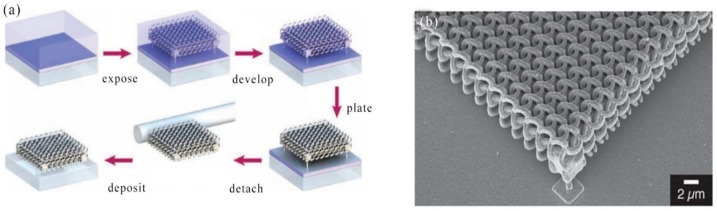
The laser-assisted fabrication of 3D structures. (**a**) Fabrication scheme of 3D metallic bichiral crystals through direct laser writing and electroless silver plating. (**b**) SEM images of a bichiral crystal with right-handed corners and righthanded helices after electroless silver plating. (**a**,**b**) reproduced with permission from [95], Wiley, 2011.

**Figure 6 nanomaterials-09-01789-f006:**
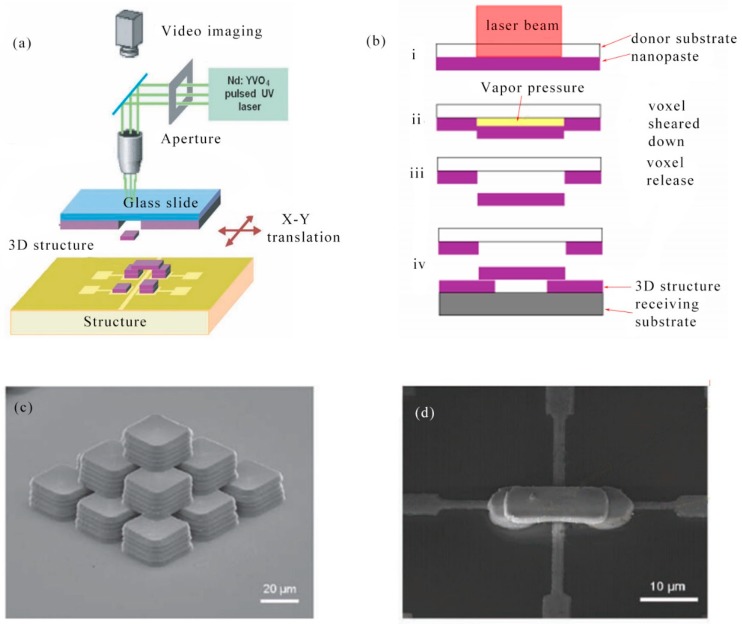
Diagram of 3D laser direct writing technology to prepare 3D micro/nano structure. (**a**) Schematic diagram of the apparatus used for 3D laser direct-write. (**b**) Conceptual diagrams illustrating the basic steps for the non-contact 3D laser direct write process. (**c**) A high aspect ratio micro pyramid. (**d**) SEM image of an interconnect bonding Cu electrodes on Si. (**a**–**d**) reproduced with permission from [101], Wiley, 2010.

**Figure 7 nanomaterials-09-01789-f007:**
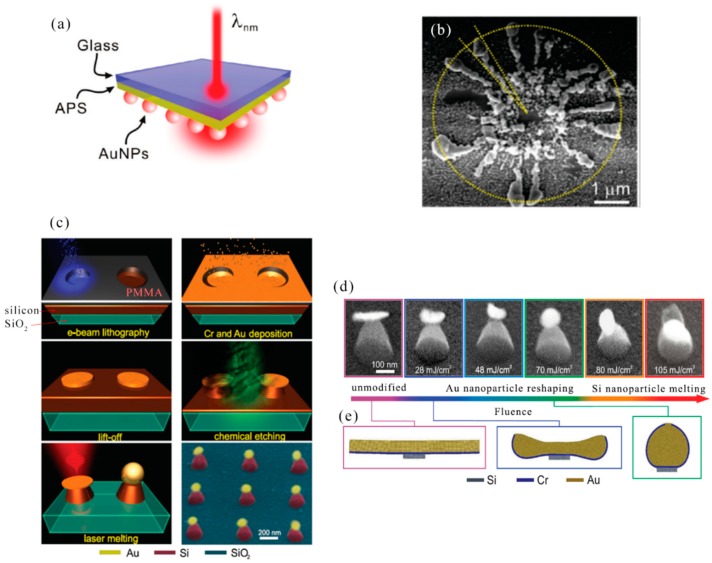
Diagram of 3D micro/nano structures by laser machining. (**a**) Schematic of the construction of the samples used during laser machining. (**b**) SEM images of the Ag deposits resulting from illumination from the glass with 4.25 mW of 632.8 nm laser light focused with a 50× objective lens to a ~1 µm spot after 10 s of illumination and the yellow dotted line presents that the width of the silver deposit increases approximately proportionally with the radial distance from the center of the deposit. (**c**) Schematic diagram of asymmetric metal dielectric (Au/Si) nanoparticles prepared by photolithography of femtosecond laser melting. (**d**) The SEM images correspond to the typical structures in the following modification regimes: Au nanodisc deformation, transformation to Au nanospheres, Si nanocone melting, and damage of the nanodimer. (**e**) Molecular dynamics simulation of the Au nanoparticle reshaping. (**a**,**b**) reproduced with permission from [104], ACS, 2010; (**c**–**e**) reproduced with permission from [105], Wiley, 2016.

**Figure 8 nanomaterials-09-01789-f008:**
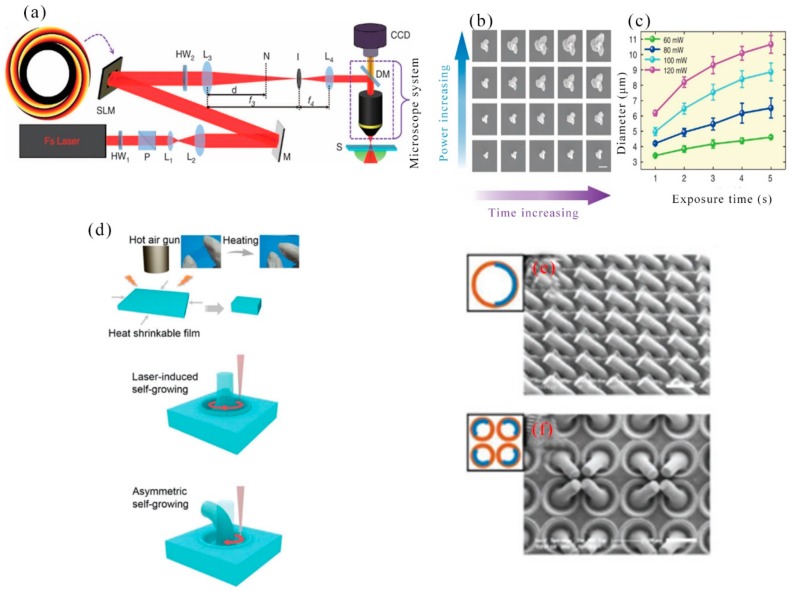
Diagram of large area micro/nano structure prepared by laser machining. (**a**) Schematic diagram of an experimental setup for 3D chiral microstructure in an isotropic material based on spatial light modulator (SLM). (**b**) SEM images of chiral microstructures with three spiral lobes achieved for different values of laser power and exposure time. (Scale bar, 5 μm). (**c**) Quantitative study of the diameter of the chiral microstructure as a function of exposure time and power. (**d**) Schematic of heat-induced shrinkage of shape-memory polymer (SMP) film by a hot air gun. Inset: pictures of polystyrene film before and after heating. The red semicircle indicates the laser scanning path. (**e**,**f**) SEM of polystyrene film after laser machining. (**a**–**c**) reproduced with permission from [113], Nature, 2017; and (**d**–**f**) reproduced with permission from [114], Wiley, 2018.

**Figure 9 nanomaterials-09-01789-f009:**
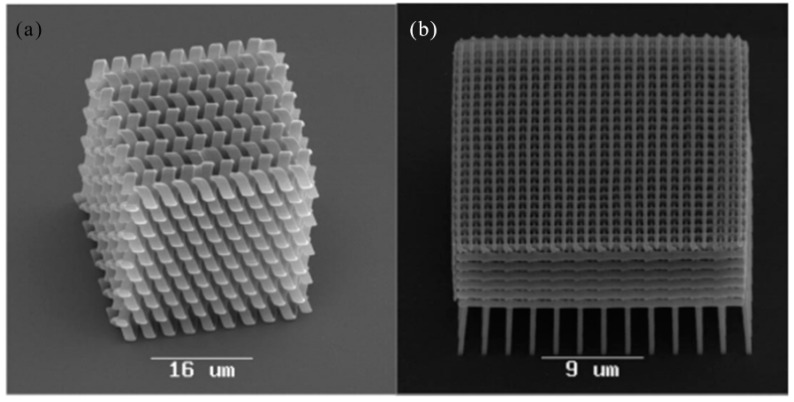
Three-dimensional photonic crystal structures fabricated by the two-photon polymerization (TPP) technique. (**a**,**b**) reproduced with permission from [118], ACS, 2008.

**Figure 10 nanomaterials-09-01789-f010:**
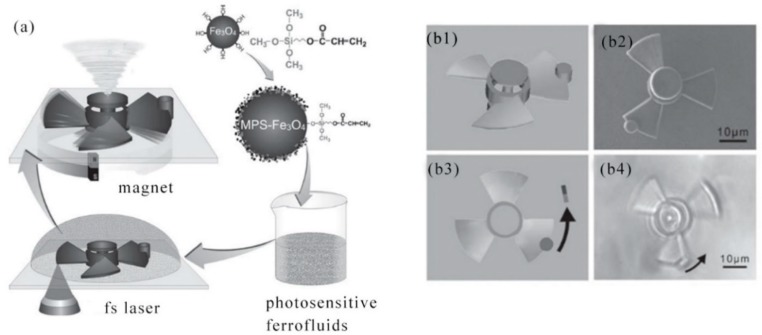
Diagram of 2D/3D polymer structures by two-photon polymerization lithography (**a**) Fabricative procedures of remotely controllable micronanomachines. (**b1**–**b4**) SEM images of the micro-turbine. (**a**,**b**) reproduced with permission from [121], Wiley, 2010.

**Figure 11 nanomaterials-09-01789-f011:**
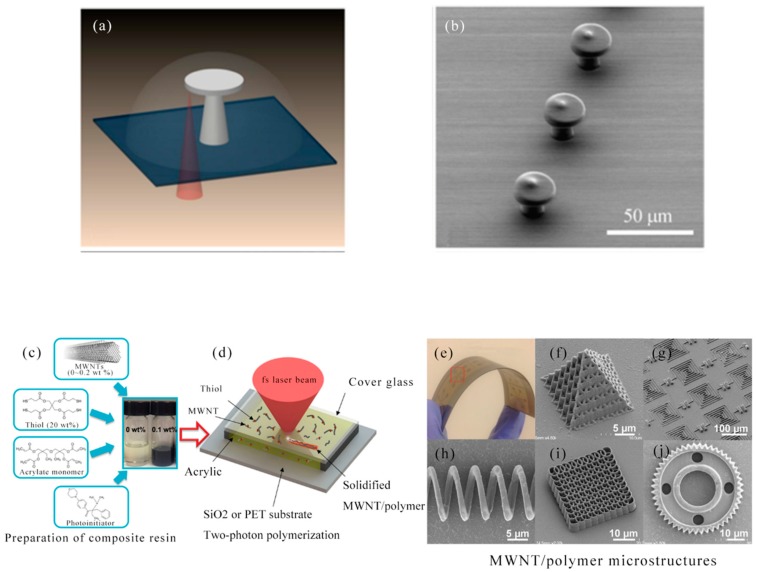
Illustration of 2D/3D polymer structures by two-photon polymerization lithography (**a**) Diagram of the fabrication process of high-performance asymmetric polymer microcavities on a cover glass. (**b**) SEM image of asymmetric polymer microcavities. (**c**) Experimental procedure in preparing multiwalled carbon nanotube-thiol-acrylate (MTA) composite resins. (**d**) Experimental setup of two-photon polymerization (TPP) fabrication. (**e**) Physical map of a bent polyethylene terephthalate (PET) substrate. (**f**–**j**) SEM micrographs of various functional micro/nano structures. The laser power and scanning speed used in the TPP fabrication were 15 mW and 0.5 mm s^−1^, respectively. (**a**,**b**) reproduced with permission from [129], AIP, 2013; and (**c**–**j**) reproduced with permission from [130], Wiley, 2016.

**Figure 12 nanomaterials-09-01789-f012:**
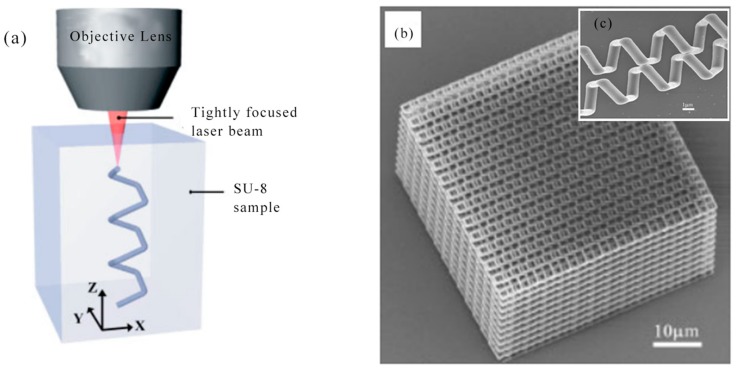
Diagram of extended periodic microstructure of a two-photon direct laser writing (DLW). (**a**) Pinciple of two-photon lithography. (**b**) SEM images of spiral structures. (**c**) A detailed SEM view of the individual spirals. (**a**–**c**) reproduced with permission from [134], Wiley, 2005.

**Figure 13 nanomaterials-09-01789-f013:**
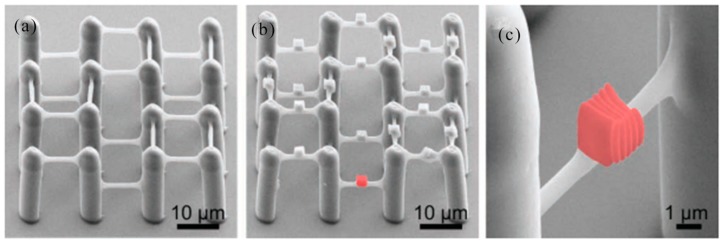
Diagram of 3D composite polymer scaffolds by DLW. (**a**) 3D frameworks are polymerized and fabricated. (**b**) Frameworks are then cast with the photoresist Ormocomp and cubes (one cube is highlighted in red) are precisely attached to the polyethylene glycol diacrylate (PEG-DA) beams in a second DLW step. (**c**) Higher magnification image of an Ormocomp cube (highlighted in red). (**a**–**c**) reproduced with permission from [137], Wiley, 2011.

**Figure 14 nanomaterials-09-01789-f014:**
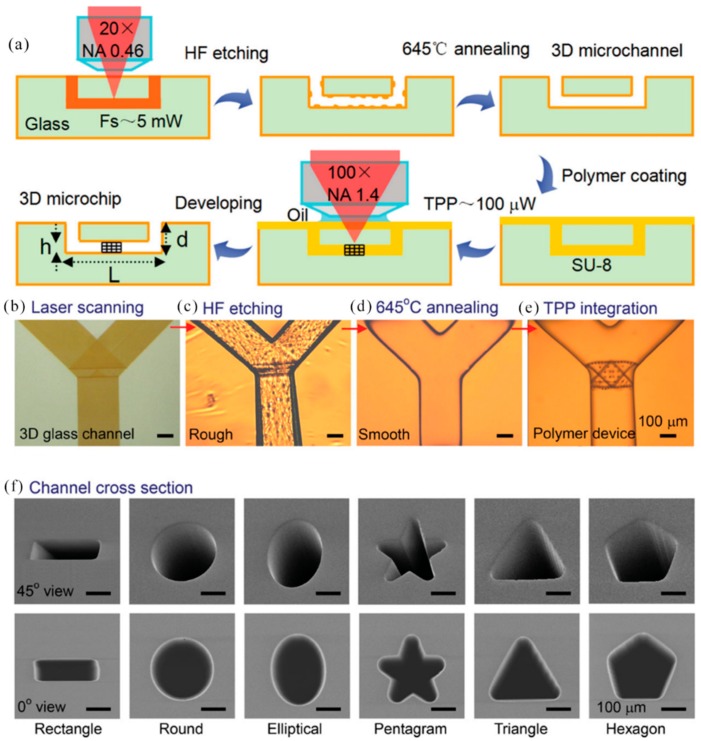
3D polymer micro/nano structures by hybrid femtosecond laser micromachining. (**a**) Schematic illustration of the fabrication procedure for a 3D ship-in-a-bottle biochip by hybrid fs laser micro processing. 3D Y-shaped microchannel after laser scanning and the first annealing (**b**), hydrofluoric acid (HF) etching (**c**), second annealing (**d**) and 3D integration of polymer microstructure by TPP (**e**). SEM images of controllable cross sections of the 3D micro-channels. (**a**–**f**) reproduced with permission from [141], Wiley, 2014.

**Figure 15 nanomaterials-09-01789-f015:**
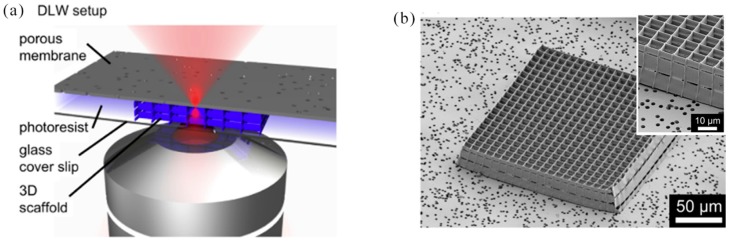
Diagram of 3D cell culture scaffolds by DLW. (**a**) Schematic diagram: direct laser writing (DLW) was applied to fabricate non-cytotoxic 3D scaffolds on porous membranes. (**b**) SEM side view and top view images of a 3D scaffold with a mesh size of 10 μm resting on a membrane with a pore diameter of 3 μm. (**a**,**b**) reproduced with permission from [142], Elsevier, 2013.

**Figure 16 nanomaterials-09-01789-f016:**
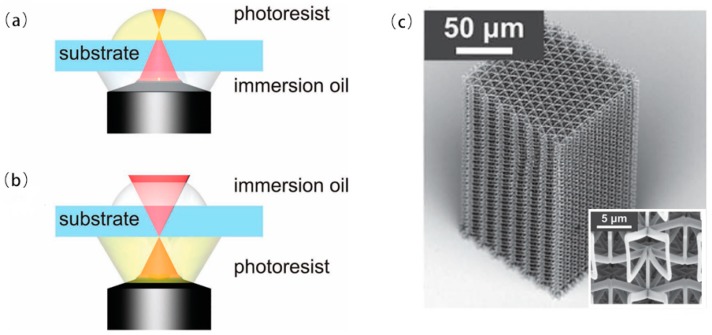
Three-dimensional microstructured mechanical metamaterial by DLW. (**a**) Diagram of conventional three-dimensional direct laser writing lithography. (**b**) Diagram of a new immersive 3D-DLW method. Sample height is no longer fundamentally restricted. (**c**) Oblique-view electron microscopy images of selected fabricated mechanical metamaterials on glass substrates. (**a**–**c**) reproduced with permission from [145], Wiley, 2012.

**Figure 17 nanomaterials-09-01789-f017:**
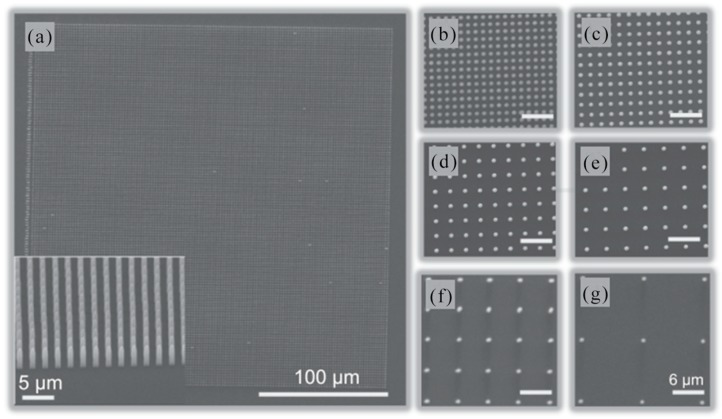
Diagram of polymer nanopillars (NP) using 3D direct laser writing. (**a**) View of a full 250 × 250 µm^2^ NP array and an inset of a tilted SEM about the edge of an array. (**b**–**g**) Different polymeric NP center-to-center spacings: 1.5 µm (**b**), 2 µm (**c**), 3 µm (**d**), 4 µm (**e**), 6 µm (**f**), and 12 µm (**g**). All NPs here have length 3 µm. (**a**–**g**) reproduced with permission from [152], Springer Nature, 2017.

**Figure 18 nanomaterials-09-01789-f018:**
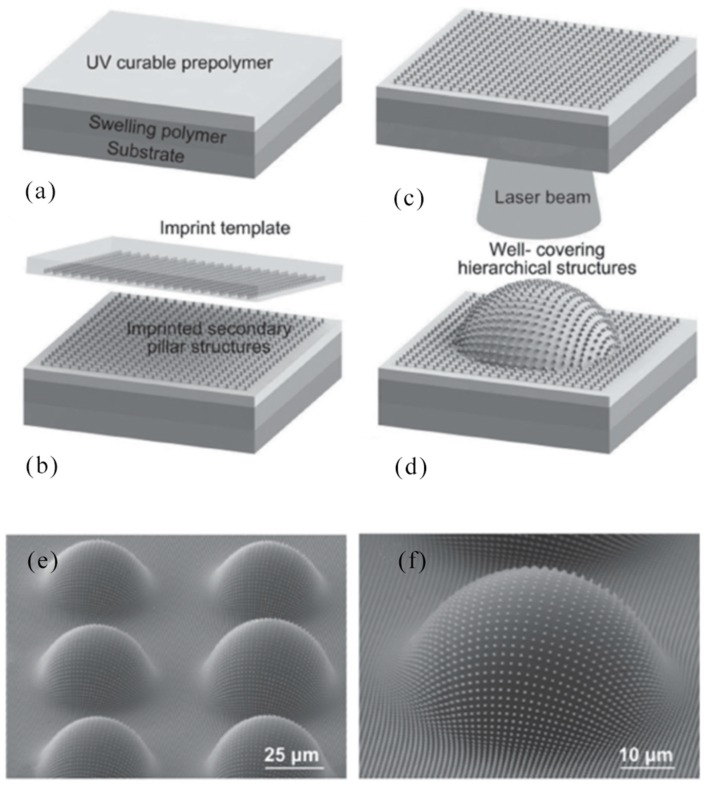
Diagram of micro/nano-scale combined layered structures by laser swelling technology. (**a**–**d**) Schematic diagram of the fabrication process: (**a**) Swelling polymer and UV curable prepolymer are fabricated on a glass substrate. (**b**) Secondary pillar structures are prepared by nanoimprinting. (**c**) Laser beam is irradiated on the swelling polymer to form a primary structure from the side of substrate. (**d**) Formation of fully covering hierarchical micro/nano structures. (**e**,**f**) SEM images of various hierarchical structures. (**a**–**f**) reproduced with permission from [162], Wiley, 2014.

**Figure 19 nanomaterials-09-01789-f019:**
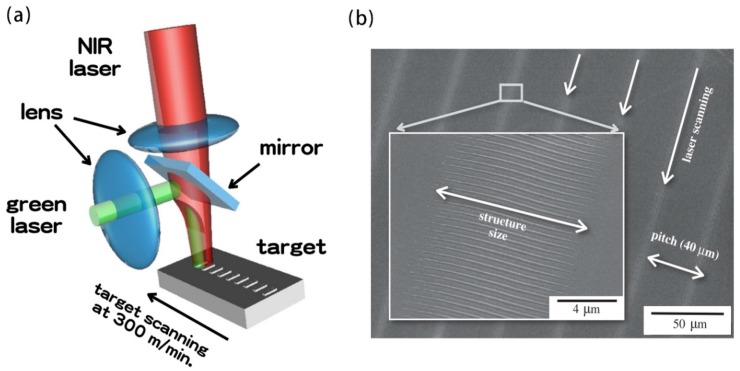
Laser-induced periodic surface structure by continuous wave (CW) laser. (**a**) schematics of two CW laser system. (**b**) SEM image of laser induced periodic surface structures (LIPSS). The laser irradiates the target at a scanning speed of 300 m/min. The inset is an enlarged view. (**a**,**b**) reproduced with permission from [171], Elsevier, 2013.

**Figure 20 nanomaterials-09-01789-f020:**
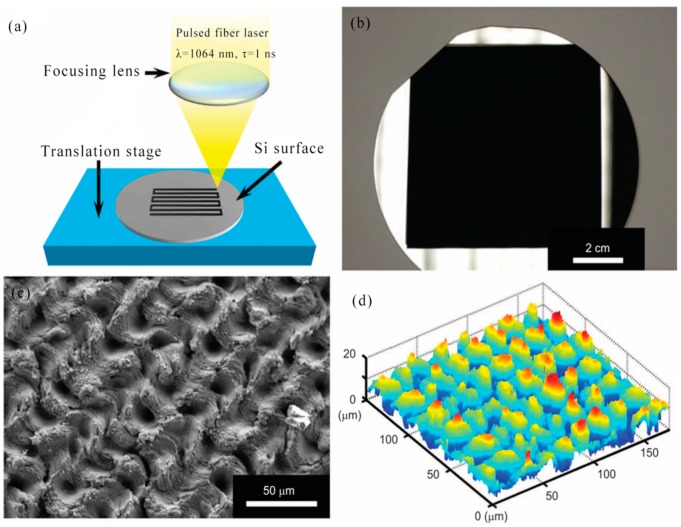
Diagram of anti-reflective microstructures on Si surfaces by DLW. (**a**) Schematic of fiber laser ablation for black Si surface fabrication. (**b**) Image of the textured black Si surface. (**c**) SEM image and (**d**) 3D surface profile. (**a**–**d**) reproduced with permission from [184], Springer Nature, 2014.

**Figure 21 nanomaterials-09-01789-f021:**
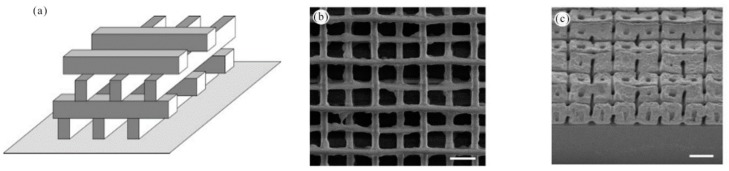
Diagram of 3D photonic crystals by DLW. (**a**) Schematic of the woodpile structure. (**b**,**c**) SEM images of a methylsilsesquioxane (MSQ) woodpile structure. The spacing between the lines was 4 μm and the lateral size of the photonic crystals was about 116 × 116 μm. (**a**,**c**) reproduced with permission from [196], Wiley, 2008.

**Figure 22 nanomaterials-09-01789-f022:**
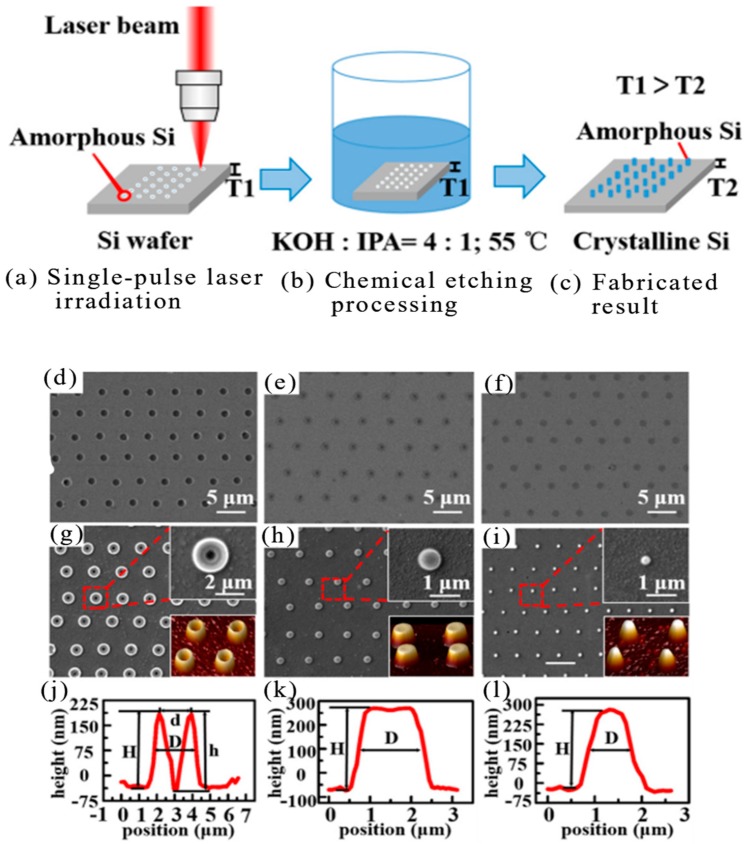
Diagram of silicon micro/nano structures of controlled size and shape by chemical translation assisted femtosecond laser single pulse irradiation. (**a**–**c**) Schematic diagram of the experimental process. (**d**–**f**) SEM images of a Si wafer after laser irradiation. (**g**–**i**) SEM images of a Si wafer after KOH etching for 90 s. Right upper insets are the magnified SEM images of single Si structures; Right lower insets are the AFM images of Si structures. (**j**–**l**) AFM images of single Si fabricated structure corresponding to (**g**–**i**). (**a**–**l**) reproduced with permission from [205], AIP, 2017.

**Figure 23 nanomaterials-09-01789-f023:**
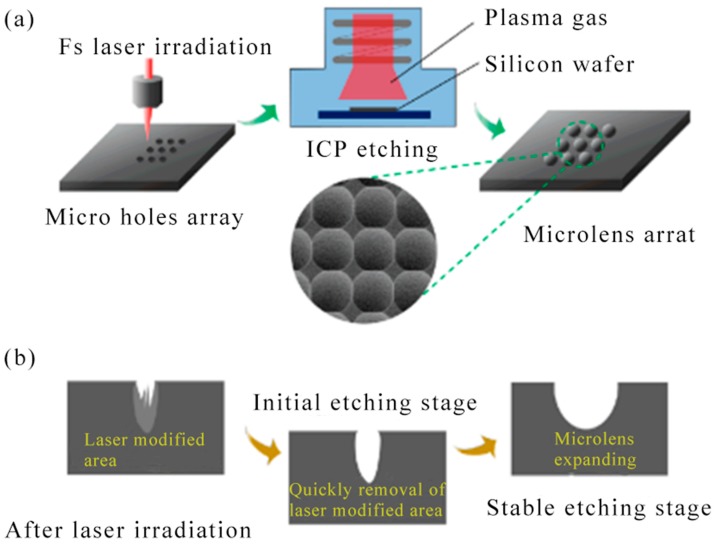
Diagram of silicon micro/nano structures fabricated by dry etching assisted femtosecond laser machining. (**a**) Diagram of the fabrication of silicon structures using dry etching assisted femtosecond laser machining (DE-FsLM). The inset is the SEM image of the prepared silicon concave structures. (**b**) Diagram of the cross-sectional profiles of the concave structures to present the etching process. (**a**,**b**) reproduced with permission from [211], Wiley, 2017.

**Figure 24 nanomaterials-09-01789-f024:**
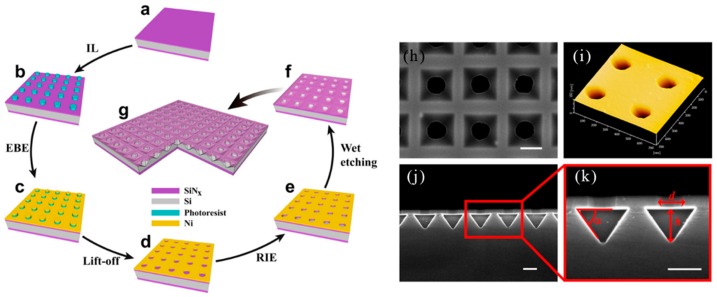
Diagram of inverted pyramidal pits (IPP) by laser machining. (**a**–**g**) Schematic view of the fabricate on process of 3D container which is composed of IPP array with nano-openings. (**h**,**i**) SEM and AFM images of the 3D container, respectively. (**j**) SEM image of the 3D container. (**k**) An detailed image of 3D containers with critical size parameters. (**a**–**k**) reproduced with permission from [217], Elsevier, 2016.

**Figure 25 nanomaterials-09-01789-f025:**
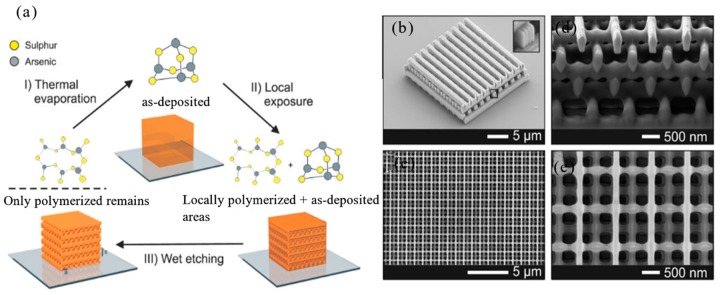
Diagram of the direct laser writing of a 3D photonic crystal. (**a**) The three fabrication steps of 3D photonic crystal by DLW. (**a**) Woodpile with rod distance 2 μm to illustrate the construction principle of the rods; Inset image shows Each rod is fabricated from eight parallel subrods to yield a rod aspect ratio of almost 1.0. (**c**–**e**) SEM images of As_2_S_3_ woodpiles. (**a**–**e**) reproduced with permission from [223], Wiley, 2006.

**Figure 26 nanomaterials-09-01789-f026:**
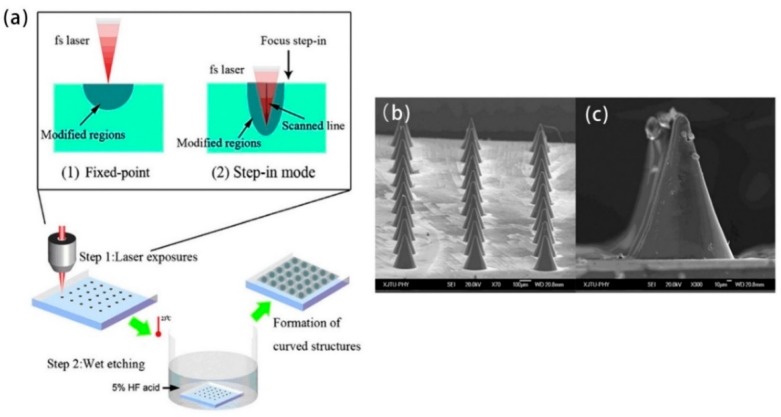
Diagram of 3D micro/nano structures by laser machining. (**a**) Schematic diagrams of the preparation process and the laser exposure methods. (**b**) Poly (dimethylsiloxane) (PDMS) replicas of the microwells of conical microstructures (**c**) magnified observations of (**b**). (**a**–**c**) reproduced with permission from [243], Elsevier, 2013.

**Figure 27 nanomaterials-09-01789-f027:**
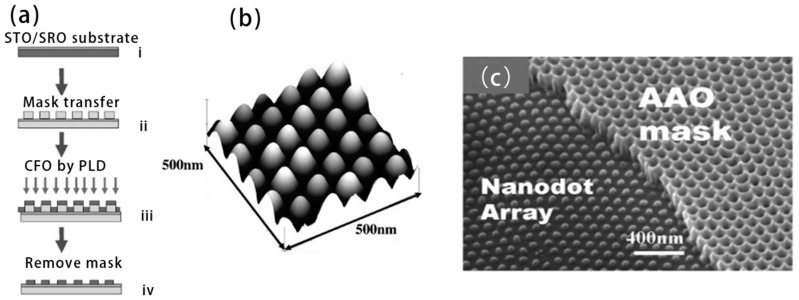
Diagram of nanodot arrays by pulsed laser deposition. (**a**) Procedure for the nanodot array fabrication. (**b**) AFM image of an as-deposited CoFe_2_O_4_ (CFO) nanodot array. (**c**) CFO nanodot array together with a partially removed anodic aluminum oxide (AAO) mask. (**a**–**c**) reproduced with permission from [254], Wiley, 2009.

**Figure 28 nanomaterials-09-01789-f028:**
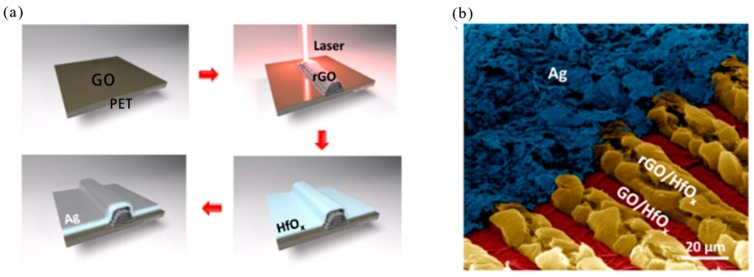
Diagram of the application of laser scribing graphene. (**a**). Diagram of the main fabrication processing steps of the laser scribing graphene-resistive random access memory (LSG-ReRAM). (**b**) Top view SEM image ofthe LSG-ReRAM in false color. (**a**,**b**) reproduced with permission from [284], ACS, 2014.

**Figure 29 nanomaterials-09-01789-f029:**
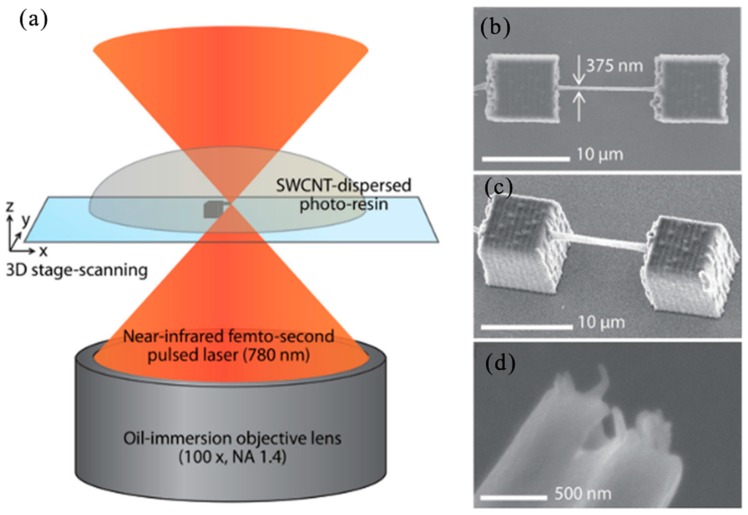
Diagram of single-walled carbon nanotubes (CNTs) 3D nanostructures fabricated by DLW. (**a**) Schematic showing 3D microfabrication of single-walled CNTs (SWCNT)/polymer composites based on TPP lithography. (**b**,**c**) SEM images of a 375 nm wide, 10 µm long nanowire, suspended between two microboxes. (**d**) SEM image of a cross-section of the nanowire. (**a**–**d**) reproduced with permission from [288], Wiley, 2014.

**Figure 30 nanomaterials-09-01789-f030:**
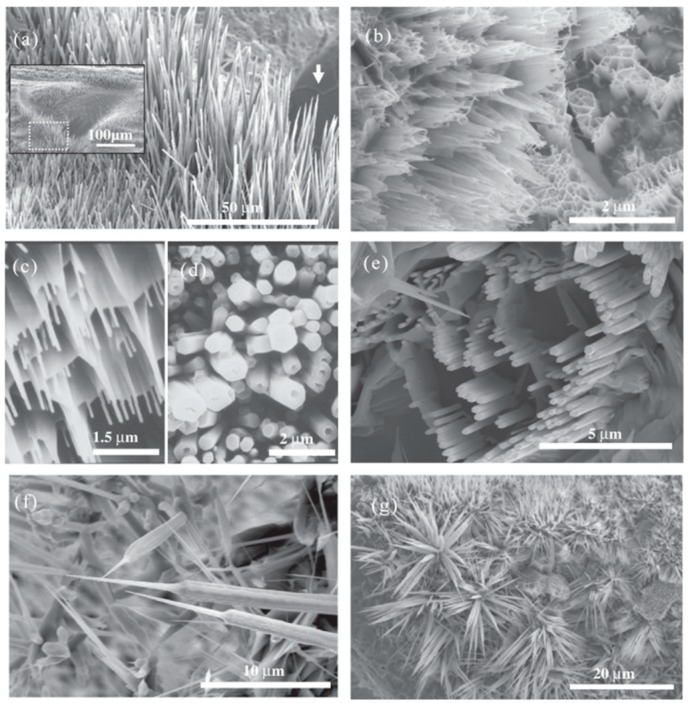
Diagram of ZnO nanostructures fabricated by CO_2_ lasers. (**a**–**g**) SEM images of the ZnO nanostructures with various morphologies. (**a**–**g**) reproduced with permission from [314], Elsevier, 2010.

**Table 1 nanomaterials-09-01789-t001:** Summary for nanoparticles prepared by laser machining on metals.

Materials	Laser Equiment	Feature Size	References
**Au/Ag**	325 nm He-Cd Laser	25 nm Au/20 nm Ag	[65]
**Au/Ni**	KrF excimer laser beam	160 nm	[58]
**Au**	excimer laser pulses	-	[59]
**Pd**	Ti:sapphire femtosecond laser system	2.0–6.0 nm	[68]
**Au**	femtosecond laser system	40–200 nm	[62]
**Ni**	pulsed Nd:YAG (yttrium aluminium garnet) laser	35 nm	[69]
**Ag**	Ti:sapphire femtosecond laser	10–100 nm	[64]
**Ag**	800 nm femtosecond laser	9 μm	[70]
**Au**	Nd:YAG laser	5.5–12.5 nm	[71]
**Au**	femtosecond laser system	-	[67]
**Au**	Ti:sapphire laser system	-	[66]
**Au**	Ti:sapphire laser system	-	[63]

**Table 2 nanomaterials-09-01789-t002:** Summary of nanowires prepared by laser machining on metals.

Materials	Laser Equiment	Feature Size	References
Cu	Ti:Sapphire laser system	5–6 µm	[72]
Au	Ti:sapphire laser	150–1000 nm	[73]
Au	Ti:Sapphire laser system	228 nm	[75]
Au	Ti:sapphire laser system	56 nm	[74]
Ag	Ti:sapphire laser system	186 nm	[76]

**Table 3 nanomaterials-09-01789-t003:** Summary of 3D micro/nano structures prepared by laser machining on metals.

Materials	Laser Equiment	Feature Size	References
Ag	Ti:sapphire laser system	400 nm	[89]
Au	Ti:sapphire femtosecond laser (740 nm, 100 fs, 82 MHz)	150 nm	[61]
Ag	Nd:YVO_4_ laser (355 nm, 30 ns)	1–10 μm	[101]
Au	Ti:sapphire laser with 780 nm, 100 fs and 82 MHz	1–10 μm	[98]
Ag	-	2 μm	[95]
Ag	Ti:sapphire laser (795 nm center wavelength, 11 MHz repetition rate, 50 fs pulse duration)	300 nm	[97]
Ag	KrF excimer laser (wavelength 248 nm, pulse width 30 ns, frequency 10 Hz)	50 nm	[90]
Ag	800 nm femtosecond laser pulse, with a width of 120 fs and mode locked at 82 MHz	50–300 nm	[99]
Cu/Au	a laser with a pulse duration of 6.7 ps and a wavelength of 515 nm	5 μm	[100]
Au	femtosecond laser pulses (center wavelength of 700 nm and a repetition rate of 80 MHz)	3 μm	[91]
Ag	laser (800 nm central wavelength, 23 fs pulse width and 80 MHz repetition rate)	10 μm	[96]

**Table 4 nanomaterials-09-01789-t004:** Summary of 3D micro/nano structures prepared by direct laser writing on organics and polymers.

Materials	Laser Equiment	Feature Size	References
Photoresist SU-8	Ti:sapphire laser system (120 fs, 1 kHz, 800 nm	0.65–1.5 μm	[132]
PEGDA hydrogels	-	5 μm	[133]
Photoresist SU-8	Laser (130 fs pulses, central wavelength 800 nm,100 Hz)	1 μm	[134]
Protein	Nd: YAG laser (a pulse width of 600 ps, a repetition rate of 7.65 kHz)	0.5 μm	[135]
Oligofluorene truxene Photoresist composites	GaN diode laser (15 mW, 374 nm)	2.5 μm	[136]
Photoresists: SU-8, IP-L, IP-G	CW laser operating at 532 nm wavelength	450 nm	[131]
Polyethylene glycol Diacrylate (PEG-DA)	Laser powers 10–20 mW	1 μm	[137]
Hybrid sol-gel resist SZ2080	Picosecond Nd:YVO laser (1064 nm and 532 nm, 8–25 ps and from 0.2 to 1 MHz)	200–500 nm	[138]
Liquid photoresist: IP-DIP	-	10 μm	[145]
Protein	Femtosecond laser (80 MHz, 120 fs pulse width, 780 nm central wavelength)	250 nm	[139]
Polyimide foil	Nd:YVO_4_ MOPA laser (355 nm, 1 MHz, <15 ps.)	100 nm	[140]
Pentaerythritol Tetraacrylate (PETTA)	-	3–10 μm	[142]
Polyimide	Laser beam (780 nm femtosecond laser, 130 fs pulses, 100 MHz repetition rate)	0.7 μm	[144]
Foturan glass	Femtosecond laser beam (1045 nm, a pulse width of 360 fs, 200 kHz)	1–5 μm	[146]
Protein	Femtosecond titanium:sapphire laser (80 MHz, 120 fs, central wavelength: 790 nm)	150 nm	[147]
Polydimethylsiloxane (PDMS)	Femtosecond laser pulses (780 nm central wavelength, 80 MHz repetition rate, 120 fs pulse width)	3.5–20 μm	[148]
SZ2080 material	Laser (300 fs, 200 kHz and 515 nm)	50 μm	[149]
Material SZ2080	femtosecond laser amplified system (515 nm, 300 fs and 200 kHz)	1 μm	[150]
Pentaerythritol triacrylate	Femtosecond laser source (140 fs, 800 nm)	3 μm	[152]

**Table 5 nanomaterials-09-01789-t005:** Summary of 2D/3D micro/nano structures prepared by laser-assisted processing on semiconductor.

Materials	Laser Equiment	Feature Size	References
Si	-	200–400 μm	[189]
GaN	The KrF excimer laser beam (wavelength 248 nm, energy 400 mJ/pulse, frequency 10 Hz)	80 nm	[190]
Ge_1_Sb_2_Te_4_ (GST)	KrF 248 nm excimer laser with a pulse duration of 23 ns	120 nm	[191]
Ge_1_Sb_2_Te_4_ (GST)	Ti:sapphire femtosecond laser (wavelength 800 nm, pulse duration 100 fs, and repetition rate from 1 to 1000 Hz)	50 nm	[192]
Si	Ti:sapphire chirped pulse amplification (CPA) system (1 mJ pulses, 1 kHz, center wavelength of about 800 nm, 100 fs)	40–200 nm	[194]
Si	KrF excimer laser (248 nm)	0.8–4.1 μm	[195]
Si	-	116 μm	[196]
Si	Nd:YAG laser (100 mJ/pulse, pulse duration of 10 ns and frequency of 10Hz)	30 nm	[200]
Si	35 fs pulse width, a central wavelength of 800 nm	313 nm–2 μm	[205]
Si	The laser system (10 fs, center wavelength of 800 nm at an average power of 400 mW, 85 MHz)	130 nm	[206]
Si	Ultraviolet laser (wavelength of 355 nm)	1.1–2.4 μm	[207]
GaN	Ti:sapphire laser system operated at 10 Hz	50 nm	[209]
Si	Ti:sapphire femtosecond laser system (100 fs pulses, 800-nm central wavelength with a repetition rate of 1 kHz)	23 μm	[211]
MoS_2_	Laser power fixed at 10 mW	300 nm	[213]
Si	LIL system with a 325 nm wavelength helium cadmium laser source	160–400 nm	[217]

Laser interference lithography (LIL).

**Table 6 nanomaterials-09-01789-t006:** Summary of feature size for 3D micro/nano structures prepared by laser-assisted processing on SiO_2_.

Laser Equiment	Feature Size	References
Fs laser system (795 nm wavelength pulses,1 kHz, 120 fs)	10 μm	[234]
KrF laser (248 nm, 30 ns)	0.75 μm	[235]
Ti:sapphire laser (0.8 mJ pulses, 150 fs, 1 kHz repetition rate)	150 nm	[237]
Femtosecond laser (1045 nm, 1 MHz, <500 fs, pulse energy of up to 500–600 nJ)	5 μm	[238]
Ti:sapphire (800 nm, 58 fs pulses, 250 kHz)	9 μm	[241]
Femtosecond laser	2.8 μm	[244]
Ti:sapphire laser (800 nm, 250 kHz, 150 fs, 4 µJ per pulse)	1 μm	[246]
Ti:sapphire (800 nm, 40 fs pulses, 1 kHz repetition rate)	80 μm	[247]
800 fs laser pulses, 1030 nm, 200 kHz, maximum average power of 40 W	30 μm	[249]

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
