# Peer review of "The Fabrication of Micro/Nano Structures by Laser Machining"

_nanomaterials, 2019, doi:10.3390/nano9121789_

Round 1

Reviewer 1 Report

It is very well written review mainly on femtosecond laser fabrication of micro/nanostructures of various materials which gives valuable information on principles, the feature sizes and possible applications.  It covers mainly advanced laser technologies often joint with other technologies suitable for fabrication of complicated micro/nanostructures for different applications.

There are some mistakes and minor spell check is required. Some places where it must be done are listed below.

Page 4

...Figure 2c must be Figure 1c

Page 7

Figure 3f shows the change in height and width and pulse energy.

Figure 3f shows the change in height and width depending on pulse energy.

Page 8,

Fig. 3

Metel film on silica

Must be:

Metal film on silica

Zhang et al. Describe

Must be:

Zhang et al. Described

Page 9

In summary, the use of spatially modulated..

Must be:

In summary by  the use of spatially modulated..

The metal substrate was processed with picosecond laser pulses in different laser machining parameters

Maybe better:

The metal substrate was processed with picosecond laser pulses with different laser machining parameters

Page 10

LIPSS on mental

Must be:

LIPSS on metal

Page 11

Conductive silver wires with a minimum width of 400 nm was generated.

Must be:

Conductive silver wires with a minimum width of 400 nm were generated.

This result meant

This result mean

Page 12 psl

In 2012, A. Kiani et al. reported the concept of generating 3D nanostructured metal alloys by irradiating two or more immiscible materials (such as a mixture of nickel oxide (NiO) and aluminum (Al)) with high repetition of ultrashort laser pulses powder[84].

Probably

In 2012, A. Kiani et al. reported the concept of generating 3D nanostructured metal alloys by irradiating two or more immiscible materials powder (such as a mixture of nickel oxide (NiO) and aluminum (Al)) with high repetition rate ultrashort laser pulses [84].

It can be seen from the preparation process of figure. 5a..

Must be:

It can be seen from the preparation process of Figure. 5a

In 2017, P. Barton et al. employed a femtosecond laser (800nm center

Must be:

In 2017, P. Barton et al. employed a femtosecond laser (800 nm center

Page 34

4.1. laser-induced periodic

4.1. Laser-induced periodic

analysis[155]

analysis [155] – additional space gap is required in many places

Author Response

Response to Reviewer 1 Comments

Point 1:

There are some mistakes and minor spell check is required. Some places where it must be done are listed below.

Page 4

Figure 2c must be Figure 1c

Page 7

Figure 3f shows the change in height and width and pulse energy.

Figure 3f shows the change in height and width depending on pulse energy.

Page 8,

Fig. 3

Metel film on silica

Must be:

Metal film on silica

Zhang et al. Describe

Must be:

Zhang et al. Described

Page 9

In summary, the use of spatially modulated.

Must be:

In summary by the use of spatially modulated.

The metal substrate was processed with picosecond laser pulses in different laser machining parameters

Maybe better:

The metal substrate was processed with picosecond laser pulses with different laser machining parameters

Page 10

LIPSS on mental

Must be:

LIPSS on metal

Page 11

Conductive silver wires with a minimum width of 400 nm was generated.

Must be:

Conductive silver wires with a minimum width of 400 nm were generated.

This result meant

This result mean

Page 12 psl

In 2012, A. Kiani et al. Reported the concept of generating 3D nanostructured metal alloys by irradiating two or more immiscible materials (such as a mixture of nickel oxide (NiO) and aluminum (Al)) with high repetition of ultrashort laser pulses powder [84].

Probably

In 2012, A. Kiani et al. Reported the concept of generating 3D nanostructured metal alloys by irradiating two or more immiscible materials powder (such as a mixture of nickel oxide (NiO) and aluminum (Al)) with high repetition rate ultrashort laser pulses [84].

It can be seen from the preparation process of figure. 5a.

Must be:

It can be seen from the preparation process of Figure. 5a

In 2017, P. Barton et al. Employed a femtosecond laser (800nm ​​center

Must be:

In 2017, P. Barton et al. Employed a femtosecond laser (800 nm center

Page 34

4.1. Laser-induced periodic

4.1. Laser-induced periodic

Response 1:

Thank you for your comments on my mistakes and minor spell check. I have corrected all mistakes you have mentioned. I used the "Track Changes" function in Microsoft Word. So any revisions have been highlighted in my manuscript. Please check my modifications in my manuscript.

Point 2:

analysis [155]

analysis [155] – additional space gap is required in many places

Response 2:

Thank you for your comments on my mistakes. I have corrected all mistakes you have mentioned. I used the "Track Changes" function in Microsoft Word. So any revisions have been highlighted in my manuscript. Please check my modifications in my manuscript.

Reviewer 2 Report

Review Report: Micro/Nanostructures by Laser Machining

The paper presents an extensive review of micro and nanofabrication of different structures using laser assisted techniques. The paper is definitely a very good contribution to the scientific community where most of the relevant literatures are covered. I recommend the manuscript for publication pending the following revisions.

Since this is a review article, I hope the authors have received permission from different publishers to use the figures in already published journal articles.

For each new section, for example "Nanowires", authors may consider making a comparison between the feature size, shape etc using the laser techniques and similar structure fabricated using nanofabrication tools. If this can be added to each section, that would be greatly beneficial to readers.

Line 25: Please replace surface plasma resonance with Surface plasmon resonances

Line: 42: Please replace some properties with different properties.

There are many examples of nanoscale features fabricated using conventional lithography processes. I would recommend authors to add some more relevant references such as below in line 66:

Line 66 - 68: Lithography has achieved unique success in creating 2D patterns with feature sizes in the range of nanometers to a few microns [ R1, R2, R3, R4, R5] , and is a key technology that has enabled Moore's Law to expand and revolutionize computing speed.[22]

R1: H Butt,  et al Enhanced reflection from arrays of silicon based inverted nanocones, Applied Physics Letters 99 (13), 133105 (2011)

R2: Kanghee Won,  et al Electrically Switchable Diffraction Grating Using a Hybrid Liquid Crystal and Carbon Nanotube‐Based Nanophotonic Device, Advanced Optical Materials 1 (5), 368-373 (2013)

R3: R Rajasekharan,  et al Electrically reconfigurable nanophotonic hybrid grating lens array, Applied Physics Letters 96 (23), 233108 (2010)

R4: Haider Butt,  et al Continuous diffraction patterns from circular arrays of carbon nanotubes, Applied Physics Letters 101 (25), 251102 (2012)

R5: Q Dai, et al Ultrasmall microlens array based on vertically aligned carbon nanofibers, Small 8 (16), 2501-2504 (2012)

Please add references at the end of each figure captions except on authors' work. In 567: prosbect should be replaced with prospect In 626: please remove 'was' Line 658: Please remove 'our study of' Line 661-663 should be : As shown in figure 9, three-dimensional photonic crystal structures fabricated by the TPP technique demonstrated the advantage of zirconium-containing materials over other TPP materials for their negligible shrinkage Line: 969: Please remove 'was investigated' Line 971: Please remove 'was used' Line 1818: Symbol error may be corrected. Line 1908: Please replace "excited" with exhibited

Author Response

Response to Reviewer 2 Comments

Point 1:

Since this is a review article, I hope the authors have received permission from different publishers to use the figures in already published journal articles.

Response 1:

Thank you for your comments. Since I can’t obtain the copyright permissions of Figure 5(c, d), Figure 10 (c, d), Figure 16, Figure 19, Figure 26 (a-e), Figure 28 (a-f). I don’t use them in my manuscript. And after my revision, Figure 16 and Figure 19 are replaced by new Figures. Now, all copyright permissions of 30 Figures have been obtained.

Point 2:

For each new section, for example "Nanowires", authors may consider making a comparison between the feature size, shape etc using the laser techniques and similar structure fabricated using nanofabrication tools. If this can be added to each section, that would be greatly beneficial to readers.

Response 2:

Thank you for your comments. For each new section, if I make a comparison between using the laser techniques and other nanofabrication tools, my manuscript will be richer in content. And I think it will be more beneficial to readers. However, my initial thought is to cover mainly advanced laser technologies in metal, organics or polymers, semiconductors, glass, oxides, carbon materials, piezoelectric materials. And my focus is to give comparison of different laser machining techniques.

Point 3:

Line 25: Please replace surface plasma resonance with Surface plasmon resonances.

Line: 42: Please replace some properties with different properties.

Response 3:

Thank you for your comments on my mistakes. I have corrected all mistakes you have mentioned. And I used the "Track Changes" function in Microsoft Word. So any revisions have been highlighted in my manuscript. Please check my modifications in my manuscript.

Point 4:

There are many examples of nanoscale features fabricated using conventional lithography processes. I would recommend authors to add some more relevant references such as below in line 66:

Line 66 - 68: Lithography has achieved unique success in creating 2D patterns with feature sizes in the range of nanometers to a few microns [ R1, R2, R3, R4, R5], and is a key technology that has enabled Moore's Law to expand and revolutionize computing speed. [22]

R1: H Butt, et al Enhanced reflection from arrays of silicon based inverted nanocones, Applied Physics Letters 99 (13), 133105 (2011)

R2: Kanghee Won, et al Electrically Switchable Diffraction Grating Using a Hybrid Liquid Crystal and Carbon Nanotube‐Based Nanophotonic Device, Advanced Optical Materials 1 (5), 368-373 (2013)

R3: R Rajasekharan, et al Electrically reconfigurable nanophotonic hybrid grating lens array, Applied Physics Letters 96 (23), 233108 (2010)

R4: Haider Butt et al Continuous diffraction patterns from circular arrays of carbon nanotubes, Applied Physics Letters 101 (25), 251102 (2012)

R5: Q Dai, et al Ultrasmall microlens array based on vertically aligned carbon nanofibers, Small 8 (16), 2501-2504 (2012)

Response 4:

Thank you for your comments. There many examples of nanoscale features fabricated using conventional lithography processes. Except for examples you advised, I have also cited other papers on lithography techniques.

Point 5:

Please add references at the end of each figure captions except on authors' work. In 567: prosbect should be replaced with prospect in 626: please remove 'was' Line 658: Please remove 'our study of' Line 661-663 should be: As shown in figure 9, three-dimensional photonic crystal structures fabricated by the TPP technique demonstrated the advantage of zirconium-containing materials over other TPP materials for their negligible shrinkage Line: 969: Please remove 'was investigated' Line 971: Please remove 'was used' Line 1818: Symbol error may be corrected. Line 1908: Please replace "excited" with exhibited.

Response 5:

Thank you for your comments. I have added references at the end of each figure captions. And the mistakes you mentioned have been revised. Please check my revisions in my manuscript.
